# Decoupled Subgraph Federated Learning

**Javad Aliakbari**[1]    **Johan Östman**[2]    **Alexandre Graell i Amat**[1]

[1]Chalmers University of Technology    [2]AI Sweden

## Abstract

We address the challenge of federated learning on graph-structured data distributed across multiple clients. Specifically, we focus on the prevalent scenario of interconnected subgraphs, where interconnections between different clients play a critical role. We present a novel framework for this scenario, named FEDSTRUCT, that harnesses deep structural dependencies. To uphold privacy, unlike existing methods, FEDSTRUCT eliminates the necessity of sharing or generating sensitive node features or embeddings among clients. Instead, it leverages explicit global graph structure information to capture inter-node dependencies. We validate the effectiveness of FEDSTRUCT through experimental results conducted on six datasets for semi-supervised node classification, showcasing performance close to the centralized approach across various scenarios, including different data partitioning methods, varying levels of label availability, and number of clients.

## 1 Introduction

Many real-world data are graph-structured, where nodes represent entities and edges capture their relationships. For example, in anti-money laundering, nodes symbolize accounts and edges correspond to confidential transactions.

Graph neural networks (GNNs) are specialized neural networks for graph-structured data, showing success in fields such as drug discovery, social networks, or traffic flow modeling (Stokes et al., 2020; Fan et al., 2019; Jiang and Luo, 2022). In many practical applications, graph data is inherently distributed across multiple clients, i.e., the global graph encompasses multiple, non-overlapping subgraphs. For example, in anti-money laundering, local graphs represent internal transactional networks of financial institutions.

For this and many other applications, data sharing among clients is often restricted due to privacy, regulations, or proprietary restrictions. Federated learning (FL) (McMahan et al., 2017) offers a way to utilize global graph-structured data while maintaining data privacy. Various flavors of federated GNNs exist (Liu et al., 2022). This paper focuses on one of the most prevalent scenarios, **subgraph federated learning (SFL)** (Zhang et al., 2021), where clients hold disjoint subgraphs that together form a global graph with interconnections between the different local subgraphs.

Training a GNN involves aggregating feature representations of neighboring nodes to generate more expressive embeddings (Kipf and Welling, 2017; Hamilton et al., 2017; Veličković et al., 2018). In SFL, a key challenge is facilitating training when some neighboring nodes, whose information is crucial for training, reside in other clients without sharing raw data across clients. This setup falls under the well-known **communication-privacy-accuracy trilemma** (Chen et al., 2020) which necessitates solutions that preserve the privacy of local data and supports minimal communication overhead between clients and the central server, while achieving high accuracy for the end-to-end task.

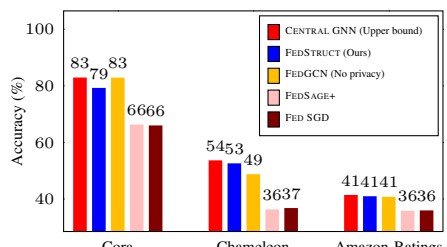

**Figure 1:** Node classification accuracy. For all datasets, FEDSTRUCT exhibits performance close to the centralized setting (CENTRAL GNN).

Several approaches have been proposed to address this trilemma (Zhang et al., 2021) (FEDSAGE+), (Peng et al., 2022) (FEDNI), (Chen et al., 2021; Du and Wu, 2022; Lei et al., 2023) (Yao et al., 2024)

(FEDGCN), (Baek et al., 2023) (FEDPUB). However, except (Baek et al., 2023), these methods rely on sharing node features or embeddings between clients, which raises significant **privacy concerns**. Moreover, there are challenges related to **performance**, as the global model needs to accurately aggregate distributed knowledge, and **communication cost**, as transmitting features or embeddings between clients can be resource-intensive.

**To address privacy risks**, inspired by Cui et al. (2022), we observe that the global graph structure alone can be highly informative and is less sensitive than node features. This opens up the possibility of training robust classifiers without exposing raw data, **achieving high performance** that surpasses that of standard FL.

**Our contribution.** We address the problem of subgraph FL for node classification within a global graph containing multiple, non-overlapping, subgraphs belonging to different clients. More precisely, we consider the prevalent scenario where cross-subgraph interconnections are known (such as in transaction networks). However, we assume that neither the server nor the clients have knowledge of the global graph connections or the node features, i.e., the node features and intra-connections for each subgraph remain private. Our contributions include:

- Building on the observation above, we propose a novel SFL framework, named FEDSTRUCT, that exploits deep structural dependencies and tackles the key challenges of privacy, performance, and communication cost. To **safeguard privacy**, FEDSTRUCT decouples graph structure and node feature information: Unlike existing approaches, FEDSTRUCT eliminates the need for sharing or generating node features or embeddings. Instead, it leverages global graph structure information to capture inter-node dependencies among clients. FEDSTRUCT minimizes structural information exchanged between clients, limiting privacy leakage and **reducing communication complexity** while achieving **utility close to a centralized approach**.
- We introduce a method to generate task-dependent node structure embeddings, coined HOP2VEC, that adapts to the graph and demonstrates competitive performance compared to task-agnostic methods such as NODE2VEC (Grover and Leskovec, 2016), graphlet counting (Pržulj, 2007), or the technique used in FEDSTAR (Tan et al., 2023).
- By integrating a decoupled GCN and leveraging deep structural dependencies within the global graph, we effectively tackle the semi-supervised learning scenario. Further, FEDSTRUCT relies on non-local neighbor-extension and inter-layer combination, techniques commonly used to handle heterophilic graphs (Zheng et al., 2022). To the best of our knowledge, **FEDSTRUCT is the first SFL framework capable of handling heterophilic graphs**.
- We validate the effectiveness of FEDSTRUCT on six datasets for semi-supervised node classification, showcasing excellent performance close to a centralized approach in multiple scenarios with different data partitioning methods, availability of training labels, and number of clients. Particularly, FEDSTRUCT yields outstanding performance in scenarios with limited number of labeled training nodes. In heavily semi-supervised settings, it significantly outperforms both FEDSAGE+ and FEDPUB. Moreover, it achieves peformance close to that of FEDCGN (which does not provide privacy and assumes server access to the global adjacency matrix) (see Figure 1). The source code is publicly available in the Github Link.

## 2 RELATED WORK

**SFL with no knowledge of cross-subgraph interconnections.** Relevant works include (Zhang et al., 2021) (FEDSAGE+), (Peng et al., 2022) (FEDNI), (Zhang et al., 2024) (FEDDEP), (Baek et al., 2023) (FEDPUB), and (Zhang et al., 2022; Liu et al., 2023). FEDSAGE+, the first method for subgraph FL, generates features for missing 1-hop neighbors across subgraphs using a variational autoencoder. FEDNI extends this by using a GAN to generate higher-quality node features. The in-painting idea is further expanded to handle heterogeneous graphs in (Zhang et al., 2022) and missing links in (Liu et al., 2023). FEDDEP builds on FEDSAGE+ by generating embeddings that capture deeper structural information (up to $k$-hop neighbors). A limitation of in-painting methods, like FEDSAGE+ and FEDNI, is the unpredictable quality of the generated features, which can either lead to poor models or expose sensitive information through confident predictions. FEDPUB avoids in-painting by employing personalized aggregation based on the functional similarity between client models.

**SFL with knowledge of cross-subgraph interconnections.** Relevant works include (Yao et al., 2024) (FEDGCN), (Lei et al., 2023) (FEDCOG), and (Chen et al., 2021; Du and Wu, 2022). FEDGCN

securely transmits cross-client neighbor information once before training, preventing the server from accessing local data but exposes aggregated node features to neighboring clients. This can risk data leakage since node features often contain meaningful patterns. FEDCOG decouples local subgraphs into internal and border graphs, with graph convolution divided between them. It requires sharing intermediate embeddings, effectively performing a convolution on the global graph. Other methods employ sampling techniques to address the challenges of SFL (Chen et al., 2021; Du and Wu, 2022).

In summary, all prior approaches rely on sharing original or generated node features or embeddings (except FEDPUB) and assume homophily, which often does not hold in real-world settings.

**Structural information in GNNs.** Recent studies have revealed limitations in the capacity of GNNs to capture the structure of the underlying graph. To overcome this obstacle, some works explicitly incorporate structural information into the learning, showing superior performance compared to standard GNNs (Bouritsas et al., 2023; Tan et al., 2023). To increase GNN's representation power, Bouritsas et al. (2023) introduces structural information to the aggregation function. The authors demonstrate that the proposed architecture is strictly more expressive than standard GNNs. FED-STAR (Tan et al., 2023) presents an FL scheme where clients share explicit structural information to enhance local performance. No work has leveraged explicit structural information in SFL.

## 3 PRELIMINARIES

**Graph notation.** We consider a graph denoted by $\mathcal{G} = (\mathcal{V}, \mathcal{E}, \boldsymbol{X}, \boldsymbol{Y})$, where $\mathcal{V}$ is the set of $n$ nodes, $\mathcal{E}$ the set of edges, $\boldsymbol{X} \in \mathbb{R}^{n \times d}$ the node feature matrix, and $\boldsymbol{Y} \in \mathbb{R}^{n \times c}$ the label matrix. For each node $v \in \mathcal{V}$, we denote by $\boldsymbol{x}_v \in \mathbb{R}^d$ its corresponding feature vector and by $\boldsymbol{y}_v \in \mathbb{R}^c$ its corresponding one-hot encoded label vector. We consider a semi-supervised learning scenario and denote by $\tilde{\mathcal{V}} \subseteq \mathcal{V}$ the set of nodes that possess labels. The labels of the remaining nodes are set to $\boldsymbol{0}$. We also denote by $\mathcal{N}_{\mathcal{G}}(v) = \{u | (u, v) \in \mathcal{E}\}$ the neighbors of node $v$.

For a given matrix $\boldsymbol{M}$, let $M_{uv}$ be its $(u, v)$-th element. The topological information of the whole graph is described by the adjacency matrix $\boldsymbol{A} \in \mathbb{R}^{n \times n}$, where $A_{uv} = 1$ if $(u, v) \in \mathcal{E}$. We define the diagonal matrix of node degrees as $\boldsymbol{D} \in \mathbb{R}^{n \times n}$, where $D_{uu} = \sum_v A_{uv}$. Furthermore, we denote by $\tilde{\boldsymbol{A}} = \boldsymbol{A} + \boldsymbol{I}$ the self-loop adjacency matrix, and by $\hat{\boldsymbol{A}} = \tilde{\boldsymbol{D}}^{-1} \tilde{\boldsymbol{A}}$ the normalized self-loop adjacency matrix, where $\tilde{d}_{uu} = \sum_{v \in \mathcal{V}} \tilde{A}_{uv}$. We also define $[m] = \{1, \ldots, m\}$.

**GNNs.** Modern GNNs use neighborhood aggregation followed by a learning transformation to iteratively update node representations at each layer. Let $\boldsymbol{h}_v^{(l)}$ be the feature embedding of node $v$ at layer $l$, with $\boldsymbol{h}_v^{(0)} = \boldsymbol{x}_v$. At layer $l \in [L]$ of the GNN we have $\boldsymbol{h}_{\mathcal{N}_{\mathcal{G}}(v)}^{(l)} = \text{AGG}^{(l)}\left(\left\{\boldsymbol{h}_u^{(l-1)}, \forall u \in \mathcal{N}_{\mathcal{G}}(v)\right\}\right)$ and $\boldsymbol{h}_v = \text{UPD}^{(l)}\left(\boldsymbol{h}_v^{(l-1)}, \boldsymbol{h}_{\mathcal{N}_{\mathcal{G}}(v)}^{(l)}, \boldsymbol{\Theta}^{(l)}\right)$, where $\boldsymbol{\Theta}^{(l)}, \text{AGG}^{(l)}$ and $\text{UPD}^{(l)}$ denote the learnable weight matrix, aggregation function, and update function associated with layer $l$, respectively.

**Decoupled GCNs.** A significant limitation of GNNs is over-smoothing (Liu et al., 2020; Dong et al., 2021), which results in performance degradation when multiple layers are applied. Over-smoothing is characterized by node embeddings becoming inseparable, making it challenging to distinguish between them. This phenomenon is attributed to the interweaving of the propagation and update steps within each GNN layer (Liu et al., 2020; Dong et al., 2021). To address over-smoothing, a well-known solution (hereafter referred to as *decoupled GCN*) is to decouple the propagation and update steps (Liu et al., 2020; Dong et al., 2021). Let $\boldsymbol{f}_{\boldsymbol{\theta}}(\cdot)$ be a multi layer perceptron (MLP) network with learning parameters $\boldsymbol{\theta}$. A decoupled GCN can be described with the model

$$\boldsymbol{H}^{(L)} = \bar{\boldsymbol{A}} \boldsymbol{F}_{\boldsymbol{\theta}}(\boldsymbol{X}), \tag{1}$$

where $\boldsymbol{F}_{\boldsymbol{\theta}}(\boldsymbol{X}) = ||(\boldsymbol{f}_{\boldsymbol{\theta}}(\boldsymbol{x}_v)^{\mathsf{T}} \quad \forall v \in \mathcal{V})$, with $||$ being the concatenation operation, and $^{\mathsf{T}}$ denotes the transpose operation. $\bar{\boldsymbol{A}}$ is the $L$-hop *combined* adjacency matrix and it can be computed as

$$\bar{\boldsymbol{A}} = \sum_{l=1}^{L} \beta_l \hat{\boldsymbol{A}}^l. \tag{2}$$

The elements of $\bar{\boldsymbol{A}}$ reflect the proximity of two nodes in the graph, with $\beta_l$ determining the contribution of each hop. Parameters $\{\beta_l\}_{l=1}^L$ can be set manually or learned during training.

## 4 System Model

We consider a scenario where data is structured according to a *global graph* $\mathcal{G} = (\mathcal{V}, \mathcal{E}, \boldsymbol{X}, \boldsymbol{Y})$, which is distributed among $K$ clients such that each client owns a smaller *local* subgraph. We denote by $\mathcal{G}_i = (\mathcal{V}_i, \mathcal{V}_i^*, \mathcal{E}_i, \mathcal{E}_i^*, \boldsymbol{X}_i, \boldsymbol{Y}_i)$ the subgraph of client $i$, where $\mathcal{V}_i \subseteq \mathcal{V}$ is the set of $n_i$ nodes that reside in client $i$, referred to as *internal nodes*, for which client $i$ knows their features. $\mathcal{V}_i^*$ is the set of nodes that do not reside in client $i$ but have at least one connection to nodes in $\mathcal{V}_i$. We call these nodes *external nodes*. Importantly, client $i$ does not have access to the features of nodes in $\mathcal{V}_i^*$. Furthermore, $\mathcal{E}_i$ represents the set of edges between nodes owned by client $i$ (intra-connections), $\mathcal{E}_i^*$ the set of edges between nodes of client $i$ and nodes of other clients (interconnections), $\boldsymbol{X}_i \in \mathbb{R}^{n_i \times d}$ the node feature matrix, and $\boldsymbol{Y}_i \in \mathbb{R}^{n_i \times c}$ the label matrix for the nodes within subgraph $\mathcal{G}_i$, and we denote by $\tilde{\mathcal{V}}_i$ the set of nodes that possess labels. Similar to graph $\mathcal{G}$, $\mathcal{N}_{\mathcal{G}_i}(v)$, $\boldsymbol{A}^{(i)}$, and $\boldsymbol{D}^{(i)}$ denote the set of local neighbors, the local adjacency matrix, and the local diagonal matrix for subgraph $\mathcal{G}_i$.

### 4.1 Federated Learning

The FL problem can be formalized as learning the model parameters that minimize the aggregated loss across clients,

$$\boldsymbol{\theta}^* = \arg\min_{\boldsymbol{\theta}} \; \mathcal{L}(\boldsymbol{\theta}), \tag{3}$$

with

$$\mathcal{L}(\boldsymbol{\theta}) = \frac{1}{|\tilde{\mathcal{V}}|} \sum_{i=1}^{K} \mathcal{L}_i(\boldsymbol{\theta}), \quad \text{and} \quad \mathcal{L}_i(\boldsymbol{\theta}) = \sum_{v \in \tilde{\mathcal{V}}_i} \text{CE}(\boldsymbol{y}_v, \hat{\boldsymbol{y}}_v), \tag{4}$$

where CE is the cross-entropy loss function between the true label $\boldsymbol{y}_v$ and the predicted label $\hat{\boldsymbol{y}}_v$.

The model $\boldsymbol{\theta}$ is trained iteratively over multiple epochs. At each epoch, the clients compute the local gradients $\nabla_{\boldsymbol{\theta}} \mathcal{L}_i(\boldsymbol{\theta})$ and send them to the central server. The server updates the model through gradient descent,

$$\boldsymbol{\theta} \leftarrow \boldsymbol{\theta} - \lambda \nabla_{\boldsymbol{\theta}} \mathcal{L}(\boldsymbol{\theta}), \quad \nabla_{\boldsymbol{\theta}} \mathcal{L}(\boldsymbol{\theta}) = \frac{1}{|\tilde{\mathcal{V}}|} \sum_{i=1}^{K} \nabla_{\boldsymbol{\theta}} \mathcal{L}_i(\boldsymbol{\theta}), \tag{5}$$

where $\lambda$ is the learning rate.

## 5 FedStruct: Structure-Exploiting Subgraph Federated Learning

In this section, we introduce FedStruct, a novel SFL framework designed to leverage inter-node dependencies among clients while safeguarding privacy. The central concept of FedStruct is to harness explicit information about the global graph's underlying structure to improve node label prediction while ensuring that neither the server nor the clients have access to the node features. More precisely, at each client $i \in [K]$, node prediction is performed for a node $v \in \mathcal{V}_i$ as

$$\hat{\boldsymbol{y}}_v = \text{softmax}\Big(\boldsymbol{h}_v + \boldsymbol{z}_v\Big), \tag{6}$$

where $\boldsymbol{h}_v$ is the *node feature embedding* (NFE) and $\boldsymbol{z}_v$ is the *node structure embedding* (NSE), which encodes structural information of the nodes. Also $\boldsymbol{Z} = ||\big((\boldsymbol{z}_v^\mathsf{T}, \quad \forall v \in \mathcal{V})$, where $\boldsymbol{Z}$ is the *structure embedding matrix* (SEM) containing NSEs.

The NFEs $\boldsymbol{h}_v$ are computed locally at each client by a GNN model based on the local node features $\boldsymbol{X}_i$ and local connection $\mathcal{E}_i$ as

$$\boldsymbol{h}_v = \text{FGNN}_{\boldsymbol{\theta}_\mathrm{f}}(\boldsymbol{X}_i, \mathcal{E}_i, v) = f_{\boldsymbol{\theta}_\mathrm{f}}(v). \tag{7}$$

The NSEs $\boldsymbol{z}_v$ are generated based on the structural information of the global graph and clients need to collaborate to obtain them. Their use is anticipated to enhance node classification compared to the case of a classifier solely relying on NFEs. We describe how to generate NSEs in Sec. 5.1.

The FedStruct framework is illustrated in Figure 2 and described in Alg. 1 in App. B.

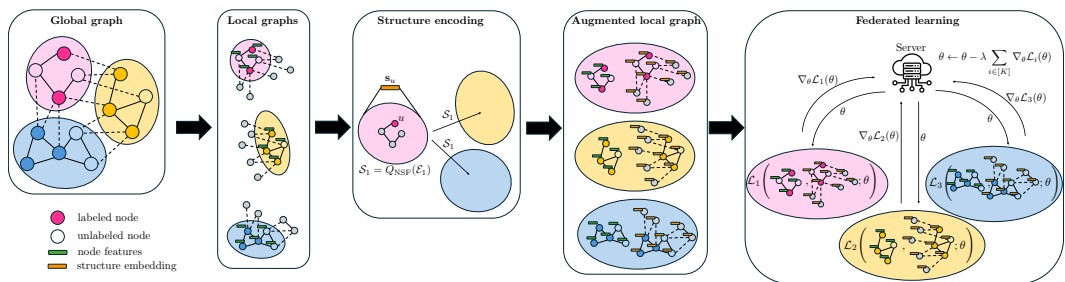

**Figure 2:** General design of the FEDSTRUCT framework. **Global Graph:** underlying graph consisting of interconnected subgraphs. **Local graphs:** clients' subgraphs augmented with external nodes (without features or labels). **Structure encoding:** Generates node structure features for each node and shares them with other clients. **Augmented local graphs:** Generate node feature embeddings and node structure embeddings. **Federated learning:** Federated learning step exploiting node feature embeddings and node structure embeddings.

## 5.1 NODE STRUCTURE FEATURES, NODE STRUCTURE EMBEDDINGS, AND NODE PREDICTION

**Node structure features.** To generate NSEs, our proposed method relies on *node structure features* (NSFs), which encapsulate structural information about the nodes, such as node degree and local neighborhood connection patterns, as well as positional information such as distance to other nodes in the graph. For each node $v \in \mathcal{V}$, the corresponding NSF, $\boldsymbol{s}_v \in \mathbb{R}^{d_s}$, is a function of the set of edges $\mathcal{E}$, $\boldsymbol{s}_v = \boldsymbol{Q}_{\mathsf{NSF}}(v, \mathcal{E})$. We define the matrix containing all NSFs as $\boldsymbol{S} = ||(\boldsymbol{s}_v^\mathsf{T}, \quad \forall v \in \mathcal{V})$. Function $\boldsymbol{Q}_{\mathsf{NSF}}$ can be defined through various node embedding algorithms, e.g., one-hot degree vector (Xu et al., 2019; Cui et al., 2022), graphlet degree vector (GDV) (Pržulj, 2007), NODE2VEC (Grover and Leskovec, 2016), or as in FEDSTAR (Tan et al., 2023) (we will refer to the NSF technique in (Tan et al., 2023) as FEDSTAR). We describe these different node embeddings in App. A. In Sec. 5.5, we introduce a method to generate task-dependent NSFs, which, in contrast to methods such as GDV and NODE2VEC, does not require knowledge of the global graph.

**Node structure embeddings.** Obtaining the NSEs requires structural information from nodes in other clients, which, with a standard GNN would involve numerous communication rounds during the training. By using a decoupled GCN, it is possible to precompute the necessary structural information for generating NSEs, thereby significantly reducing communication overhead. Consequently, FEDSTRUCT leverages a decoupled GCN to define the NSEs. Specifically, for client $i \in [K]$ and node $v \in \mathcal{V}_i$, let $\boldsymbol{g}_{\boldsymbol{\theta}_s}(\boldsymbol{s}_v)$ be an MLP network applied to NSF $\boldsymbol{s}_v$. Using Eq. 1, $\boldsymbol{Z}$ is obtained as

$$\boldsymbol{Z} = \mathrm{sGNN}_{\boldsymbol{\theta}_s}(\boldsymbol{S}, \mathcal{E}) = \bar{\boldsymbol{A}} \, \boldsymbol{G}_{\boldsymbol{\theta}_s}(\boldsymbol{S}) \,, \tag{8}$$

where $\boldsymbol{G}_{\boldsymbol{\theta}_s}(\boldsymbol{S}) = ||(\boldsymbol{g}_{\boldsymbol{\theta}_s}(\boldsymbol{s}_u)^\mathsf{T}, \forall u \in \mathcal{V})$, and $\bar{\boldsymbol{A}}$ is the $L$-hop combined adjacency matrices for graph $\mathcal{G}$ defined in Eq. 2. Note that computing $\boldsymbol{Z}$ requires the $L$-hop combined adjacency matrix $\bar{\boldsymbol{A}}$, which is not locally available. However, since $\bar{\boldsymbol{A}}$ remains static during the training, it can be precomputed once before training begins. We address this in Sec. 5.4.

**Node prediction.** Upon the creation of the NSEs, each client computes the label predictions through Eq. 6. For a node $v \in \tilde{\mathcal{V}}_i$, expanding Eq. 8 leads to

$$\boldsymbol{z}_v = \sum_{u \in \mathcal{V}} \bar{A}_{vu} \boldsymbol{g}_{\boldsymbol{\theta}_s}(\boldsymbol{s}_u) \,, \tag{9}$$

where $\bar{A}_{vu}$ is the $(v, u)$-th element of $\bar{\boldsymbol{A}}$. Using Eq. 9 in Eq. 6 leads to

$$\hat{\boldsymbol{y}}_v = \mathrm{softmax}\Big( \sum_{u \in \mathcal{V}} \bar{A}_{vu} \boldsymbol{g}_{\boldsymbol{\theta}_s}(\boldsymbol{s}_u) + f_{\boldsymbol{\theta}_f}(v) \Big) \,. \tag{10}$$

**Optimization.** We optimize the learning parameters, $\boldsymbol{\theta} = (\boldsymbol{\theta}_f || \boldsymbol{\theta}_s)$, by solving Eqs. 3–4 with $\hat{\boldsymbol{y}}_v$ in Eq. 10 using stochastic gradient descent. The local gradients are provided in Prop. 3, App. C.1.

## 5.2 DECOUPLED GRAPH CONVOLUTIONAL NETWORK AND FILTERING

Using a DECOUPLED GCN not only allows us to precompute $\bar{\boldsymbol{A}}$, but also makes the GNN more adaptable to heterophilic graphs. To illustrate this, we see the decoupled GCN as a filter. Consider the eigendecomposition of the normalized self-loop adjacency matrix $\hat{\boldsymbol{A}}$ (defined in Sec. 3),

$$\hat{\boldsymbol{A}} = \hat{\boldsymbol{U}} \hat{\boldsymbol{\Lambda}} \hat{\boldsymbol{U}}^\mathsf{T} = \sum_{j=1}^{n} \hat{\lambda}_j \hat{\boldsymbol{u}}_j \hat{\boldsymbol{u}}_j^\mathsf{T} \,, \tag{11}$$

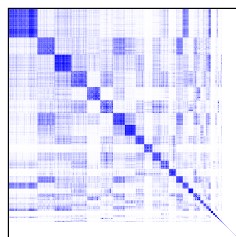 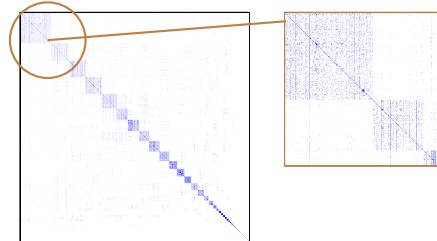

(a) The $L$-hop combined adjacency matrix of the Cora dataset      (b) The pruned $L$-hop combined adjacency matrix of the Cora dataset

**Figure 3:** Comparison between the $L$-hop combined adjacency matrix of the original dataset and the pruned version with $p = 30$. Although some details are lost in the pruning, the main community structure of the graph is preserved.

where $\hat{\boldsymbol{u}}_j$ is the $j$-th eigenvector corresponding to the eigenvalue $\hat{\lambda}_j$. Using Eq. 11 in Eq. 2 we obtain

$$\bar{\boldsymbol{A}} = \hat{\boldsymbol{U}} \left( \sum_{l=0}^{L} \beta_l \hat{\boldsymbol{\Lambda}}^l \right) \hat{\boldsymbol{U}}^{\mathsf{T}} = \hat{\boldsymbol{U}} \bar{\boldsymbol{\Lambda}} \hat{\boldsymbol{U}}^{\mathsf{T}},$$

where $\bar{\boldsymbol{\Lambda}}$ is a diagonal matrix with entries

$$\bar{\boldsymbol{\Lambda}} = T(\hat{\boldsymbol{\Lambda}}) = \mathrm{diag}\left( T(\hat{\lambda}_1), \ldots, T(\hat{\lambda}_n) \right),$$

with $T(x) = \sum_{l=0}^{L} \beta_l x^l$. Therefore, Eq. 8 can be re-written as

$$\boldsymbol{Z} = \hat{\boldsymbol{U}} \left( T(\hat{\boldsymbol{\Lambda}}) \hat{\boldsymbol{S}} \right), \tag{12}$$

where $\hat{\boldsymbol{S}} = \hat{\boldsymbol{U}}^{\mathsf{T}} \boldsymbol{G}_{\boldsymbol{\theta}_{\mathsf{s}}}(\boldsymbol{S})$ is the graph Fourier transform of input signal $\boldsymbol{G}_{\boldsymbol{\theta}_{\mathsf{s}}}(\boldsymbol{S})$. Eq. 12 corresponds to the filtering of the input signal $\boldsymbol{G}_{\boldsymbol{\theta}_{\mathsf{s}}}(\boldsymbol{S})$ with the polynomial filter $T(\hat{\boldsymbol{\Lambda}})$. The parameters $\{\beta_l\}_{l=1}^{L}$ can be used to facilitate the learning process by filtering for the relevant signal. For instance, a low-pass filter is suitable for homophilic graphs, whereas higher frequencies may be required for heterophilic graphs to capture information from nodes multiple hops away.

In practice, calculating the eigenvalue decomposition of $\hat{\boldsymbol{A}}$ in the decentralized setup is expensive as the computational complexity of the eigenvalue decomposition of an $n \times n$ matrix is $\mathcal{O}(n^3)$. Therefore, we use Eq. 8 to calculate $\boldsymbol{Z}$.

### 5.3 PRUNING

The exact computation of the $L$-hop combined adjacency matrix $\bar{\boldsymbol{A}}$ scales with $\mathcal{O}(n^2)$, making it impractical for very large networks. However real-world graph-structured datasets are often sparse. Moreover, as shown in Sec. 5.2, $\bar{\boldsymbol{A}}$ is a filtered version of $\hat{\boldsymbol{A}}$ and is, therefore, of low rank, see Figure 3a. Inspired by this, we introduce an estimation method that significantly reduces the computation complexity of calculating $\bar{\boldsymbol{A}}$ while maintaining its general structure.

The key idea is to apply a pruning technique when calculating $\hat{\boldsymbol{A}}^l$ for $l \in [L_{\mathsf{s}}]$. Specifically, we choose an integer $p \ll n$ and retain only the top $p \cdot n$ values of matrix $\hat{\boldsymbol{A}}^l$ at each iteration, reducing complexity from $L_{\mathsf{s}} \cdot n^2$ to $L_{\mathsf{s}} \cdot p \cdot n$. In Figure 3, we compare $\bar{\boldsymbol{A}}$ to its pruned version ($p = 30$) on Cora. As can be seen, pruning mostly removes off-diagonal connections (links between nodes in different communities), while preserving the main community structure of the graph. As shown in Sec. 6, this does not negatively impact the learning process, as nodes in the same community tend to have similar information, allowing the model to maintain its performance.

### 5.4 STRUCTURAL INFORMATION SHARING

For a given client $i$, training FEDSTRUCT involves computing $\nabla_{\boldsymbol{\theta}} \mathcal{L}_i(\boldsymbol{\theta})$ and predicting node labels $\hat{\boldsymbol{y}}_v, \forall v \in \tilde{\mathcal{V}}_i$. We denote by $\bar{\boldsymbol{A}}^{[i]} = ||(\bar{\boldsymbol{A}}_{v,:}^{\mathsf{T}}, \forall v \in \tilde{\mathcal{V}}_i)$ client $i$'s local partition of $\bar{\boldsymbol{A}}$, where $\bar{\boldsymbol{A}}_{v,:} = ||(\bar{A}_{vu}, \forall u \in \mathcal{V})$. By observing that Eq. 10 depends only on $\bar{\boldsymbol{A}}^{[i]}$ and $\boldsymbol{S}$, we have the following proposition.

**Proposition 1.** *For client $i$, training* FEDSTRUCT, *i.e., computing $\{\hat{\boldsymbol{y}}_v, \ \forall v \in \tilde{\mathcal{V}}_i\}$ and $\nabla_{\boldsymbol{\theta}}\mathcal{L}_i(\boldsymbol{\theta})$, requires only the local partition $\bar{\boldsymbol{A}}^{[i]}$ and $\boldsymbol{S}$.*

Prop. 1 (proved in App. C.2) states that clients do not require knowledge of the global adjacency matrix to predict a node's label or compute gradients. Instead, only the local partition of the global $L$-hop combined adjacency matrix, $\bar{\boldsymbol{A}}^{[i]}$, is required to operate FEDSTRUCT. Combined with the heavy pruning operation, which effectively prunes $\bar{\boldsymbol{A}}^{[i]}$, this approach significantly limits the structural information shared between clients. Optionally, homomorphic encryption may be used. As a result, it is very difficult to reconstruct other clients' adjacency matrices from the pruned $\bar{\boldsymbol{A}}^{[i]}$, enhancing privacy. Additionally, due to graph isomorphism, $\bar{\boldsymbol{A}}^{[i]}$ cannot be leveraged to uniquely reconstruct other clients' adjacency matrices, further reinforcing privacy. Clients have to collaboratively calculate $\bar{\boldsymbol{A}}^{[i]}$. In App. D, we provide an algorithm and an example to obtain $\bar{\boldsymbol{A}}^{[i]}$ before training begins.

### 5.5 HOP2VEC: TASK-DEPENDENT NODE STRUCTURE EMBEDDING

Well-known NSF methods like NODE2VEC and GDV require knowledge of the $L$-hop neighborhood of a node, are task agnostic, and hinge on a homophily assumption (see App. A). To circumvent these shortcomings, we propose HOP2VEC, a novel method to generate task-dependent NSFs that captures structural information beyond direct neighbors without requiring knowledge of the $L$-hop neighborhood and is applicable to heterophilic graphs.

The idea behind HOP2VEC is to view NSFs as learnable features. To this end, NSFs are randomly initialized and updated during the training of FEDSTRUCT. More precisely, for fixed $\boldsymbol{\theta}$, the NSFs are optimized by solving

$$\boldsymbol{S}^* = \arg\min_{\boldsymbol{S}} \mathcal{L}(\boldsymbol{\theta}, \boldsymbol{S}) \,.^1 \tag{13}$$

The optimization is carried out through gradient descent,

$$\boldsymbol{S} \leftarrow \boldsymbol{S} - \lambda_{\mathsf{s}} \nabla_{\boldsymbol{S}} \mathcal{L}(\boldsymbol{\theta}, \boldsymbol{S}) \,, \tag{14}$$

where $\lambda_{\mathsf{s}}$ denotes the learning rate.

Using HOP2VEC, we note that the learning of $\boldsymbol{g}_{\boldsymbol{\theta}_{\mathsf{s}}}(\boldsymbol{s}_u)$ (see Sec. 5.1, e.g., Eq. 8) is equivalent to the learning of $\boldsymbol{s}_u$, i.e., the purpose of passing the NSFs through the sGNN is to generate task-dependent NSEs. Since the NSFs $\{\boldsymbol{s}_u\}_{u \in \mathcal{V}}$ are subject to optimization, we can integrate the learning parameters of $\boldsymbol{g}_{\boldsymbol{\theta}_{\mathsf{s}}}(\boldsymbol{s}_u)$ into $\boldsymbol{s}_u$ and rewrite Eq. 10 as

$$\hat{\boldsymbol{y}}_v = \mathrm{softmax}\Big(\sum_{u \in \mathcal{V}} \bar{A}_{vu} \boldsymbol{s}_u + f_{\boldsymbol{\theta}_{\mathsf{f}}}(v)\Big) \,. \tag{15}$$

To operate FEDSTRUCT, as in Sec. 5.4, client $i$ only need access to $\bar{\boldsymbol{A}}^{[i]}$ as shown in Prop. 2.

**Proposition 2.** *Let $\mathcal{L}_i(\boldsymbol{\theta}, \boldsymbol{S})$ in Eq. 4 use the cross-entropy loss and let $\boldsymbol{S} = ||(\boldsymbol{s}_q^{\mathsf{T}}, \ \ \forall q \in \mathcal{V})$ represent the matrix containing NSFs generated by* HOP2VEC. *Additionally, let $\hat{\boldsymbol{y}}_v$ be as in Eq. 15. The gradient $\nabla_{\boldsymbol{S}}\mathcal{L}_i(\boldsymbol{\theta}, \boldsymbol{S})$ is given by*

$$\nabla_{\boldsymbol{S}}\mathcal{L}_i(\boldsymbol{\theta}, \boldsymbol{S}) = \sum_{v \in \tilde{\mathcal{V}}_i} \bar{\boldsymbol{A}}_{v,:}(\hat{\boldsymbol{y}}_v - \boldsymbol{y}_v)^{\mathsf{T}} \,. \tag{16}$$

*Proof.* See App. C.3. ∎

The FEDSTRUCT framework with HOP2VEC is described in Alg. 2 in App. B.

### 5.6 COMMUNICATION COMPLEXITY

In Table 1, we present the communication complexity of FEDSTRUCT alongside that of FEDSAGE+. We divide the communication complexity into two parts: a pre-training component, accounting for events taking place before the training is initiated, and an online component accounting for the actual training phase. In the table, parameters $E$, $K$, and $n$ represent the number of training rounds, clients,

---

[1]In Eq. 13 we made explicit that the loss function $\mathcal{L}(\boldsymbol{\theta})$ (introduced in Eq. 3 and Eq. 4) is contingent on $\boldsymbol{S}$ through the estimated labels $\hat{\boldsymbol{y}}_v$ (see Eq. 10).

**Table 1:** Communication Complexity of FEDSTRUCT.

| ALGORITHM | PRE-TRAINING | TRAINING |
|---|---|---|
| FEDSTRUCT | $\mathcal{O}(K \cdot d \cdot n + L_\mathsf{s} \cdot K \cdot p \cdot n)$ | $\mathcal{O}(E \cdot K \cdot |\boldsymbol{\theta}|)$ |
| FEDSTRUCT + HOP2VEC | $\mathcal{O}(L_\mathsf{s} \cdot K \cdot p \cdot n)$ | $\mathcal{O}(E \cdot K \cdot |\boldsymbol{\theta}| + E \cdot K \cdot d \cdot n)$ |
| FEDSAGE+ | $0$ | $\mathcal{O}(E \cdot K^2 \cdot |\boldsymbol{\theta}| + E \cdot K \cdot d \cdot n)$ |

and nodes in the graph, respectively. Parameter $d$ is the node feature size (all feature dimensions are assumed to be equal to $d$), and $|\boldsymbol{\theta}|$ is the number of model parameters (all model parameters are assumed to be of the same order).

As seen in Table 1, the online complexity of FEDSTRUCT is significantly lower than that of FED-SAGE+, which is on the order of $n$. The online complexity of FEDSTRUCT with HOP2VEC, however, is also on the order of $n$. The pre-training complexity of FEDSTRUCT is on the order of $L_\mathsf{s} \cdot p \cdot n$. Overall the complexity of FEDSTRUCT is the same of FEDSAGE+, i.e., on the order of $n$.

# 6 EVALUATION

To demonstrate the performance and versatility of FEDSTRUCT, we conduct experiments on six datasets pertaining to node classification under scenarios with varying i) amounts of labeled training nodes, ii) number of clients, iii) number of layers, and iv) data partitioning methods . The interconnections between clients heavily depend on the latter. The datasets considered are: Cora (Sen et al., 2008), Citeseer (Sen et al., 2008), Pubmed (Namata et al., 2012), Chameleon (Pei et al., 2020), Amazon Photo (Shchur et al., 2018), and Amazon Ratings (Platonov et al., 2023). Statistics of the different datasets are provided in App. E.

**Experimental setting.** We focus on a strongly semi-supervised setting where data is split into training, validation, and test sets containing $10\%$, $10\%$, and $80\%$ of the nodes, respectively. We artificially partition the datasets into interconnected subgraphs that are then allocated to the clients. Here, we consider a random partitioning, which assigns nodes to subgraphs uniformly at random. This partitioning constitutes a very challenging setting as the number of interconnections is high and, hence, the learning scheme must exploit such connections, and is relevant in, e.g., transaction networks, where accounts do not necessarily have a preference to interact with nodes in the same subgraph. In App. F, we also provide details and results for two other partitionings using the Louvain algorithm, as in (Zhang et al., 2021), and the K-means algorithm (Lei et al., 2023).

To provide upper and lower benchmarks, all experiments are also conducted for a centralized setting utilizing the global graph and a localized setting using only the local subgraphs. In App. F we also give results for an MLP approach to highlight the importance of utilizing the spatial structure within the data. For the GNN, we rely on GRAPHSAGE (Hamilton et al., 2017) with two or three layers, depending on the dataset (decoupled GCN performs similarly as shown in Table 6 in App. E). We further compare FEDSTRUCT to vanilla FL (FEDSGD GNN), FEDPUB, and FEDSAGE+ (Zhang et al., 2021), which do not exploit cross-clients interconnections, and FEDGCN (Yao et al., 2024) as an SFL method that exploits interconnection. For FEDCGN, it is important to note that the server must access the global adjacency matrix to aggregate encrypted node features (shared by the clients) and forward the result. Moreover, the homomorphic encryption only protects against the server, not against the clients, which have access to aggregated node features of other clients, constituting a breach of privacy as recently shown in (Ngo et al., 2024).

For FEDSTRUCT, we consider three methods to create NSFs: one-hot degree vector (DEG), FEDSTAR (FED⋆), and the task-dependent approach proposed in Sec. 5.1 HOP2VEC (H2V), see App. A for details. For H2V, we also provide the performance of the non-pruned version (FEDSTRUCT (H2V)-F), for comparison. Throughout, we set the filter parameters (see Sec. 5.2) $\beta_{L_\mathsf{s}} = 1$ and $\beta_l = 0$ otherwise, effectively filtering out links between dissimilar nodes. While these parameters could be optimized for specific graphs, we chose this setup to demonstrate the robustness of FEDSTRUCT.

**Overall performance.** In Table 2, we report the node classification accuracy, after model convergence, on the different datasets for a random partitioning over 10 independent runs. The accuracy difference between central GNN and local GNN provides insights into the potential gains by utilizing FL. For example, on the Cora and Amazon Ratings datasets with 10 clients, the gap is $43.78\%$ and $9.05\%$, respectively. For these datasets, FEDSGD GNN closes the gap to $16.94\%$ and $5.24\%$, respectively. FEDSAGE+ yields similar performance as FEDSGD GNN for all datasets. FEDPUB also exhibits similar performance due to the low number of labeled nodes and random partitioning, as detecting

**Table 2:** Node classification accuracy with random partitioning. Nodes are split into train-val-test as 10%-10%-80%. For each result, the mean and standard deviation are shown for 10 independent runs. The top performance is highlighted in black bold, and the second-best in blue bold. Edge homophily ratio ($h$) is given in brackets.

| | CORA ($h = 0.81$) | | | CITESEER ($h = 0.74$) | | | PUBMED ($h = 0.80$) | | |
|---|---|---|---|---|---|---|---|---|---|
| CENTRAL GNN | 82.94±1.26 | | | 69.37±1.07 | | | 85.12±1.15 | | |
| | 5 CLIENTS | 10 CLIENTS | 20 CLIENTS | 5 CLIENTS | 10 CLIENTS | 20 CLIENTS | 5 CLIENTS | 10 CLIENTS | 20 CLIENTS |
| FEDSGD GNN | 67.55±1.05 | 66.00±1.51 | 64.47±1.26 | 64.59±0.80 | 63.38±0.76 | 63.91±1.09 | 84.74±0.36 | 84.66±0.22 | 84.55±0.52 |
| FEDSAGE+ | 68.03±0.87 | 66.33±1.69 | 64.64±1.72 | 64.38±0.91 | 63.93±0.97 | 63.63±1.24 | 84.61±0.35 | 84.64±0.37 | 84.45±0.37 |
| FEDPUB | 67.59±1.16 | 61.82±1.84 | 48.47±2.66 | 64.50±0.95 | 62.91±0.76 | 52.08±3.78 | 82.16±0.49 | 82.39±0.41 | 82.30±0.44 |
| FEDGCN-2HOP [2] | **82.21±0.95** | **82.90±0.95** | **82.39±1.26** | **70.20±1.13** | **70.49±1.03** | **70.42±0.81** | **85.46±0.69** | **85.73±0.77** | **85.90±0.80** |
| FEDSTRUCT (DEG) | 72.24±0.81 | 69.89±1.85 | 66.01±1.82 | 65.35±1.05 | 63.54±0.64 | 62.34±1.00 | 84.03±0.42 | 84.01±0.38 | 83.64±0.36 |
| FEDSTRUCT (FED⋆) | 72.25±0.90 | 69.61±1.87 | 66.13±1.54 | 65.54±1.07 | 63.38±0.90 | 61.97±0.89 | 83.98±0.38 | 84.02±0.35 | 83.69±0.47 |
| FEDSTRUCT (H2V) | 79.34±0.85 | 79.27±0.90 | 78.47±1.26 | **66.20±0.83** | 65.43±0.98 | 64.33±0.92 | 84.67±0.34 | 85.02±0.43 | 85.24±0.40 |
| FEDSTRUCT (H2V)-F | **79.99±1.04** | **79.88±0.92** | **78.80±1.48** | 66.15±1.09 | **65.93±0.81** | **64.96±0.79** | **85.24±0.44** | **85.79±0.60** | **86.09±0.42** |
| LOCAL GNN | 49.36±1.56 | 39.24±1.64 | 31.02±1.63 | 49.00±1.52 | 39.88±1.62 | 32.72±1.12 | 79.00±0.67 | 75.27±0.59 | 70.74±0.43 |

| | CHAMELEON ($h = 0.23$) | | | AMAZON PHOTO ($h = 0.82$) | | | AMAZON RATINGS ($h = 0.38$) | | |
|---|---|---|---|---|---|---|---|---|---|
| CENTRAL GNN | 54.38±1.96 | | | 94.10±0.30 | | | 41.42±0.80 | | |
| | 5 CLIENTS | 10 CLIENTS | 20 CLIENTS | 5 CLIENTS | 10 CLIENTS | 20 CLIENTS | 5 CLIENTS | 10 CLIENTS | 20 CLIENTS |
| FEDSGD GNN | 40.23±1.89 | 36.80±1.70 | 34.62±1.47 | 92.17±0.37 | **91.55±0.34** | 90.72±0.37 | 37.06±0.69 | 35.96±0.46 | 36.31±0.54 |
| FEDSAGE+ | 40.05±2.11 | 36.32±1.59 | 34.68±1.69 | **92.18±0.33** | 91.33±0.47 | 90.65±0.59 | 37.01±0.75 | 35.85±0.39 | 36.17±0.53 |
| FEDPUB | 40.15±1.44 | 33.31±1.37 | 28.34±1.13 | 90.19±0.38 | 88.05±0.68 | 85.53±0.74 | 36.55±0.63 | 35.72±0.60 | 36.18±0.93 |
| FEDGCN-2HOP [2] | 50.35±1.92 | 48.77±1.73 | 49.35±2.18 | **93.76±0.23** | **93.72±0.41** | **93.72±0.54** | 40.87±0.37 | 40.78±0.56 | 40.35±0.59 |
| FEDSTRUCT (DEG) | 44.33±1.56 | 41.82±1.78 | 40.13±1.85 | 90.73±0.32 | 89.72±0.43 | 89.42±0.48 | 39.35±0.47 | 38.67±0.66 | 38.31±0.54 |
| FEDSTRUCT (FED⋆) | 44.71±1.44 | 41.89±1.67 | 40.06±2.02 | 90.75±0.39 | 89.78±0.38 | 89.14±0.38 | 39.31±0.51 | 38.65±0.44 | 38.36±0.58 |
| FEDSTRUCT (H2V) | **53.33±2.25** | **52.60±1.25** | **51.85±1.26** | 91.14±0.46 | 90.93±0.27 | 91.25±0.38 | **41.14±0.43** | **40.97±0.64** | **40.55±0.25** |
| FEDSTRUCT (H2V)-F | **53.20±1.97** | **53.09±1.85** | **51.89±1.09** | 90.57±0.54 | 90.74±0.29 | **91.29±0.58** | **41.13±0.37** | **41.07±0.56** | **40.83±0.36** |
| LOCAL GNN | 35.63±1.79 | 29.60±1.25 | 23.70±0.66 | 87.91±0.65 | 77.12±1.75 | 60.44±0.98 | 34.32±0.46 | 32.80±0.43 | 31.68±0.52 |

[2] **FEDGCN lacks privacy** as the server must have access to the global adjacency matrix and aggregated node features and 2-hop structures are shared between clients, which also constitutes a privacy breach as shown in (Ngo et al., 2024). Moreover, the official implementation overlooks isolated external neighbors removal, potentially enhancing prediction performance above its actual capabilities.

communities in this setup cannot be done efficiently. Moreover, FEDGCN is included as a method that considers the inter-connections between nodes. This method, relying on server access to the global adjacency matrix (see App. C.4), shares 2-hop aggregated node features with other clients and, therefore, it is not private.

From Table 2, it can be seen that FEDSTRUCT further improves performance compared to FEDSGD GNN and FEDSAGE+. The improvement is significant for the Cora and Chameleon datasets. For Chameleon and 20 clients, FEDSTRUCT with HOP2VEC achieves an accuracy of 52.76% compared to 34.33% for FEDSAGE+. The table also shows that the performance improves with NSFs able to collect more global information as can be seen by the improvements between DEG, FED⋆, and H2V. Notably, FEDSTRUCT achieves performance close to that of the non-private framework FEDGCN and the centralized approach. Also, notice that for Chameleon and Amazon-ratings, two heterophilic graphs, FEDSTRUCT outperforms FEDGCN.

Compared to, e.g., FEDSGD GNN, FEDSTRUCT remains robust to an increasing number of clients and, as shown in App. F, across different partitioning methods, highlighting its ability to exploit inter-client connections. Furthermore, FEDSTRUCT (H2V) exhibits only a slight reduction in performance compared to FEDSTRUCT (H2V)-F, while significantly reducing communication complexity (see Sec. 5.6). Additional results, including federated averaging and central decoupled GCN, are provided in App. F for various partitionings, varying number of labeled training nodes, and degrees of heterophily.

## 6.1 IMPACT OF THE NUMBER OF LABELED TRAINING NODES

As noted by Liu et al. (2020), decoupled GCN addresses over-smoothing. Dong et al. (2021) further showed its suitability for semi-supervised learning by leveraging pseudo-labels from unlabeled nodes. Figure 4 (a), shows FEDSTRUCT's performance on Cora for varying NSF generators and labeled node ratios, using a $x$%-10%-$(90 - x)$% train-val-test split, where $x$ is the percentage of labeled nodes.

FEDSTRUCT with H2V comes close to CENTRAL GNN across all labeled node fractions, performing well even with just 5% labeled nodes, achieving a top-1 accuracy of 76%. In contrast, FEDSAGE+ and FEDPUB show a sharp decline, dropping from a top-1 accuracy of 75% with 50% labeled nodes

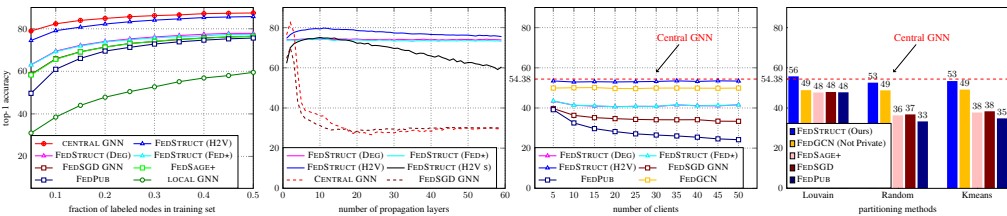

**Figure 4:** (a) Accuracy vs training-ratio for CORA with random partitioning and 10 clients; (b) Accuracy vs number of propagation layers on CORA with K-means partitioning and 5 clients; (c) Accuracy vs number of clients on CHAMELEON with random partitioning; (d) Accuracy on CHAMELEON with 10 clients for various partitioning methods.

to a top-1 accuracy of 60% and 50%, respectively, with 5% labeled nodes. Finally, it can be seen that LOCAL GNN deteriorates with fewer labels.

## 6.2 IMPACT OF THE NUMBER OF LAYERS IN THE DECOUPLED GCN

We evaluate the impact of the number of GNN/decoupled GCN layers on the Cora dataset. We compare FEDSTRUCT with different NSEs to CENTRAL GNN and FEDSGD GNN (using GRAPHSAGE). We also assess FEDSTRUCT's performance when only NSEs are used during inference (H2V S). We consider a setup with K-means partitioning and 10 clients.

Figure 4 (b) shows that GNNs suffer from oversmoothing after just a few layers, limiting the ability to use distant nodes. In contrast, FEDSTRUCT's performance remains stable across a varying number of layers, suggesting effective use of distant nodes. FEDSTRUCT (H2V S) performs poorly with fewer than 10 or more than 30 layers, peaking between 10-20 layers. This is due to the NSEs being unable to capture extended neighborhoods with few layers, and over-smoothing with many layers. H2V S achieves 75% accuracy, while H2V reaches 80%, indicating that while NSEs provide useful information, NFEs are necessary to achieve performance close to CENTRAL GNN.

## 6.3 IMPACT OF THE NUMBER CLIENTS AND PARTITIONING METHOD

Next, we evaluate the impact of the number of clients on the performance of FEDSTRUCT with different NSEs, alongside FEDSGD GNN, FEDPUB, and FEDGCN, using the Chameleon dataset with random partitioning. As shown in Figure 4 (c), FEDSTRUCT and FEDGCN show stable performance as the number of clients increases. FEDSTRUCT's ability to leverage deeper structures allows it to perform nearly as well as CENTRAL GNN. In contrast, FEDPUB and FEDSGD degrade with more clients, with FEDPUB performing the worst due to the increased difficulty in predicting communities as the number of clients grows.

In Figure 4 (d), we present the performance on Chameleon with 10 clients under different partitioning methods. FEDSTRUCT consistently achieves the best performance across all scenarios, even surpassing CENTRAL GNN with the less challenging Louvain partitioning. Both FEDSTRUCT and FEDGCN (which does not provide privacy) exhibit robustness across different partitioning methods, while the other frameworks are more sensitive to the choice of partitioning.

## 7 CONCLUDING REMARKS

We introduced FEDSTRUCT, a framework for SFL that decouples node and structure features, explicitly exploiting structural dependencies. FEDSTRUCT effectively addresses the privacy-communication-utility trilemma as follows:

**Privacy**: Unlike other SFL frameworks, which require sharing original or generated node features and/or embeddings among clients, FEDSTRUCT eliminates this need and minimizes sensitive information sharing. Privacy considerations are detailed in App. G.1.

**Utility**: FEDSTRUCT performs close to a centralized approach, excelling in semi-supervised learning with few labeled nodes, and significantly outperforms earlier methods like FEDSAGE+ and FEDPUB in challenging scenarios. It is also robust across different partitionings, client numbers, and heterophily.

**Communications**: FEDSTRUCT's communication overhead scales linearly with the number of nodes, comparable to benchmarks like FEDSAGE+.

ACKNOWLEDGMENTS

This work was partially supported by the Swedish Research Council (VR) under grants 2020-03687 and 2023-05065, by the Wallenberg AI, Autonomous Systems and Software Program (WASP) funded by the Knut and Alice Wallenberg Foundation, and by the Swedish innovation Agency (Vinnova) under grant 2022-03063.

The computations were enabled by resources provided by the National Academic Infrastructure for Supercomputing in Sweden (NAISS), partially funded by the Swedish Research Council through grant agreement no. 2022-06725.

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

**Table 3:** Comparing NSF generator algorithms

| DATA | DEG | GDV | NODE2VEC | FED⋆ | HOP2VEC |
|------|-----|-----|----------|------|---------|
| TASK DEPENDENT | ✗ | ✗ | ✗ | ✗ | ✓ |
| GLOBAL PROPERTIES | ✗ | ✗ | ✓ | ✓ | ✓ |
| FAST | ✓ | ✗ | ✗ | ✗ | ✓ |
| NO PARAMETER TUNING | ✓ | ✓ | ✗ | ✗ | ✓ |
| LOCALLY COMPUTABLE | ✓ | ✗ | ✗ | ✓ | ✓ |

## A    NODE STRUCTURE FEATURE GENERATION

In this appendix, we provide some background on the creation of NSFs. The simplest method considered in this paper is the one-hot degree encoding (DEG), which simply creates a vector with a single non-zero entry whose position indicates the node degree. Naturally, the one-hot encoding is unable to capture anything beyond 1-hop neighbors.

Graphlet degree vectors (GDVs) go beyond the 1-hop neighborhood by representing the local structure of a node within the node structure feature (NSF). This is achieved by, for each node, counting the number of occurrences within a pre-defined set of graphlets (Pržulj, 2007). Although GDV is more expressive than DEG, this approach suffers from high complexity as the number of graphlets grows very fast with the number of nodes.

NODE2VEC is an unsupervised node embedding algorithm that creates NSFs from random walks in the graph by capturing both local and global properties (Grover and Leskovec, 2016). It uses controlled random walks to explore the graph, generating node sequences similar to sentences in a language. For each node in the graph, multiple random walks are performed creating a large corpus of node sequences. These sequences are then used to train a skip-gram model to predict the context of a given node, i.e., the neighboring nodes in the random walks. Once trained, the NSFs are obtained by querying the skip-gram model with a node and extracting its latent representation.

FED⋆ is proposed in (Tan et al., 2023) to capture structural similarities in the context of federated graph classification. In particular, for each node in a local graph-structured data record, clients create an NSF by concatenating NSFs created from DEG and from a random walk. We adapt this methodology to our setting by extracting the diagonal element corresponding to a given node from the powers of $\hat{A}$, during the process of computing $\bar{A}$, in place of (Tan et al., 2023, Eq. 6).

As shown in Appendix H, NODE2VEC yields superior performance to DEG, GDV, and FED⋆ due to its ability to capture global properties of the graph. Note, however, that NODE2VEC and GDV require access to the adjacency matrix of the global graph which is not generally available for our setting. Moreover, the objective function of NODE2VEC contains multiple hyperparameters and is task agnostic, i.e., its representations do not account for the node classification task at hand.

The proposed HOP2VEC leverages the advantages of NODE2VEC while also alleviating its shortcomings within our setting. In HOP2VEC, instead of learning the NSFs before FEDSTRUCT initiates, we start from a random NSF and train a generator within FEDSTRUCT. As a result, we obtain NSFs that are tailored toward the task, e.g., to minimize miss-classification. Notably, in sharp contrast to NODE2VEC, HOP2VEC does not require knowledge of the complete adjacency matrix. Moreover, compared to NODE2VEC, HOP2VEC is faster and does not need any hyperparameter tuning. Finally, when paired with DECOUPLED GCN, it is able to capture the global properties of the graph. A summary of the different NSF-generating methods is given in Table 3.

The NSFs obtained by HOP2VEC are task-dependent, capture deep structure dependencies, and do not rely on homophily. The first property is inherent to the training—the NSFs are formed to minimize misclassification. The second property results from the fact that the NSFs are optimized based on $\bar{A}_{uv}$ through Eq. 10—since coefficient $\bar{A}_{uv}$ encompasses the $L$-hop neighbors of node $v$, and $L$ can be large, HOP2VEC is able to capture the global graph's structure information. Furthermore, the parameters $\beta_l$ of the trainable coefficients $\bar{A}_{uv}$ (see Sec. 3) can be adjusted (or learned) to account for heterophilic graphs (we discuss heterophily in App. G.2).

## B  TRAINING PROCESS OF FEDSTRUCT

The training process of FEDSTRUCT with generic NSEs is described in Algorithm 1. The training process of FEDSTRUCT using HOP2VEC is described in Algorithm 2.

---

**Algorithm 1** FEDSTRUCT

---

**input** K client with their respective subgraph $\{\mathcal{G}_i\}_{i=1}^K$ FEDSTRUCT model parameters $\boldsymbol{\theta} = (\boldsymbol{\theta}_f || \boldsymbol{\theta}_s)$
  **for** i=1 to K **do**
    Collaboratively obtain $\bar{\boldsymbol{A}}^{[i]}$ based on Algorithm 3
    Locally compute NSFs $\mathcal{S}_i = \{\boldsymbol{s}_u, \forall u \in \mathcal{V}_i\}$
    Share $\mathcal{S}_i$ with all the clients
  **end for**
  **for** e=1 to Epochs **do**
    **for** i=1 to K **do**
      Client $i$ collects $\boldsymbol{\theta}$ from the server
      **for** $v \in \tilde{\mathcal{V}}_i$ **do**
        Calculate $\hat{\boldsymbol{y}}_v$ based on Eq. 10
      **end for**
      Calculate $\mathcal{L}_i(\boldsymbol{\theta})$ based on Eq. 4
      Calculate $\nabla_{\boldsymbol{\theta}} \mathcal{L}_i(\boldsymbol{\theta})$ based on Proposition 3
      Send $\nabla_{\boldsymbol{\theta}} \mathcal{L}_i(\boldsymbol{\theta})$ to the server
    **end for**
    Calculate $\nabla_{\boldsymbol{\theta}} \mathcal{L}(\boldsymbol{\theta})$ based on Eq. 5
    $\boldsymbol{\theta} \leftarrow \boldsymbol{\theta} - \lambda \nabla_{\boldsymbol{\theta}} \mathcal{L}(\boldsymbol{\theta})$
  **end for**

---

**Algorithm 2** FEDSTRUCT using HOP2VEC

---

**input** K client with their respective subgraph $\{\mathcal{G}_i\}_{i=1}^K$ FEDSTRUCT model parameters $\boldsymbol{\theta} = (\boldsymbol{\theta}_f || \boldsymbol{\theta}_s)$
  Server initialize $\nabla_{\boldsymbol{S}} \mathcal{L}(\boldsymbol{\theta}, \boldsymbol{S}) = \boldsymbol{0}$
  **for** i=1 to K **do**
    Collaboratively obtain $\bar{\boldsymbol{A}}^{[i]}$ based on Algorithm 3
    Create $\mathcal{S}_i = \{\boldsymbol{s}_u, \forall u \in \mathcal{V}_i\}$ by randomly initialize $\boldsymbol{s}_u \quad \forall u \in \mathcal{V}_i$
    Share $\mathcal{S}_i$ with all the clients
    Collect $\mathcal{S}_j \quad \forall j \in [K]$ and create $\boldsymbol{S} = ||(\boldsymbol{s}_v^{\mathsf{T}}, \quad \forall v \in \mathcal{V})$
  **end for**
  **for** e=1 to Epochs **do**
    **for** i=1 to K **do**
      Client $i$ collects $\boldsymbol{\theta}$ from the server
      Client $i$ collects $\nabla_{\boldsymbol{S}} \mathcal{L}(\boldsymbol{\theta}, \boldsymbol{S})$ from the server
      Client $i$ updates $\boldsymbol{S} \leftarrow \boldsymbol{S} - \lambda_s \nabla_{\boldsymbol{S}} \mathcal{L}(\boldsymbol{\theta}, \boldsymbol{S})$
      **for** $v \in \tilde{\mathcal{V}}_i$ **do**
        Calculate $\hat{\boldsymbol{y}}_v$ based on Eq. 15
      **end for**
      Calculate $\mathcal{L}_i(\boldsymbol{\theta}, \boldsymbol{S})$ based on Eq. 4
      Calculate $\nabla_{\boldsymbol{\theta}} \mathcal{L}_i(\boldsymbol{\theta}, \boldsymbol{S})$ based on Proposition 3
      Calculate $\nabla_{\boldsymbol{S}} \mathcal{L}_i(\boldsymbol{\theta}, \boldsymbol{S})$ based on Proposition 2
      Send $\nabla_{\boldsymbol{\theta}} \mathcal{L}_i(\boldsymbol{\theta}, \boldsymbol{S})$ and $\nabla_{\boldsymbol{S}} \mathcal{L}_i(\boldsymbol{\theta}, \boldsymbol{S})$ to the server
    **end for**
    Calculate $\nabla_{\boldsymbol{\theta}} \mathcal{L}(\boldsymbol{\theta}, \boldsymbol{S})$ based on Eq. 5
    $\boldsymbol{\theta} \leftarrow \boldsymbol{\theta} - \lambda \nabla_{\boldsymbol{\theta}} \mathcal{L}(\boldsymbol{\theta}, \boldsymbol{S})$
    Calculate $\nabla_{\boldsymbol{S}} \mathcal{L}(\boldsymbol{\theta}, \boldsymbol{S}) = \frac{1}{|\tilde{\mathcal{V}}|} \sum_{i=1}^K \nabla_{\boldsymbol{S}} \mathcal{L}_i(\boldsymbol{\theta}, \boldsymbol{S})$
  **end for**

---

## C  PROOF SECTION

### C.1  LOCAL GRADIENT

To prove Proposition 1, we need the following Lemma and proposition.

**Lemma 1.** *Let $\hat{\boldsymbol{y}} = \mathrm{softmax}(\boldsymbol{q})$ and $\mathcal{L} = \mathrm{CE}(\boldsymbol{y}, \hat{\boldsymbol{y}})$, where* $\mathrm{CE}$ *is the cross-entropy loss function and $\boldsymbol{y}$ is the label corresponding to the input vector $\boldsymbol{q}$, with $\sum_{i=1}^{c} y_i = 1$ and $c$ being the number of classes. The gradient of $\mathcal{L}$ with respect to $\boldsymbol{z}$ is equal to*

$$\nabla_{\boldsymbol{q}} \mathcal{L} = \hat{\boldsymbol{y}} - \boldsymbol{y}. \tag{17}$$

*Proof.* Using the definition of cross-entropy

$$\mathrm{CE}(\boldsymbol{y}, \hat{\boldsymbol{y}}) = -\sum_{i=1}^{c} y_i \log(\hat{y}_i), \tag{18}$$

and the definition of softmax,

$$\mathrm{softmax}(\boldsymbol{q}) = \|\left( \frac{e^{q_i}}{\sum_{j=1}^{c} e^{q_j}} \quad \forall i \in [c] \right), \tag{19}$$

we can rewrite $\mathcal{L}$ as

$$\begin{aligned} \mathcal{L} &= \sum_{i=1}^{c} y_i \big( \mathrm{LSE}(\boldsymbol{q}) - q_i \big) \\ &= \mathrm{LSE}(\boldsymbol{q}) \sum_{i=1}^{c} y_i - \sum_{i=1}^{c} y_i z_i \\ &= \mathrm{LSE}(\boldsymbol{q}) - \boldsymbol{q}^{\mathsf{T}} \boldsymbol{y}, \end{aligned} \tag{20}$$

where $\mathrm{LSE}(\boldsymbol{q}) = \log(\sum_{j=1}^{c} e^{q_j})$ is the log-sum-exp function. The partial derivative of $\mathrm{LSE}(\boldsymbol{q})$ with respect to $\boldsymbol{q}$ is the $\mathrm{softmax}$ function

$$\begin{aligned} \nabla_{\boldsymbol{q}} \mathrm{LSE}(\boldsymbol{z}) &= \|(\frac{\partial \mathrm{LSE}(\boldsymbol{q})}{\partial q_k} \quad \forall k \in [c]) \\ &= \|(\frac{e^{q_k}}{\sum_{j=1}^{c} e^{q_j}} \quad \forall k \in [c]) \\ &= \mathrm{softmax}(\boldsymbol{q}). \end{aligned} \tag{21}$$

Therefore, using Eq. 21 and taking the derivative of Eq. 20 with respect to $\boldsymbol{z}$ leads to

$$\nabla_{\boldsymbol{q}} \mathcal{L} = \nabla_{\boldsymbol{q}} \mathrm{LSE}(\boldsymbol{q}) - \nabla_{\boldsymbol{q}}(\boldsymbol{q}^{\mathsf{T}} \boldsymbol{y}) = \hat{\boldsymbol{y}} - \boldsymbol{y}.$$

$\square$

The local gradient $\nabla_{\boldsymbol{\theta}} \mathcal{L}_i(\boldsymbol{\theta})$ is given by the following proposition.

**Proposition 3.** *Let $\mathcal{L}_i(\boldsymbol{\theta})$ in Eq. 4 use the cross-entropy loss with $\boldsymbol{\theta} = (\boldsymbol{\theta}_{\mathsf{f}} \| \boldsymbol{\theta}_{\mathsf{s}})$ and let $\hat{\boldsymbol{y}}_v$ be as in Eq. 10. The local gradient*

$$\nabla_{\boldsymbol{\theta}} \mathcal{L}_i(\boldsymbol{\theta}) = \|\left( \frac{\partial \mathcal{L}_i(\boldsymbol{\theta})}{\partial \theta_j}, \quad \forall j \in [|\boldsymbol{\theta}|] \right)$$

*is given by*

$$\frac{\partial \mathcal{L}_i(\boldsymbol{\theta})}{\partial \theta_{\mathsf{s},j}} = \sum_{v \in \tilde{\mathcal{V}}_i, u \in \mathcal{V}} \bar{A}_{vu} \frac{\partial \boldsymbol{g}_{\boldsymbol{\theta}_{\mathsf{s}}}(\boldsymbol{s}_u)}{\partial \theta_{\mathsf{s},j}} (\hat{\boldsymbol{y}}_v - \boldsymbol{y}_v), \qquad \frac{\partial \mathcal{L}_i(\boldsymbol{\theta})}{\partial \theta_{\mathsf{f},j}} = \sum_{v \in \tilde{\mathcal{V}}_i} \frac{\partial f_{\boldsymbol{\theta}_{\mathsf{f}}}(v)}{\partial \theta_{\mathsf{f},j}} (\hat{\boldsymbol{y}}_v - \boldsymbol{y}_v). \tag{22}$$

*Proof.* By the chain rule and Lemma 1, we have

$$
\begin{aligned}
\frac{\partial \mathcal{L}_i(\boldsymbol{\theta})}{\partial \theta_j} &= \sum_{v \in \tilde{\mathcal{V}}_i} \frac{\partial \boldsymbol{q}_v}{\partial \theta_j} \nabla_{\underline{1}_v} \mathcal{L}_i(\boldsymbol{\theta}) \\
&= \sum_{v \in \tilde{\mathcal{V}}_i} \frac{\partial \boldsymbol{q}_v}{\partial \theta_j} (\hat{\boldsymbol{y}}_v - \boldsymbol{y}_v) \quad \forall j \in [|\boldsymbol{\theta}|] \,,
\end{aligned}
\tag{23}
$$

where $\boldsymbol{q}_v = \sum_{u \in \mathcal{V}} \bar{A}_{vu} \boldsymbol{g}_{\boldsymbol{\theta}_{\mathsf{s}}}(\boldsymbol{s}_u) + f_{\boldsymbol{\theta}_{\mathsf{f}}}(v)$. Taking the derivative of $\boldsymbol{q}_v$ with respect to the different entries of $\boldsymbol{\theta}$ leads to

$$
\frac{\partial \boldsymbol{q}_v}{\partial \theta_{\mathsf{s},j}} = \sum_{u \in \mathcal{V}} \bar{A}_{vu} \frac{\partial \boldsymbol{g}_{\boldsymbol{\theta}_{\mathsf{s}}}(\boldsymbol{s}_u)}{\partial \theta_{\mathsf{s},j}} \qquad\qquad \forall j \in [|\boldsymbol{\theta}_{\mathsf{s}}|] \tag{24}
$$

$$
\frac{\partial \boldsymbol{q}_v}{\partial \theta_{\mathsf{f},j}} = \frac{\partial f_{\boldsymbol{\theta}_{\mathsf{f}}}(v)}{\partial \theta_{\mathsf{f},j}} \qquad\qquad \forall j \in [|\boldsymbol{\theta}_{\mathsf{f}}|]. \tag{25}
$$

Substituting Eq. 24 and Eq. 25 into Eq. 23 and separating the summation over $\mathcal{V}$ over the different clients concludes the proof. $\qquad\square$

## C.2 PROOF OF PROPOSITION 1

1. To obtain $\{\hat{\boldsymbol{y}}_v, \forall v \in \tilde{\mathcal{V}}_i\}$ in Eq. 10, client $i$ only needs external inputs $\bar{\boldsymbol{A}}^{[i]}$ and $\boldsymbol{S}$.

2. To calculate $\nabla_{\boldsymbol{\theta}} \mathcal{L}_i(\boldsymbol{\theta})$, client $i$ only needs external inputs $\bar{\boldsymbol{A}}^{[i]}$ and $\boldsymbol{S}$.

By combining 1 and 2 we prove Proposition 1.

## C.3 PROOF OF PROPOSITION 2

First notice that

$$
\nabla_{\boldsymbol{S}} \mathcal{L}_i(\boldsymbol{\theta}, \boldsymbol{S}) = ||\left((\nabla_{\boldsymbol{s}_p} \mathcal{L}_i(\boldsymbol{\theta}, \boldsymbol{S}))^\mathsf{T} \quad \forall p \in \mathcal{V}\right). \tag{26}
$$

Using the chain rule and Lemma 1, we have

$$
\begin{aligned}
\nabla_{\boldsymbol{s}_p} \mathcal{L}_i(\boldsymbol{\theta}, \boldsymbol{S}) &= \sum_{v \in \tilde{\mathcal{V}}_i} \frac{\partial \boldsymbol{q}_v}{\partial \boldsymbol{s}_p} \nabla_{\boldsymbol{q}_v} \mathcal{L}_i(\boldsymbol{\theta}, \boldsymbol{S}) \\
&= \sum_{v \in \tilde{\mathcal{V}}_i} \frac{\partial \boldsymbol{q}_v}{\partial \boldsymbol{s}_p} (\hat{\boldsymbol{y}}_v - \boldsymbol{y}_v) \quad \forall p \in \mathcal{V} \,,
\end{aligned}
\tag{27}
$$

where $\boldsymbol{q}_v = \sum_{u \in \mathcal{V}} \bar{A}_{vu} \boldsymbol{s}_u + f_{\boldsymbol{\theta}_{\mathsf{f}}}(v)$. Taking the derivative of $\boldsymbol{q}_v$ with respect to $\boldsymbol{s}_p$ leads to

$$
\begin{aligned}
\frac{\partial \boldsymbol{q}_v}{\partial \boldsymbol{s}_p} &= \sum_{u \in \mathcal{V}} \bar{A}_{vu} \frac{\partial \boldsymbol{s}_u}{\partial \boldsymbol{s}_p} \\
&= \bar{A}_{vp} \frac{\partial \boldsymbol{s}_p}{\partial \boldsymbol{s}_p} \\
&= \bar{A}_{vp} \boldsymbol{I} \quad \forall p \in \mathcal{V}.
\end{aligned}
\tag{28}
$$

Substituting Eq. 28 into Eq. 27 leads to

$$
\nabla_{\boldsymbol{s}_p} \mathcal{L}_i(\boldsymbol{\theta}, \boldsymbol{S}) = \sum_{v \in \tilde{\mathcal{V}}_i} \bar{A}_{vp}(\hat{\boldsymbol{y}}_v - \boldsymbol{y}_v) \quad \forall p \in \mathcal{V}. \tag{29}
$$

Finally, substituting Eq. 29 into Eq. 26 concludes the proof.

### C.4 FEDGCN WITH 2 HOPS NEEDS GLOBAL ADJACENCY MATRIX KNOWLEDGE

In the FedGCN scheme, as detailed in Equation 3 on page 5 of the FedGCN paper, to calculate $\hat{y}_i$, the following information must be sent to the server by Client $z$:

$$\sum_{j \in \mathcal{N}_i} \mathbb{I}_z(c(j)) \boldsymbol{A}_{ij} \boldsymbol{x}_j \tag{30}$$

$$\boldsymbol{h}_{zj} = \sum_{m \in \mathcal{N}_j} \mathbb{I}_z(c(m)) \boldsymbol{A}_{jm} \boldsymbol{x}_m \quad \forall j \in \mathcal{N}_i \setminus i. \tag{31}$$

Once the server obtains $\boldsymbol{h}_{zj}$ for all clients $z \in [K]$, it calculates the aggregate $\boldsymbol{h}_j = \sum_{z \in [K]} \boldsymbol{h}_{zj}$ and forwards this result to the client where node $i$ resides. However, as $\boldsymbol{h}_j$ is a quantity pertaining to node $j$, the server must know that $i$ and $j$ are connected. Similar arguments hold for all other nodes. Hence, the server needs access to the global adjacency matrix for this scheme to work with 2 hops.

## D SCHEME TO OBTAIN THE LOCAL PARTITION OF THE $L$-HOP COMBINED ADJACENCY MATRIX

### D.1 ALGORITHM

As discussed in Section 5, FEDSTRUCT with decoupled GCN only requires the clients to have access to their local partition of the global $L$-hop combined adjacency matrix, i.e., $\bar{\boldsymbol{A}}^{[i]}$. In this appendix, we present an algorithm that enables clients to acquire this matrix in a privacy-preserving manner. The scheme for obtaining the local partitions of the $L$-hop combined adjacency matrix $\bar{\boldsymbol{A}}^{[i]}$ is designed specifically to ensure that no client has access to the entire $\ell$-hop ($\ell \in [1, \ldots, L]$) adjacency matrix $\hat{\boldsymbol{A}}^\ell$.

We denote by $\tilde{\boldsymbol{A}}^{[i]} = ||(\tilde{\boldsymbol{a}}_v^\mathsf{T}, \quad \forall v \in \mathcal{V}_i)$ and $\hat{\boldsymbol{A}}^{[i]} = ||(\hat{\boldsymbol{a}}_v^\mathsf{T}, \quad \forall v \in \mathcal{V}_i)$ the local partition of $\tilde{\boldsymbol{A}}$ and $\hat{\boldsymbol{A}}$ pertaining to the nodes of client $i$, respectively. Further, note that $\hat{\boldsymbol{A}}^{[i]} = \left(\tilde{\boldsymbol{D}}^{[i]}\right)^{-1} \tilde{\boldsymbol{A}}^{[i]}$, where $\tilde{\boldsymbol{D}}^{[i]} \in \mathbb{R}^{|\mathcal{V}_i| \times |\mathcal{V}_i|}$ is the diagonal matrix of node degrees, with $\tilde{\boldsymbol{D}}^{[i]}(v, v) = \sum_{u \in \mathcal{V}} \tilde{a}_{vu}, \forall v \in \mathcal{V}_i$. Also, let $\tilde{\boldsymbol{A}}_j^{[i]} \in \mathbb{R}^{|\mathcal{V}_i| \times |\mathcal{V}_j|}$ denote the submatrix of $\tilde{\boldsymbol{A}}^{[i]}$ that connects client $i$ to client $j$ for $j \in [K]$ and define $\hat{\boldsymbol{A}}_j^{[i]}$ analogously. Consequently, we have $\hat{\boldsymbol{A}}_j^{[i]} = \left(\tilde{\boldsymbol{D}}^{[i]}\right)^{-1} \tilde{\boldsymbol{A}}_j^{[i]}$.

Each client $i$ has access to:

- $\tilde{\boldsymbol{A}}_j^{[i]} \in \mathbb{R}^{|\mathcal{V}_i| \times |\mathcal{V}_j|}$ for all $j \in [K]$ : representing the outgoing edges from client $i$ to client $j$ .

- $\tilde{\boldsymbol{A}}_i^{[j]} \in \mathbb{R}^{|\mathcal{V}_j| \times |\mathcal{V}_i|}$ for all $j \in [K]$ : representing the incoming edges from client $j$ to client $i$ .

We note that client $i$ can compute $\hat{\boldsymbol{A}}_j^{[i]}$ locally as it knows the destination of its outgoing edges, i.e., $\tilde{\boldsymbol{A}}_j^{[i]}$ for all $j \in [K]$, and can compute $\tilde{\boldsymbol{D}}^{[i]}$.

In client $i$, we are interested in acquiring its local partition of $\bar{\boldsymbol{A}}$, i.e., $\bar{\boldsymbol{A}}^{[i]} = ||(\bar{\boldsymbol{a}}_v^\mathsf{T}, \quad \forall v \in \mathcal{V}_i)$. Let $\bar{\boldsymbol{A}}_j^{[i]} \in \mathbb{R}^{|\mathcal{V}_i| \times |\mathcal{V}_j|}$ denote the submatrix of $\bar{\boldsymbol{A}}^{[i]}$ that connects client $i$ to client $j$ for $j \in [K]$. Based on Eq. 2 we can write $\bar{\boldsymbol{A}}_j^{[i]}$ as

$$\bar{\boldsymbol{A}}_j^{[i]} = \sum_{l=1}^{L_\mathrm{s}} \beta_\ell [\hat{\boldsymbol{A}}_j^{[i]}]^\ell, \quad j \in [K], \tag{32}$$

where $[\hat{\boldsymbol{A}}_j^{[i]}]^\ell$ is the submatrix of $\hat{\boldsymbol{A}}^\ell$ that connects client $i$ to client $j$ for $j \in [K]$. Since $\hat{\boldsymbol{A}}^\ell = \hat{\boldsymbol{A}}\hat{\boldsymbol{A}}^{\ell-1}$ we can easily show that

$$[\hat{\boldsymbol{A}}_j^{[i]}]^\ell = \sum_{k \in [K]} \hat{\boldsymbol{A}}_k^{[i]} [\hat{\boldsymbol{A}}_j^{[k]}]^{\ell-1} \tag{33}$$

$$= (\tilde{\boldsymbol{D}}^{[i]})^{-1} \sum_{k \in [K]} \underbrace{\tilde{\boldsymbol{A}}_k^{[i]} [\hat{\boldsymbol{A}}_j^{[k]}]^{\ell-1}}_{\text{computed at client } k}. \tag{34}$$

By induction, we can demonstrate that if each client $i$ has access to $[\hat{A}_j^{[i]}]^{\ell-1}$ for all $j \in [K]$ , they can then collaborate with each other to compute $[\hat{A}_j^{[i]}]^\ell$. Here's a step-by-step breakdown:

1. Base case: For $\ell = 1$ , we have $[\hat{A}_j^{[i]}]^1 = \hat{A}_j^{[i]}$. This information is available to Client $i$ from the start for all $j \in [K]$ .

2. Inductive step: Suppose Client $i$ knows $[\hat{A}_j^{[i]}]^{\ell-1}$ for all $j \in [K]$ . Then, Client $k$ can compute the intermediate term

$$B_{ijk} = \tilde{A}_k^{[i]}[\hat{A}_j^{[k]}]^{\ell-1} \tag{35}$$

for all $j \in [K]$ (since it knows $[\hat{A}_j^{[k]}]^{\ell-1}$ and $\tilde{A}_k^{[i]}$ for all $i, j \in [K]$) and share it with Client $i$. Notice that Client $i$ cannot reconstruct $[\hat{A}_j^{[k]}]^{\ell-1}$ from $B_{ijk}$ due to the low rank nature of $\tilde{A}_k^{[i]}$. Additionally, Client $k$ also prunes $B_{ijk}$ to remove any concern. Based on Eq. 34, Client $i$ can compute $[\hat{A}_j^{[i]}]^\ell$ by:

$$[\hat{A}_j^{[i]}]^\ell = (\tilde{D}^{[i]})^{-1} \sum_{k \in [K]} B_{ijk} . \tag{36}$$

To apply pruning, in each iteration, client $k$ sends only the top $\lceil \frac{p}{K} \rceil \cdot n_i$ values of $B_{ijk}$ to Client $i$ where $p$ is the pruning parameter. Therefore, client $i$ receives an estimate of $[\hat{A}_j^{[i]}]^\ell$ as

$$[\hat{A}_j^{[i]}]^\ell \approx (\tilde{D}^{[i]})^{-1} \sum_{k \in [K]} \underbrace{\tilde{B}_{ijk}}_{\text{computed at client } k} \quad , \quad j \in [K] . \tag{37}$$

where $\tilde{B}_{ijk}$ is the pruned version of matrix $B_{ijk}$ with pruning parameter $\lceil \frac{p}{K} \rceil$. This summation allows Client $i$ to update its local adjacency matrix partition without requiring access to the entire global adjacency matrix. At each step $\ell$ , Client $i$ only accesses $[\hat{A}_j^{[i]}]^\ell$ for all $j \in [K]$ and does not learn the full global matrix $[\hat{A}]^\ell$. Moreover, if additional security is required, the summation over $B_{ijk}$ could be performed on a secure server using homomorphic encryption.

As the final step, Client $i$ can compute $\bar{A}_j^{[i]}$ sequentially based on Eq. 32. The resulting algorithm is shown in Algorithm 3. Finally, we notice that Algorithm 3 requires each client $k$ to evaluate $L_s \times n_k \times n^2$ matrix products and to communicate $L_s \times p \times n$ matrices. This communication complexity is linear with $n$ and it is performed only once before the training initiates.

## D.2 EXAMPLE

To illustrate our algorithm and its privacy mechanism, we provide an example to compute the 2-hop combined adjacency matrix $\bar{A}$, defined in Section 3.

As stated in Section 2, SFL can be categorized into two groups:

1. SFL with no knowledge of cross-subgraph interconnections.

2. SFL with knowledge of cross-subgraph interconnections.

This paper focuses on the second scenario, where the interconnections between clients (i.e., between subgraphs) are known to the involved clients. Specifically, client $i$ knows its incoming and outgoing edges to every other client $j$, meaning it knows $\tilde{A}_i^{[j]}$ and $\tilde{A}_j^{[i]}$ for all $j \in [K]$.

It is important to note that the knowledge of $\tilde{A}_i^{[j]}$ and $\tilde{A}_j^{[i]}$ does not imply that the entire global adjacency matrix is accessible to client $i$. Each client has access only to the rows and columns corresponding to its own nodes. As an example, consider a graph with 9 nodes, partitioned across 3

---

**Algorithm 3** Private acquisition of $\bar{A}^{[i]}$

---

**input** $\{\tilde{A}_j^{[i]}, \tilde{A}_i^{[j]}\}_{j=1}^K, \tilde{D}^{[i]}$, power $L$, and list of weights $\{\beta_\ell\}_{\ell=1}^L$
**output** local matrix $\{\bar{A}^{[i]}\}$
  **for** $i = 1$ to $K$ **do**
    **for** $j = 1$ to $K$ **do**
      Initialize $\bar{A}_j^{[i]} = \beta_1(\tilde{D}^{[i]})^{-1}\tilde{A}_j^{[i]}$
    **end for**
  **end for**
  **for** $\ell = 2$ to $L_s$ **do**
    **for** $i = 1$ to $K$ **do**
      **for** k=1 to K **do**
        Client $k$ calculate $B_{ijk} = \tilde{A}_k^{[i]}[\hat{A}_j^{[K]}]^{\ell-1} \quad \forall j \in [K]$
        Client $k$ sends the pruned matrix $\tilde{B}_{ijk}$ to client $i$ for $j \in [K]$
      **end for**
      Client $i$ collects $\tilde{B}_{ijk}$ from clients $k \in [K]$
      Client $i$ stores $[\hat{A}_j^{[i]}]^\ell = (\tilde{D}^{[i]})^{-1}\sum_{k=1}^K \tilde{B}_{ijk}$ for $j \in [K]$
      $\bar{A}_j^{[i]} \leftarrow \bar{A}_j^{[i]} + \beta_\ell[\hat{A}_j^{[i]}]^\ell$ for $j \in [K]$
      $\bar{A}^{[i]} = [\bar{A}_1^{[i]}, \bar{A}_2^{[i]}, \ldots, \bar{A}_K^{[i]}]$
    **end for**
  **end for**

---

clients, given by the following adjacency matrix:

$$\mathbf{A} = \begin{bmatrix} 0 & 1 & 1 & 0 & 0 & 0 & 0 & 0 & 0 \\ 1 & 0 & 1 & 0 & 0 & 0 & 0 & 0 & 0 \\ 1 & 1 & 0 & 0 & 0 & 1 & 0 & 0 & 0 \\ 0 & 0 & 0 & 0 & 1 & 1 & 0 & 0 & 0 \\ 0 & 0 & 0 & 1 & 0 & 1 & 0 & 0 & 0 \\ 0 & 0 & 1 & 1 & 1 & 0 & 0 & 0 & 1 \\ 0 & 0 & 0 & 0 & 0 & 0 & 0 & 1 & 1 \\ 0 & 0 & 0 & 0 & 0 & 0 & 1 & 0 & 1 \\ 0 & 0 & 0 & 0 & 0 & 1 & 1 & 1 & 0 \end{bmatrix},$$

where $A_{ij} = 1$ indicates a connection between nodes $i$ and $j$, and $A_{ij} = 0$ indicates no connection.

In this setting, Client 1 comprises nodes $\{1, 2, 3\}$, Client 2 comprises nodes $\{4, 5, 6\}$, and Client 3 comprises nodes $\{7, 8, 9\}$.

Below, we depict the entries of the adjacency matrix known to Client 2 using 1 (there is a connection), 0 (there is no connection) and '?' (unknown connection):

$$\text{Known entries of } \boldsymbol{A} \text{ for Client 2} = \begin{bmatrix} ? & ? & ? & 0 & 0 & 0 & ? & ? & ? \\ ? & ? & ? & 1 & 0 & 0 & ? & ? & ? \\ ? & ? & ? & 0 & 0 & 1 & ? & ? & ? \\ 0 & 1 & 0 & 0 & 1 & 1 & 0 & 0 & 1 \\ 0 & 0 & 0 & 1 & 0 & 1 & 0 & 0 & 0 \\ 0 & 0 & 1 & 1 & 1 & 0 & 0 & 0 & 0 \\ ? & ? & ? & 0 & 0 & 0 & ? & ? & ? \\ ? & ? & ? & 0 & 0 & 0 & ? & ? & ? \\ ? & ? & ? & 1 & 0 & 0 & ? & ? & ? \end{bmatrix} \tag{38}$$

Client 2 knows all its internal connections (e.g., node 6 is connected to node 5) as well as its external edges (e.g., node 4 is connected to node 2 in Client 1 and node 9 in Client 3. The incoming edges to Client 2 from Client 3 are given by

$$\tilde{A}_{i=2}^{[j=3]} = \begin{bmatrix} 0 & 0 & 0 \\ 0 & 0 & 0 \\ 1 & 0 & 0 \end{bmatrix}. \tag{39}$$

That is, there is only one connection between Client 2 and Client 3 (between node 4 and node 9). Thus, while Client 2 has access to its own local and inter-client edges, it remains unaware of the internal connections of other clients or the interconnections between any other clients $j \in \{1, 3\}$ and $k \in \{1, 3\}$.

Hence, Client $i$ knows a portion of the global adjacency matrix $\mathbf{A}$, i.e., $\tilde{\boldsymbol{A}}^{[i]}$, corresponding to the connections between its own nodes and to some nodes in other clients. In particular, Client 1 knows

$$\text{internal connections (including self loops): } \tilde{\boldsymbol{A}}_1^{[1]} = \begin{bmatrix} 1 & 1 & 1 \\ 1 & 1 & 1 \\ 1 & 1 & 1 \end{bmatrix},$$

$$\text{outgoing connections: } \tilde{\boldsymbol{A}}_2^{[1]} = \begin{bmatrix} 0 & 0 & 0 \\ 0 & 0 & 0 \\ 0 & 0 & 1 \end{bmatrix} \quad \tilde{\boldsymbol{A}}_3^{[1]} = \begin{bmatrix} 0 & 0 & 0 \\ 0 & 0 & 0 \\ 0 & 0 & 0 \end{bmatrix},$$

$$\text{incoming connections: } \tilde{\boldsymbol{A}}_1^{[2]} = \begin{bmatrix} 0 & 0 & 0 \\ 0 & 0 & 0 \\ 0 & 0 & 1 \end{bmatrix} \quad \tilde{\boldsymbol{A}}_1^{[3]} = \begin{bmatrix} 0 & 0 & 0 \\ 0 & 0 & 0 \\ 0 & 0 & 0 \end{bmatrix}.$$

Similarly, Client 2 knows

$$\text{internal connections (including self loops): } \tilde{\boldsymbol{A}}_2^{[2]} = \begin{bmatrix} 1 & 1 & 1 \\ 1 & 1 & 1 \\ 1 & 1 & 1 \end{bmatrix},$$

$$\text{outgoing connections: } \tilde{\boldsymbol{A}}_1^{[2]} = \begin{bmatrix} 0 & 0 & 0 \\ 0 & 0 & 0 \\ 0 & 0 & 1 \end{bmatrix} \quad \tilde{\boldsymbol{A}}_3^{[2]} = \begin{bmatrix} 0 & 0 & 0 \\ 0 & 0 & 0 \\ 0 & 0 & 1 \end{bmatrix},$$

$$\text{incoming connections: } \tilde{\boldsymbol{A}}_2^{[1]} = \begin{bmatrix} 0 & 0 & 0 \\ 0 & 0 & 0 \\ 0 & 0 & 1 \end{bmatrix} \quad \tilde{\boldsymbol{A}}_2^{[3]} = \begin{bmatrix} 0 & 0 & 0 \\ 0 & 0 & 0 \\ 0 & 0 & 1 \end{bmatrix}.$$

and Client 3 knows

$$\text{internal connections (including self loops): } \tilde{\boldsymbol{A}}_3^{[3]} = \begin{bmatrix} 1 & 1 & 1 \\ 1 & 1 & 1 \\ 1 & 1 & 1 \end{bmatrix},$$

$$\text{outgoing connections: } \tilde{\boldsymbol{A}}_1^{[3]} = \begin{bmatrix} 0 & 0 & 0 \\ 0 & 0 & 0 \\ 0 & 0 & 0 \end{bmatrix} \quad \tilde{\boldsymbol{A}}_2^{[3]} = \begin{bmatrix} 0 & 0 & 0 \\ 0 & 0 & 0 \\ 0 & 0 & 1 \end{bmatrix},$$

$$\text{incoming connections: } \tilde{\boldsymbol{A}}_3^{[1]} = \begin{bmatrix} 0 & 0 & 0 \\ 0 & 0 & 0 \\ 0 & 0 & 0 \end{bmatrix} \quad \tilde{\boldsymbol{A}}_3^{[2]} = \begin{bmatrix} 0 & 0 & 0 \\ 0 & 0 & 0 \\ 0 & 0 & 1 \end{bmatrix}.$$

Using $\tilde{\boldsymbol{A}}_j^{[i]}, j \in \{1, 2, 3\}$, Client $i$ can compute both the degree matrix $\tilde{\boldsymbol{D}}^{[i]}$ and the normalized adjacency matrix $\hat{\boldsymbol{A}}_j^{[i]}$ for all $j \in [K]$. For simplicity, we assume that $\tilde{\boldsymbol{A}}_j^{[i]} = \hat{\boldsymbol{A}}_j^{[i]} \quad \forall i, j \in [K]$ in this example, as they differ only by a normalization constant.

Following Eq. 35, we define the intermediate matrix $\boldsymbol{B}_{ijk} = \tilde{\boldsymbol{A}}_k^{[i]} \hat{\boldsymbol{A}}_j^{[k]}$. The following steps outline the remaining computations:

1. **Calculating $\boldsymbol{B}_{ijk}$.**

   Client 1 calculates $\boldsymbol{B}_{211}$, $\boldsymbol{B}_{221}$, $\boldsymbol{B}_{231}$ as follows,

$$\boldsymbol{B}_{211} = \begin{bmatrix} 0 & 0 & 1 \\ 0 & 0 & 1 \\ 0 & 0 & 1 \end{bmatrix} \quad \boldsymbol{B}_{221} = \begin{bmatrix} 0 & 0 & 0 \\ 0 & 0 & 0 \\ 0 & 0 & 1 \end{bmatrix} \quad \boldsymbol{B}_{231} = \begin{bmatrix} 0 & 0 & 0 \\ 0 & 0 & 0 \\ 0 & 0 & 0 \end{bmatrix}. \tag{40}$$

**Table 4:** Statistics of the datasets.

| DATA | CORA | CITESEER | PUBMED | CHAMELEON | AMAZON PHOTO | AMAZON RATINGS |
|---|---|---|---|---|---|---|
| # CLASSES | 7 | 6 | 3 | 5 | 8 | 5 |
| $|\mathcal{V}|$ | 2708 | 3327 | 19717 | 2277 | 7650 | 24492 |
| $|\mathcal{E}|$ | 5278 | 4676 | 44327 | 36101 | 238162 | 93050 |
| # FEATURES | 1433 | 3703 | 500 | 2325 | 745 | 300 |
| EDGE HOMOPHILY RATIO | 0.81 | 0.74 | 0.80 | 0.23 | 0.82 | 0.38 |

Client 1 sends these matrices to Client 2. Notice that Client 2, by receiving $\boldsymbol{B}_{211}$, $\boldsymbol{B}_{221}$, $\boldsymbol{B}_{231}$, cannot reconstruct $\tilde{\boldsymbol{A}}_1^{[1]}$, $\tilde{\boldsymbol{A}}_3^{[1]}$, and $\tilde{\boldsymbol{A}}_1^{[3]}$ due to the low rank nature of the adjacency matrix. Moreover, Client 1 also prunes $\boldsymbol{B}_{ijk}$.

Similarly Client 3 calculates $\boldsymbol{B}_{213}$, $\boldsymbol{B}_{223}$, $\boldsymbol{B}_{233}$ as

$$\boldsymbol{B}_{213} = \begin{bmatrix} 0 & 0 & 0 \\ 0 & 0 & 0 \\ 0 & 0 & 0 \end{bmatrix} \quad \boldsymbol{B}_{223} = \begin{bmatrix} 0 & 0 & 0 \\ 0 & 0 & 0 \\ 0 & 0 & 1 \end{bmatrix} \quad \boldsymbol{B}_{233} = \begin{bmatrix} 0 & 0 & 1 \\ 0 & 0 & 1 \\ 0 & 0 & 1 \end{bmatrix}, \qquad (41)$$

which are sent to Client 2 after pruning. A similar procedure applies for the other combinations of $\boldsymbol{B}_{ijk}$, as outlined in Algorithm 3.

2. **Calculating $[\hat{\boldsymbol{A}}_j^{[i]}]^2$.**

   Upon receiving $\boldsymbol{B}_{211}$, $\boldsymbol{B}_{221}$, $\boldsymbol{B}_{231}$ and $\boldsymbol{B}_{213}$, $\boldsymbol{B}_{223}$, $\boldsymbol{B}_{233}$, Client 2 can compute $[\hat{\boldsymbol{A}}_1^{[2]}]^2$, $[\hat{\boldsymbol{A}}_2^{[2]}]^2$, $[\hat{\boldsymbol{A}}_3^{[2]}]^2$ as

$$[\hat{\boldsymbol{A}}_1^{[2]}]^2 = (\tilde{\boldsymbol{D}}^{[2]})^{-1} (\boldsymbol{B}_{211} + \boldsymbol{B}_{212} + \boldsymbol{B}_{213}), \qquad (42)$$

$$[\hat{\boldsymbol{A}}_2^{[2]}]^2 = (\tilde{\boldsymbol{D}}^{[2]})^{-1} (\boldsymbol{B}_{221} + \boldsymbol{B}_{222} + \boldsymbol{B}_{223}), \qquad (43)$$

$$[\hat{\boldsymbol{A}}_3^{[2]}]^2 = (\tilde{\boldsymbol{D}}^{[2]})^{-1} (\boldsymbol{B}_{231} + \boldsymbol{B}_{232} + \boldsymbol{B}_{233}). \qquad (44)$$

   Note that $\boldsymbol{B}_{212}$, $\boldsymbol{B}_{222}$, $\boldsymbol{B}_{232}$ can be computed locally in Client 2. Using Eqs. 42– 44, Client 2 can update its local 2-hop adjacency matrix without access to the global adjacency matrix. Clients 1 and 3 follow the same procedure to compute $[\hat{\boldsymbol{A}}_j^{[1]}]^2$ and $[\hat{\boldsymbol{A}}_j^{[3]}]^2$ for each $j \in \{1, 2, 3\}$, respectively.

This procedure can be extended to any number of hops $\ell$. At hop $\ell$, Client $i$ only accesses $[\hat{\boldsymbol{A}}_j^{[i]}]^\ell$ for all $j \in [K]$ and does not learn the full global matrix $[\hat{\boldsymbol{A}}]^\ell$. Moreover, if additional security is required, the summation over $\boldsymbol{B}_{ijk}$ could be performed on a secure server using homomorphic encryption.

# E   EXPERIMENTAL SETTING

In this section, we provide more details about the experiments in Section 6 and Appendix F. In Table 4, statistics of the different datasets are shown. The edge homophily ratio, measuring the fraction of edges that connect nodes with the same label, provides a measure of the homophily within the dataset. Typically, a value above $0.5$ is considered homophilic (Zheng et al., 2022). According to this rule-of-thumb, among the datasets considered in this paper, two would be considered heterophilic, i.e., Chameleon and Amazon Ratings.

Next, we provide the hyperparameters for the experiments. In Table 5, we provide the step sizes $\lambda$ and $\lambda_s$ for the gradient descent step during the training, the weight decay in the L2 regularization, the number of training iterations (epochs), the number of layers $L$ in the node feature embedding, the number of layers $L_s$ in the DECOUPLED GCN, the dimensionality of the NSFs, $d_s$, the pruning parameter $p$, and the model architecture of the node feature and node structure feature predictors, $\boldsymbol{f}_{\boldsymbol{\theta}_f}$ and $\boldsymbol{g}_{\boldsymbol{\theta}_s}$, respectively.

All the experiments are obtained using an Nvidia A30 with 24GB of memory.

**Table 5:** Hyper-parameters of the datasets.

| DATA | CORA | CITESEER | PUBMED | CHAMELEON | AMAZON PHOTO | AMAZON RATINGS |
|------|------|----------|--------|-----------|--------------|----------------|
| $\lambda$ | 0.002 | 0.002 | 0.008 | 0.003 | 0.005 | 0.002 |
| $\lambda_s$ | 0.002 | 0.002 | 0.008 | 0.003 | 0.005 | 0.002 |
| WEIGHT DECAY | 0.0005 | 0.0005 | 0.001 | 0.0003 | 0.001 | 0.0002 |
| EPOCHS | 40 | 60 | 125 | 60 | 150 | 70 |
| $L$ | 2 | 2 | 2 | 1 | 1 | 2 |
| $L_s$ | 10 | 20 | 20 | 1 | 3 | 5 |
| $d_s$ | 256 | 256 | 256 | 256 | 256 | 256 |
| $p$ | 30 | 30 | 30 | 30 | 30 | 30 |
| $\theta_f$ LAYERS | [ 1433,64,7 ] | [3703,64,6] | [500,128,3] | [2325,64,5] | [745,256,8] | [300,64,5] |
| $\theta_s$ LAYERS | [256,256,7] | [256,128,64,6] | [256,128,3] | [256,256,5] | [256,256,8] | [256,256,5] |

# F    ADDITIONAL RESULTS

In this appendix, we provide additional results that we could not include in the main paper due to space constraints. In particular, we provide results for different data partitioning methods: the Louvain algorithm, the K-means algorithm, and random partitioning.

The Louvain algorithm aims at finding communities with high link density, effectively creating subgraphs with limited interconnections. Here, we follow Zhang et al. (2021) where the global graph is first divided into a number of subgraphs using the Louvain algorithm. As we strive to have an even number of nodes among the clients, we inspect the size of the subgraphs and if any exceeds $n/K$ nodes, it is split in two. This procedure is repeated until no subgraph has more than $n/K$ nodes. Next, as the number of subgraphs may be larger than $K$, some subgraphs must be merged. To this end, we sort the subgraphs in descending order with respect to their size and assign the first $K$ subgraphs to the $K$ clients. If there are more than $K$ subgraphs, we assign the $(K + 1)$-th subgraph by iterating the $K$ clients and assigning it to the first client for which the total number of nodes after a merge would not exceed $n/K$. This process is repeated until there are only $K$ subgraphs remaining. This procedure results in subgraphs with approximately the same number of nodes and a low number of interconnections.

For the K-means algorithm, we follow Lei et al. (2023) and partition the data into $K$ clusters by proximity in the node feature space. Similarly to the Louvain partitioning, we split any subgraph that exceeds $n/K$ nodes in two and assign it to another subgraph such that the resulting number of nodes is less than $n/K$. Compared to the previous partitioning, the K-means approach does not consider the topology, hence, there will be more interconnections between the different subgraphs compared to the Louvain-based partitioning. Further, as highlighted in (Lei et al., 2023), as nodes with similar features tend to have the same label, this partitioning method creates a highly heterogeneous partitioning as each subgraph tends to have an over-representation of nodes of a given class.

Finally, we consider a random partitioning where each node is allocated to a client uniformly at random. As this partitioning does not take into account the topology or the features, we end up with a large number of interconnections where each subgraph has the same distribution over class labels, i.e., the data is homogeneous across clients. Notably, given the large number of interconnections, this setting arguably constitutes the most challenging scenario in subgraph FL as it is paramount to exploit the interconnections to achieve good performance.

## F.1    PERFORMANCE UNDER DIFFERENT PARTITIONING METHODS

In Table 6, we show the performance of FEDSTRUCT over each of the different partitionings for 10 clients and the train-val-test split according to 10%-10%-80%. We use this split to focus on a challenging semi-supervised scenario. Furthermore, each of the results is reported using the mean and standard deviation obtained from 10 independent runs.

First, as the central version of the training with GNN does not depend on the partitioning, it is the same as in Table 2. We also report the performance of central training using DECOUPLED GCN and when employing an MLP. By comparing the performance of CENTRAL GNN with CENTRAL MLP, one may infer the gains of accounting for the spatial structure within the data. As seen in the table,

**Table 6:** Classification accuracies for FEDSTRUCT with an underlying decoupled GCN. The results are shown for 10 clients with a 10–10–80 train-val-test split.

| | CORA | | | CITESEER | | | PUBMED | | |
|---|---|---|---|---|---|---|---|---|---|
| | LOUVAIN | RANDOM | KMEANS | LOUVAIN | RANDOM | KMEANS | LOUVAIN | RANDOM | KMEANS |
| CENTRAL GNN | 82.94± 1.26 | | | 69.37± 1.07 | | | 85.12± 1.15 | | |
| CENTRAL DGCN | 83.48 ± 1.31 | | | 69.05 ± 1.20 | | | 85.61 ± 0.18 | | |
| CENTRAL MLP | 65.47±0.019 | | | 63.67± 0.81 | | | 84.31± 0.22 | | |
| FEDAVG GNN | 81.05± 0.82 | 64.64± 1.87 | 66.47± 1.52 | 69.71± 0.73 | 65.41± 1.54 | 65.58± 1.09 | 85.67± 0.13 | 85.19± 0.35 | 85.79± 0.26 |
| FEDSGD GNN | 81.22± 0.99 | 66.00± 1.51 | 67.57± 1.28 | 69.25± 1.19 | 63.38± 0.76 | 64.64± 1.27 | 84.87± 0.76 | 84.66± 0.22 | 84.85± 0.33 |
| FEDAVG DGCN | 77.49± 3.06 | 62.60± 3.59 | 66.18± 1.35 | 69.88± 0.90 | 65.04± 1.96 | 67.10± 1.02 | 84.36± 0.19 | 84.42± 0.26 | 85.41± 0.32 |
| FEDSGD DGCN | 82.05± 0.92 | 67.15± 2.52 | 69.85± 1.06 | 68.51± 1.38 | 63.23± 0.85 | 65.39± 1.04 | 84.74± 0.17 | 83.92± 0.40 | 84.68± 0.32 |
| FEDAVG MLP | 68.59± 1.99 | 65.78± 1.84 | 64.81± 1.83 | 66.03± 0.80 | 64.41± 0.75 | 64.63± 1.09 | 85.81± 0.34 | 84.71± 0.26 | 84.75± 0.43 |
| FEDSGD MLP | 65.41± 1.33 | 65.67± 1.62 | 64.42± 1.60 | 64.87± 1.04 | 63.68± 0.74 | 64.11± 0.80 | 84.05± 0.41 | 84.29± 0.31 | 84.28± 0.41 |
| FEDSAGE+ | 81.06± 0.89 | 66.33± 1.69 | 67.35± 1.18 | 69.21± 1.17 | 63.93± 0.97 | 64.33± 0.79 | 84.31± 1.62 | 84.64± 0.37 | 84.91± 0.30 |
| FEDPUB | 78.75± 1.33 | 61.82± 1.84 | 64.38± 1.50 | 69.20± 1.09 | 62.91± 0.76 | 62.56± 1.29 | 85.16± 0.29 | 82.39± 0.41 | 83.85± 0.64 |
| FEDGCN-1HOP | 82.24± 1.13 | 82.91± 1.24 | 81.60± 0.95 | 70.24± 0.81 | 70.63± 0.87 | 69.68± 1.14 | 86.72± 0.17 | 86.05± 0.65 | 85.84± 0.53 |
| FEDGCN-2HOP | 81.88± 1.10 | 82.90± 0.95 | 82.00± 0.76 | 70.45± 0.85 | 70.49± 1.03 | 69.29± 1.45 | 86.62± 0.20 | 85.73± 0.77 | 85.83± 0.48 |
| FEDSTRUCT (DEG) | 82.18± 0.79 | 69.89± 1.85 | 71.31± 1.42 | 68.29± 1.35 | 63.54± 0.64 | 65.84± 1.36 | 84.65± 0.17 | 84.01± 0.38 | 84.65± 0.45 |
| FEDSTRUCT (FED⋆) | 81.27± 1.13 | 69.61± 1.87 | 70.89± 1.21 | 68.29± 1.47 | 63.38± 0.90 | 65.75± 1.15 | 84.71± 0.13 | 84.02± 0.35 | 84.77± 0.40 |
| FEDSTRUCT (H2V) | 81.30± 0.97 | 79.27± 0.90 | 78.36± 0.82 | 68.65± 1.49 | 65.43± 0.98 | 66.44± 1.07 | 82.23± 0.33 | 85.02± 0.43 | 84.81± 0.42 |
| FEDSTRUCT (H2V)-F | 81.02± 0.60 | 79.88± 0.92 | 78.79± 0.96 | 68.35± 1.45 | 65.93± 0.81 | 66.84± 0.55 | 83.03± 0.41 | 85.79± 0.60 | 85.47± 0.57 |
| LOCAL GNN | 75.36± 1.43 | 39.24± 1.64 | 47.08± 2.55 | 59.15± 1.52 | 39.88± 1.62 | 50.90± 5.66 | 76.40± 3.49 | 75.27± 0.59 | 75.00± 2.83 |
| LOCAL DGCN | 76.78± 1.18 | 41.35± 1.33 | 51.17± 1.98 | 59.62± 1.45 | 41.99± 1.35 | 53.04± 5.33 | 81.54± 0.47 | 76.65± 0.48 | 80.10± 0.49 |
| LOCAL MLP | 67.26± 2.05 | 40.34± 1.64 | 46.44± 2.15 | 54.62± 1.64 | 41.28± 0.89 | 52.02± 5.49 | 81.30± 0.44 | 75.98± 0.53 | 78.81± 0.38 |

| | CHAMELEON | | | AMAZON PHOTO | | | AMAZON RATINGS | | |
|---|---|---|---|---|---|---|---|---|---|
| | LOUVAIN | RANDOM | KMEANS | LOUVAIN | RANDOM | KMEANS | LOUVAIN | RANDOM | KMEANS |
| CENTRAL GNN | 54.38± 1.96 | | | 94.10± 0.30 | | | 41.42± 0.80 | | |
| CENTRAL DGCN | 55.05 ± 1.56 | | | 92.77 ± 0.38 | | | 41.05 ± 0.41 | | |
| CENTRAL MLP | 31.10± 1.71 | | | 87.69± 1.37 | | | 37.74± 0.39 | | |
| FEDAVG GNN | 45.87± 1.81 | 33.06± 1.84 | 37.74± 2.96 | 92.45± 0.80 | 90.25± 0.42 | 89.32± 0.73 | 40.09± 0.48 | 36.31± 0.79 | 37.35± 0.70 |
| FEDSGD GNN | 47.95± 2.01 | 36.80± 1.70 | 38.35± 2.04 | 93.91± 0.40 | 91.55± 0.34 | 91.41± 0.37 | 40.93± 0.88 | 35.96± 0.46 | 37.40± 0.32 |
| FEDAVG DGCN | 42.01± 1.47 | 31.08± 1.24 | 34.12± 3.40 | 90.87± 0.34 | 88.42± 0.60 | 87.76± 0.52 | 40.14± 0.60 | 37.52± 0.37 | 37.96± 0.36 |
| FEDSGD DGCN | 48.01± 1.43 | 34.78± 1.39 | 35.73± 2.02 | 91.97± 0.39 | 89.85± 0.39 | 90.06± 0.32 | 40.69± 0.49 | 37.92± 0.39 | 38.22± 0.33 |
| FEDAVG MLP | 32.63± 2.31 | 29.49± 1.37 | 28.43± 1.69 | 90.16± 1.15 | 85.39± 1.26 | 85.87± 2.50 | 37.84± 0.60 | 37.23± 0.38 | 37.40± 0.19 |
| FEDSGD MLP | 31.91± 2.14 | 33.13± 1.24 | 32.46± 1.86 | 87.66± 0.86 | 88.39± 1.07 | 87.99± 1.35 | 37.42± 0.40 | 37.65± 0.38 | 37.29± 0.51 |
| FEDSAGE+ | 47.70± 1.68 | 36.32± 1.59 | 37.73± 1.58 | 93.82± 0.34 | 91.33± 0.47 | 91.39± 0.40 | 40.54± 0.71 | 35.85± 0.39 | 37.55± 0.38 |
| FEDPUB | 47.80± 1.65 | 33.31± 1.37 | 34.82± 2.07 | 91.51± 0.32 | 88.05± 0.68 | 88.84± 0.58 | 40.37± 0.66 | 35.72± 0.60 | 37.00± 0.62 |
| FEDGCN-1HOP | 52.95± 3.33 | 48.23± 1.81 | 48.35± 1.68 | 93.28± 0.59 | 93.60± 0.28 | 93.62± 0.31 | 41.13± 0.52 | 40.76± 0.35 | 40.56± 0.23 |
| FEDGCN-2HOP | 48.87± 2.00 | 48.77± 1.73 | 49.14± 1.98 | 93.54± 0.36 | 93.72± 0.41 | 93.59± 0.31 | 41.19± 0.56 | 40.78± 0.56 | 40.82± 0.42 |
| FEDSTRUCT (DEG) | 49.66± 1.61 | 41.82± 1.78 | 42.16± 1.86 | 92.05± 0.44 | 89.72± 0.43 | 90.17± 0.28 | 40.93± 0.29 | 38.67± 0.66 | 38.78± 0.35 |
| FEDSTRUCT (FED⋆) | 49.36± 1.89 | 41.89± 1.67 | 42.54± 1.56 | 92.02± 0.41 | 89.78± 0.38 | 90.07± 0.42 | 41.03± 0.37 | 38.65± 0.44 | 38.89± 0.49 |
| FEDSTRUCT (H2V) | 55.74± 1.09 | 52.60± 1.25 | 53.40± 1.69 | 91.21± 0.42 | 90.93± 0.27 | 91.42± 0.72 | 40.61± 0.51 | 40.97± 0.64 | 41.06± 0.42 |
| FEDSTRUCT (H2V)-F | 55.75± 1.46 | 53.09± 1.85 | 52.95± 1.72 | 90.40± 0.58 | 90.74± 0.29 | 91.11± 0.61 | 40.86± 0.56 | 41.07± 0.56 | 41.23± 0.43 |
| LOCAL GNN | 46.71± 2.42 | 29.60± 1.25 | 30.84± 2.60 | 91.24± 0.73 | 77.12± 1.75 | 79.48± 1.59 | 40.23± 0.71 | 32.80± 0.43 | 35.73± 0.41 |
| LOCAL DGCN | 46.23± 1.70 | 28.55± 1.08 | 30.12± 3.01 | 90.64± 0.45 | 81.64± 0.95 | 83.09± 0.83 | 41.30± 0.58 | 34.16± 0.25 | 36.32± 0.43 |
| LOCAL MLP | 31.44± 2.10 | 21.17± 0.65 | 22.70± 2.42 | 89.27± 0.75 | 69.89± 1.34 | 76.05± 0.80 | 37.67± 0.67 | 33.23± 0.39 | 35.25± 0.38 |

exploiting the graph structure brings the largest benefits for Cora and Chameleon. Moreover, it can be seen for all datasets GNN and DECOUPLED GCN yield similar results for centralized training.

In similar spirit, the performance gap between LOCAL GNN (LOCAL MLP) and CENTRAL GNN (CENTRAL MLP) indicates the potential gains of employing collaborative learning between the clients. Notably, this gap depends on the partitioning. From Table 6, it can be seen that the gap is the largest for random partitioning followed by the K-means algorithm for all datasets. This is expected, as local training suffers from a large number of interconnections. For example, in Cora, the gap is 42.83%, 35.26%, and 6.54% for random, K-means, and Louvain partitioning, respectively. Furthermore, LOCAL GNN and LOCAL DCGN performs similarly.

Considering FEDSGD GNN and FEDSGD DGCN, it can be seen that all scenarios benefit from collaborative learning. For the Louvain partitioning, due to its community-based partitioning with the low number of interconnections, FEDSGD GNN and FEDSGD DGCN performs well for all datasets. For K-means and random partitioning, they performs worse, with the worst performance seen for Cora, Chameleon, and Amazon Ratings. FEDSAGE+ achieves similar performance to FEDSGD GNN whereas FEDPUB is inferior in most settings. The FEDSAGE+ IDEAL scheme, incorporating the knowledge of the node IDs of missing neighbors in other clients, is based on a mended graph with flawless in-painting of missing one-hop neighbors. Hence, this scheme serves as an upper bound on techniques based on in-painting of one-hop neighbors, such as FEDSAGE+ and FEDNI. Notably, this scheme completely violates privacy as the node features of missing clients cannot be shared between clients. From Table 6, we see that this scheme is robust to different partitionings by offering consistent performance close to the CENTRAL GNN.

We consider FEDSTRUCT with three different methods to generate NSFs: one-hot degree vectors (DEG), FED⋆, and our task-dependent method HOP2VEC (H2V), see Appendix A for information. The hyper-parameters are chosen as in Table 5. In Table 6, it can be seen that DEG, only being able to capture the local structure, perform worse than the methods able to capture more global properties of the graph. This is especially true for the K-means and random partitionings. FED⋆ achieves performance close to DEG, likely due to the random walk approach not being informative, hence, capturing essentially the same information as DEG, see Appendix A. Moreover, it can be seen that HOP2VEC performs close to FEDSAGE+ IDEAL for all scenarios and is sometimes superior, e.g., on the Chameleon datasets. This highlights the importance of going beyond the one-hop neighbors for some datasets, something that is not possible in FEDSAGE+ IDEAL.

### F.2 IMPACT OF PRUNING

To emphasize the strong performance of FEDSTRUCT and its robustness to structural noise, we compare it with a version of FEDSTRUCT that does not use pruning (denoted as -F) in Table 7. As shown in Table 7, FEDSTRUCT's performance remains very close to the version without pruning across all datasets, demonstrating its ability to handle the removal of structural information effectively.

## G DISCUSSION

### G.1 PRIVACY

Previous frameworks for subgraph FL (Zhang et al., 2021; Peng et al., 2022; Lei et al., 2023; Zhang et al., 2022; Liu et al., 2023; Zhang et al., 2024; Chen et al., 2021; Du and Wu, 2022) require the sharing of either original or generated node features and/or embeddings among clients. In stark contrast, our proposed framework, FEDSTRUCT, eliminates this need, sharing less sensitive information.

As outlined in Section 5, FEDSTRUCT requires some sharing of structural information between clients. We begin our discussion by mounting a potential attack to show that some information may be revealed from other clients. Thereafter, we discuss a strategy to mitigate such attacks by refraining from sharing individual information about nodes. Specifically, we present a slightly different version of FEDSTRUCT that relies on sharing aggregated quantities between clients without affecting the performance of the original version.

**Table 7:** Classification accuracy of FEDSTRUCT with pruning (pruning parameter $p = 30$) and without pruning. The results for pruning are presented in bold and the result without the pruning are shown with -F. The results are shown for 10 clients with a 10–10–80 train-val-test split.

| | CORA | | | CITESEER | | | PUBMED | | |
|---|---|---|---|---|---|---|---|---|---|
| CENTRAL GNN | 82.94± 1.26 | | | 69.37± 1.07 | | | 85.12± 1.15 | | |
| CENTRAL MLP | 65.47±0.019 | | | 63.67± 0.81 | | | 84.31± 0.22 | | |
| | LOUVAIN | RANDOM | KMEANS | LOUVAIN | RANDOM | KMEANS | LOUVAIN | RANDOM | KMEANS |
| FEDSTRUCT (DEG) | **82.18± 0.79** | **69.89± 1.85** | **71.31± 1.42** | **68.29± 1.35** | **63.54± 0.64** | **65.84± 1.36** | **84.65± 0.17** | **84.01± 0.38** | **84.65± 0.45** |
| FEDSTRUCT (FED★) | **81.27± 1.13** | **69.61± 1.87** | **70.89± 1.21** | **68.29± 1.47** | **63.38± 0.90** | **65.75± 1.15** | **84.71± 0.13** | **84.02± 0.35** | **84.77± 0.40** |
| FEDSTRUCT (H2V) | **81.30± 0.97** | **79.27± 0.90** | **78.36± 0.82** | **68.65± 1.49** | **65.43± 0.98** | **66.44± 1.07** | **82.23± 0.33** | **85.02± 0.43** | **84.81± 0.42** |
| FEDSTRUCT (DEG)-F | 81.90± 0.94 | 68.94± 2.66 | 71.09± 1.10 | 68.39± 1.52 | 63.45± 0.67 | 65.55± 0.98 | 84.75± 0.17 | 84.11± 0.33 | 84.75± 0.37 |
| FEDSTRUCT (FED★)-F | 81.68± 0.90 | 68.60± 2.85 | 70.61± 1.69 | 68.39± 1.48 | 63.74± 0.85 | 65.45± 1.21 | 84.77± 0.18 | 84.10± 0.33 | 84.80± 0.34 |
| FEDSTRUCT (H2V)-F | 81.02± 0.60 | 79.88± 0.92 | 78.79± 0.96 | 68.35± 1.45 | 65.93± 0.81 | 66.84± 0.55 | 83.03± 0.41 | 85.79± 0.60 | 85.47± 0.57 |

| | CHAMELEON | | | AMAZON PHOTO | | | AMAZON RATINGS | | |
|---|---|---|---|---|---|---|---|---|---|
| | LOUVAIN | RANDOM | KMEANS | LOUVAIN | RANDOM | KMEANS | LOUVAIN | RANDOM | KMEANS |
| CENTRAL GNN | 54.38± 1.96 | | | 94.10± 0.30 | | | 41.42± 0.80 | | |
| CENTRAL MLP | 31.10± 1.71 | | | 87.69± 1.37 | | | 37.74± 0.39 | | |
| FEDSTRUCT (DEG) | **49.66± 1.61** | **41.82± 1.78** | **42.16± 1.86** | **92.05± 0.44** | **89.72± 0.43** | **90.17± 0.28** | **40.93± 0.29** | **38.67± 0.66** | **38.78± 0.35** |
| FEDSTRUCT (FED★) | **49.36± 1.89** | **41.89± 1.67** | **42.54± 1.56** | **92.02± 0.41** | **89.78± 0.38** | **90.07± 0.42** | **41.03± 0.37** | **38.65± 0.44** | **38.89± 0.49** |
| FEDSTRUCT (H2V) | **55.74± 1.09** | **52.60± 1.25** | **53.40± 1.69** | **91.21± 0.42** | **90.93± 0.27** | **91.42± 0.72** | **40.61± 0.51** | **40.97± 0.64** | **41.06± 0.42** |
| FEDSTRUCT (DEG)-F | 49.80± 1.54 | 41.94± 1.58 | 42.23± 1.84 | 92.07± 0.35 | 90.31± 0.41 | 90.68± 0.34 | 40.95± 0.37 | 38.58± 0.53 | 38.88± 0.43 |
| FEDSTRUCT (FED★)-F | 49.23± 1.61 | 41.60± 1.80 | 42.45± 2.04 | 92.00± 0.45 | 90.12± 0.48 | 90.64± 0.35 | 40.98± 0.32 | 38.67± 0.60 | 38.97± 0.52 |
| FEDSTRUCT (H2V)-F | 55.75± 1.46 | 53.09± 1.85 | 52.95± 1.72 | 90.40± 0.58 | 90.74± 0.29 | 91.11± 0.61 | 40.86± 0.56 | 41.07± 0.56 | 41.23± 0.43 |

In FEDSTRUCT using task-agnostic NSFs, each client is provided with $S$, i.e., the NSFs of all clients. Consider an honest-but-curious client $i$ with NSFs $\{s_u : u \in \mathcal{V}_i\}$. Client $i$ may target client $j$ by identifying NSFs within client $j$ that closely match some of its own NSFs. By this simple procedure, client $i$ may compare the topology of the matched local nodes and infer the topology of some of the nodes within $j$. Furthermore, as $\theta_s$ is shared among all clients, client $i$ would potentially be able to guess the labels of the identified nodes within client $j$. Given a node's label and topology, it may be further possible to infer something about the node features. Using task-dependent NSF, i.e., HOP2VEC, the situation is similar.

Next, we discuss how to alter FEDSTRUCT to prevent reconstruction of the local NSFs, therefore mitigating the aforementioned attack, starting from the task-agnostic NSF generators. From Proposition 3, the summation over $\mathcal{V}$ may be split into a summation over clients. The prediction and local gradients for client $i$ may then be written as

$$\hat{\boldsymbol{y}}_v = \text{softmax}\Big( \sum_{j \in [K]} \sum_{u \in \mathcal{V}_j} \bar{A}_{vu} \boldsymbol{g}_{\boldsymbol{\theta}_s}(\boldsymbol{s}_u) + f_{\boldsymbol{\theta}_i}(v) \Big), \quad v \in \tilde{\mathcal{V}}_i, \tag{45}$$

$$\frac{\partial \mathcal{L}_i(\boldsymbol{\theta})}{\partial \theta_{s,q}} = \sum_{v \in \tilde{\mathcal{V}}_i} \sum_{j \in [K]} \sum_{u \in \mathcal{V}_j} \bar{A}_{vu} \frac{\partial \boldsymbol{g}_{\boldsymbol{\theta}_s}(\boldsymbol{s}_u)}{\partial \theta_{s,q}}(\hat{\boldsymbol{y}}_v - \boldsymbol{y}_v), \quad \forall q \in [|\boldsymbol{\theta}_s|]. \tag{46}$$

Notably, client $i$ may evaluate its local gradient provided access to

$$\bar{\boldsymbol{s}}_v^{(j)} = \sum_{u \in \mathcal{V}_j} \bar{A}_{vu} \boldsymbol{g}_{\boldsymbol{\theta}_s}(\boldsymbol{s}_u), \quad \forall v \in \tilde{\mathcal{V}}_i \tag{47}$$

$$\tilde{\boldsymbol{s}}_{vq}^{(j)} = \sum_{u \in \mathcal{V}_j} \bar{A}_{vu} \frac{\partial \boldsymbol{g}_{\boldsymbol{\theta}_s}(\boldsymbol{s}_u)}{\partial \theta_{s,q}}, \quad \forall q \in [|\boldsymbol{\theta}_s|], \forall v \in \tilde{\mathcal{V}}_i. \tag{48}$$

for each client $j \in [K]$. For client $j \neq i$ to be able to evaluate these quantities, it needs access to $\bar{A}_{vu}$ for all $v \in \tilde{\mathcal{V}}_i$ and $u \in \tilde{\mathcal{V}}$, i.e., the weighted sum of the $\ell$-hop paths, $\ell \in [L]$, from nodes in client $i$ to nodes in client $j$. Following Appendix D, client $j$ has access to $a_{vu}$ for $v \in \tilde{\mathcal{V}}_j$ and $u \in \mathcal{V}$. Notably, $a_{vu} \neq A_{uv}$ due to different degree normalizations. Hence, to obtain the required quantitites in Eq. 47 and Eq. 48, client $i$ may share $\{\bar{A}_{vu} : v \in \tilde{\mathcal{V}}_i, u \in \mathcal{V}_j\}$ with client $j \in [K]$ after the algorithm in Appendix D has finished.

Compared to FEDSTRUCT, the gradients of the above method cannot be calculated locally but bring the benefit of clients only knowing their own local NSFs rather than the individual NSFs of others.

Because of this, the training has to be done in two phases for each iteration where phase 1 accounts for each client $i \in [K]$ to collect the quantities in Eq. 47 and Eq. 48 from all other clients and where each client computes the local gradients and shares these with the server during phase 2. Note that the communication complexity in each training iteration does not increase as Eq. 47 and Eq. 48 amounts to sharing $|\tilde{\mathcal{V}}_i|c$ and $|\boldsymbol{\theta}_s|c$ parameters, respectively, where $c$ is the number of class labels. Hence, in total, each client $i$ must share $(|\tilde{\mathcal{V}} \setminus \tilde{\mathcal{V}}_i| + |\boldsymbol{\theta}_s|)c$ parameters in each iteration with the other clients. Note that this sharing can be made either via the server or via peer-to-peer links.

Although a formal privacy analysis seems formidable, we may follow an approach similar to (Lei et al., 2023) to argue around the difficulty of perfectly reconstructing the local NSFs. To this end, we take a conservative approach and consider honest-but-curious clients where all but one client is colluding. Hence, we may consider two clients where client 1 is the attacker and client 2 is the target. In each iteration, client 1 receives $\tilde{\boldsymbol{s}}_v^{(2)} \in \mathbb{R}^c$ and $\tilde{\boldsymbol{s}}_{vq}^{(2)} \in \mathbb{R}^c$ for $v \in \tilde{\mathcal{V}}_1$ and $q \in [|\boldsymbol{\theta}_2|]$. To evaluate $\tilde{\boldsymbol{s}}_v^{(2)}$, client 2 utilizes $\|\boldsymbol{a}_v\|_0$ $d_s$-dimensional NSFs where the L0-norm counts the number of non-zero elements. Hence, for each received aggregate $\tilde{\boldsymbol{s}}_v^{(2)}$, client 1 obtains $c$ observations from $\|\boldsymbol{a}_v\|_0$ $d_s$ unknowns. In total, client 1 will collect $|\tilde{\mathcal{V}}_1|$ such observations, i.e., it observes $c|\tilde{\mathcal{V}}_1|$ parameters. Assuming a, for client 2, the worst-case scenario with all $\boldsymbol{a}_v$ being non-zero in the same locations with $m$ non-zero elements, client 2 will use the same inputs for each observation in client 1. Hence, if $nd_s > c|\tilde{\mathcal{V}}_1|$, client 1 is unable to recover individual NSFs from client 2. Notably, client 2 has full control as it knows $m$ and can count the number of queries from client 1 and refuse to answer if it exceeds $md_s/c$. Similarly, to evaluate $\tilde{\boldsymbol{s}}_{vq}^{(2)}$, client 2 utilizes $\|\boldsymbol{a}_v\|_0$ $d_s$-dimensional inputs to compute a $c$-dimensional output. Hence, client 2 will share $|\boldsymbol{\theta}_s|c$ parameters, computed from $\|\boldsymbol{a}_v\|_0$ $d_s$ parameters $|\tilde{\mathcal{V}}_1|$ times. Hence, again considering the worst-case view, if $md_s > c|\boldsymbol{\theta}_s||\tilde{\mathcal{V}}_1|$, client 1 cannot reconstruct the local gradient. Again, client 2 may monitor the number of queries such that the condition is not violated.

When using Hop2Vec as the NSF generator, we no longer optimize over $\boldsymbol{\theta}_s$ but rather over the NSFs $\boldsymbol{S}$ directly. The problem we face in FedStruct is that each client has access to $\boldsymbol{S}$. To alleviate this issue, we use Theorem 2 to split the summation over $\mathcal{V}$ to obtain

$$\bar{\boldsymbol{s}}_v^{(j)} = \sum_{u \in \mathcal{V}_j} \bar{A}_{vu} \boldsymbol{s}_u, \quad \forall v \in \tilde{\mathcal{V}}_i. \tag{49}$$

Furthermore, the local gradients for the NSFs in client $i$ are given as

$$\nabla_{\boldsymbol{S}} \mathcal{L}_i(\boldsymbol{\theta}, \boldsymbol{S}) = \sum_{v \in \tilde{\mathcal{V}}_i} \bar{\boldsymbol{A}}_{v,:} (\hat{\boldsymbol{y}}_v - \boldsymbol{y}_v)^{\mathsf{T}}. \tag{50}$$

We note that Eq. 49 is required to compute the local predictions on $\tilde{\mathcal{V}}_i$, and hence to obtain Eq. 50. In this version of FedStruct, the training procedure will be divided into three phases. Phase 1 amounts to each client evaluating Eq. 49 for all other clients and sharing the results. This results in the sharing of $|\tilde{\mathcal{V}} \setminus \tilde{\mathcal{V}}_i|c$ in total for each client. Phase 2 amounts to each client evaluating its local gradient in Eq. 50 and sharing it with the server, similar to the original FedStruct. Next, the server aggregates the local gradients over $\boldsymbol{S}$ and partitions the result with respect to the entries residing in each client (the server must know the unique node-IDs in the clients) and returns only the NSF gradients corresponding to the nodes residing in each client. Using this procedure, clients will only have information about the NSFs pertaining to their local nodes.

Next, we again consider honest-but-curious clients with all but one colluding against a single target. We denote the colluding clients as client 1 and the target as client 2. Client 1 observes Eq. 49 which constitutes $c$ observations from $\|\boldsymbol{a}_v\|_0$ $c$-dimensional inputs. Assuming a worst-case scenario as above with all $\boldsymbol{a}_v$ having $m$ non-zero entries in the same locations, client 1 observes in total $c|\tilde{\mathcal{V}}_1|$ parameters from $mc$ inputs. Hence, to prevent client 1 from perfectly reconstructing individual NSFs, client 2 must ensure that $m > |\tilde{\mathcal{V}}_1|$. Again, client 2 has full control over this by counting the number of queries ensuring it is below $m$.

To summarize, we are able to alter FedStruct to reveal less information about individual nodes without impacting the performance. Further, each client may prevent perfect reconstruction from being possible by limiting the amount of information that is shared with other clients. To go beyond perfect reconstruction and understand the risk of approximate reconstruction seems formidable.

## G.2 HETEROPHILY

Employing a decoupled GCN within the FEDSTRUCT framework allows the nodes to access larger receptive fields compared to standard GNNs. This proves advantageous in handling heterophily graphs. This is because in heterophilic graphs, neighboring nodes may not provide substantial information about the node labels. In such scenarios, the mixing of embeddings from higher-order neighbors involves a greater number of nodes with the same class, promoting increased similarity among nodes of the same class. Higher-order neighbor mixing is one of the well-known approaches to deal with heterophilic graphs (Zheng et al., 2022). Notably, (Abu-El-Haija et al., 2019) aggregates embeddings from multi-hop neighbors showing superior performance compared to one-hop neighbor aggregation. To show why higher-order neighborhoods help in heterophilic settings, (Zhu et al., 2020) theoretically establish that, on average, the labels of 2-hop neighbors exhibit greater similarity to the ego node label when the labels of 1-hop neighbors are conditionally independent from the ego node.

Furthermore, the parameters $\{\beta_j\}_{j=1}^{L_s}$ can be trained or adjusted to control the mixing weight of different $l$-hop neighbors. This allows the network to explicitly capture both local and global structural information within the graph. Specifically, different powers of the normalized adjacency matrix $\hat{\boldsymbol{A}}^l$, $l \in [L_s]$, collect information from distinct localities in the graph. Smaller powers capture more local information, while larger powers tend to collect more global information. Consequently, selecting the appropriate parameters $\{\beta_j\}_{j=1}L_s$ enables the model to adapt to various structural properties within graphs.

An additional well-established strategy for addressing heterophilic graphs involves the discovery of potential neighbors, as proposed by (Zheng et al., 2022). The concept of potential neighbor discovery broadens the definition of neighbors by identifying nodes that may not be directly linked to the ego nodes but share similar neighborhood patterns in different regions of the graph. GEOM-GCN (Pei et al., 2020) introduces a novel approach by defining a geometric Euclidean space and establishing a new neighborhood for nodes that are proximate in this latent space. To this aim, FEDSTRUCT employs NSFs to foster long-range similarity between nodes that may not be in close proximity in the graph but exhibit proximity in the latent structural space. Specifically, the use of methods such as STRUC2VEC (Ribeiro et al., 2017) for generating NSFs enables the creation of vectors that share similarity, even among nodes that may not be directly connected in the graph. Consequently, the NSEs produced by these NSFs also demonstrate similarity, contributing to consistent node label predictions. This approach showcases FEDSTRUCT's capacity to capture latent structural relationships and enhance the model's ability to make accurate predictions in the context of heterophilic graphs.

The robustness of FEDSTRUCT in handling heterophilic graphs is demonstrated in Table 2 and Table 6 for the Chameleon and Amazon Ratings datasets. Such datasets are heterophilic datasets, with Chameleon having a higher degree of heterophily than Amazon Ratings (see Table 4). For both datasets FEDSTRUCT yields performance close to Central GNN and significantly outperforms FEDSAGE+ and FEDPUB. In Table 2, for the Chameleon dataset and 20 clients, FEDSTRUCT with HOP2VEC achieves an accuracy of 52.76% compared to 34.33% for FEDSAGE+. For the Amazon Ratings dataset and 20 clients, FEDSTRUCT with HOP2VEC achieves an accuracy of 41.16% compared to 36.09% for FEDSAGE+. Note that the improvement is larger for the more heterophilic dataset. Similar results can be observed for different partitioning methods, see Table 6. For Chameleon, FEDSTRUCT with HOP2VEC outperforms FEDSAGE+ with 6.33%, 16.88% and 15.73% for Louvain, random and K-means partitioning, respectively. For Amazon ratings, FEDSTRUCT with HOP2VEC outperforms FEDSAGE+ with 0.07%, 5.05%, and 3.12% for Louvain, random and K-means partitioning, respectively.

We highlight that we have not specifically optimized the parameters of FEDSTRUCT to handle heterophilic graph structures, hence it might be possible to improve the performance of FEDSTRUCT for these heterophilic graphs.

The robustness of FEDSTRUCT to different degrees of homophily/heterophily (in contrast to frameworks such as FEDSAGE+ and FEDPUB), underscores the adaptability and effectiveness of our framework to diverse graph scenarios, affirming its potential for a wide range of applications.

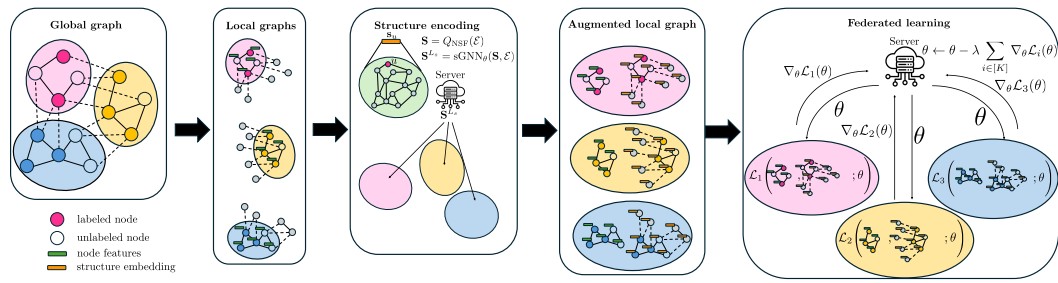

**Figure 5:** FEDSTRUCT framework when the server has knowledge of the global graph's connections. **Global Graph:** underlying graph consisting of interconnected subgraphs. **Local graphs:** clients' subgraphs augmented with external nodes (without features or labels). **Structure encoding:** The server generates node structure features and node structure embeddings for each node and shares them with the clients. **Augmented local graphs:** Generate node feature embeddings. **Federated learning:** Federated learning step exploiting node feature embeddings and node structure embeddings.

# H  FEDSTRUCT WITH KNOWLEDGE OF THE GLOBAL GRAPH

In this section, we discuss FEDSTRUCT for the scenario where the server has complete knowledge of the global graph's connections. This scenario is commonly encountered in applications such as smart grids and pandemic prediction. Importantly, also in this scenario we assume that the clients remain unaware of the local graphs of other clients.

If the server has knowledge of the global graph, the NSEs may be computed centrally. Hence, contrary to the case where the central server lacks knowledge of the global graph, any GNN model may be used, i.e., it is not restricted to decoupled GCN (although, in Sec. H.3, we discuss the advantages of using a decoupled GCN). Furthermore, any conventional NSF methods may be used such as NODE2VEC and GDV that require knowledge of the $L$-hop neighborhood.

At each client $i \in [K]$, node prediction is performed based on both the NFEs $\boldsymbol{h}_v$ and *node structure embeddings* (NSEs), $\boldsymbol{z}_v$. To this aim, we pass the NFEs and NSEs through a fully connected layer with parameters $\boldsymbol{\Theta}_{\mathsf{c}}$,

$$\hat{\boldsymbol{y}}_v = \mathrm{softmax}(\boldsymbol{\Theta}_{\mathsf{c}}^{\mathsf{T}}(\boldsymbol{z}_v || \boldsymbol{h}_v)) = \mathrm{softmax}\big((\boldsymbol{\Theta}_{\mathsf{c}}^{(\mathsf{s})})^{\mathsf{T}}\boldsymbol{z}_v + (\boldsymbol{\Theta}_{\mathsf{c}}^{(\mathsf{f})})^{\mathsf{T}}\boldsymbol{h}_v\big) \quad \forall v \in \mathcal{V}_i \,. \qquad (51)$$

We denote the vectorized version of $\boldsymbol{\Theta}_{\mathsf{c}}$ by $\boldsymbol{\theta}_{\mathsf{c}}$. From equation 51, we have that $\boldsymbol{\theta}_{\mathsf{c}} = \boldsymbol{\theta}_{\mathsf{c}}^{(\mathsf{s})} || \boldsymbol{\theta}_{\mathsf{c}}^{(\mathsf{f})}$.

As the server is able to generate the NSEs, at each iteration, NSE updates will be sent to the corresponding clients. For clients to send back gradient updates, they require the NSEs and their derivatives with respect to the generator weight parameters, as seen in Proposition 4.

The FEDSTRUCT framework for this setting is conceptually shown in Figure 5.

Using a generic GNN, the local gradients are given in the following theorem.

**Proposition 4.** *Let $\mathcal{L}_i(\boldsymbol{\theta})$, $\boldsymbol{\theta} = (\boldsymbol{\theta}_{\mathsf{f}} || \boldsymbol{\theta}_{\mathsf{s}} || \boldsymbol{\theta}_{\mathsf{c}})$, be the local training loss for client $i$ in Eq. 4 and $\hat{\boldsymbol{y}}_v$ be given in Eq. 51. The gradient*

$$\nabla_{\boldsymbol{\theta}} \mathcal{L}_i(\boldsymbol{\theta}) = || \left( \frac{\partial \mathcal{L}_i(\boldsymbol{\theta})}{\partial \theta_j} \quad \forall j \in [|\boldsymbol{\theta}|] \right) \qquad (52)$$

*is given by*

$$\frac{\partial \mathcal{L}_i(\boldsymbol{\theta})}{\partial \theta_{\mathsf{s},j}} = \sum_{v \in \tilde{\mathcal{V}}} \frac{\partial \boldsymbol{z}_v}{\partial \theta_{\mathsf{s},j}} \boldsymbol{\Theta}_{\mathsf{c}}^{(\mathsf{s})}(\hat{\boldsymbol{y}}_v - \boldsymbol{y}_v) \quad \forall j \in [|\boldsymbol{\theta}_{\mathsf{s}}|],$$

$$\frac{\partial \mathcal{L}_i(\boldsymbol{\theta})}{\partial \theta_{\mathsf{f},j}} = \sum_{v \in \tilde{\mathcal{V}}} \frac{\partial \boldsymbol{h}_v}{\partial \theta_{\mathsf{f},j}} \boldsymbol{\Theta}_{\mathsf{c}}^{(\mathsf{f})}(\hat{\boldsymbol{y}}_v - \boldsymbol{y}_v) \quad \forall j \in [|\boldsymbol{\theta}_{\mathsf{f}}|],$$

$$\frac{\partial \mathcal{L}_i(\boldsymbol{\theta})}{\partial \theta_{\mathsf{c},j}^{(\mathsf{s})}} = \sum_{v \in \tilde{\mathcal{V}}} \frac{\partial \big((\boldsymbol{\Theta}_{\mathsf{c}}^{(\mathsf{s})})^{\mathsf{T}} \boldsymbol{z}_v\big)}{\partial \theta_{\mathsf{c},j}^{(\mathsf{s})}} (\hat{\boldsymbol{y}}_v - \boldsymbol{y}_v) \quad \forall j \in [|\boldsymbol{\theta}_{\mathsf{c}}^{(\mathsf{s})}|],$$

$$\frac{\partial \mathcal{L}_i(\boldsymbol{\theta})}{\partial \theta_{\mathsf{c},j}^{(\mathsf{f})}} = \sum_{v \in \tilde{\mathcal{V}}} \frac{\partial \big((\boldsymbol{\Theta}_{\mathsf{c}}^{(\mathsf{f})})^{\mathsf{T}} \boldsymbol{h}_v\big)}{\partial \theta_{\mathsf{c},j}^{(\mathsf{f})}} (\hat{\boldsymbol{y}}_v - \boldsymbol{y}_v) \quad \forall j \in [|\boldsymbol{\theta}_{\mathsf{c}}^{(\mathsf{f})}|] \,,$$

*where $\theta_{\cdot,j}$ denotes the $j$-th entry of vector $\boldsymbol{\theta}_{\cdot,j}$.*

---

**Algorithm 4** Algorithm for FEDSTRUCT with knowledge of the global graph

---

**input** Global graph $\mathcal{G}$, sGNN and fGNN models, NSF generator function $\boldsymbol{Q}_{\mathsf{NSF}}$, $K$ clients with respective subgraphs $\{\mathcal{G}_i\}_{i=1}^K$, model parameters $\boldsymbol{\theta} = (\boldsymbol{\theta}_{\mathsf{f}} || \boldsymbol{\theta}_{\mathsf{s}} || \boldsymbol{\theta}_{\mathsf{c}})$
   $\boldsymbol{S} \leftarrow \boldsymbol{Q}_{\mathsf{NSF}}(\mathcal{E})$
   **for** e=1 to Epochs **do**
      $\boldsymbol{S}^{(L,)} \leftarrow \mathrm{sGNN}_{\boldsymbol{\theta}_s}(\boldsymbol{S}, \mathcal{E})$
      **for** i=1 to K **do**
         Client $i$ collects $\boldsymbol{\theta}$ from the server
         Client $i$ collects $\{\boldsymbol{z}_v, \forall v \in \tilde{\mathcal{V}}_i\}$ from the server
         Client $i$ collects $\{\frac{\partial \boldsymbol{z}_v}{\partial \boldsymbol{\theta}_s}, \forall v \in \tilde{\mathcal{V}}_i\}$ from the server
         $\boldsymbol{H}_i^{(L)} = \mathrm{fGNN}_{\boldsymbol{\theta}_i}(\boldsymbol{X}_i, \mathcal{E}_i)$
         **for** $v \in \tilde{\mathcal{V}}_i$ **do**
            $\hat{\boldsymbol{y}}_v = \mathrm{softmax}(\boldsymbol{\Theta}_{\mathsf{c}}^{\mathsf{T}} . (\boldsymbol{z}_v || \boldsymbol{h}_v))$
         **end for**
         Calculate $\mathcal{L}_i(\boldsymbol{\theta})$ based on Eq. 4
         Calculate $\nabla_{\boldsymbol{\theta}} \mathcal{L}_i(\boldsymbol{\theta})$ from Proposition 4
         Send $\nabla_{\boldsymbol{\theta}} \mathcal{L}_i(\boldsymbol{\theta})$ to the server
      **end for**
      Calculate $\nabla_{\boldsymbol{\theta}} \mathcal{L}(\boldsymbol{\theta})$ based on Eq. 5
      $\boldsymbol{\theta} \leftarrow \boldsymbol{\theta} - \lambda \nabla_{\boldsymbol{\theta}} \mathcal{L}(\boldsymbol{\theta})$
   **end for**

---

The proof of the theorem is given in Section H.1.

Note that client $i$ cannot compute $\nabla_{\boldsymbol{\theta}} \mathcal{L}_i(\boldsymbol{\theta})$ directly as calculating $\frac{\partial \boldsymbol{z}_v}{\partial \theta_{s,j}}$ requires knowledge of the global graph connections. Hence, the server must provide $\frac{\partial \boldsymbol{z}_v}{\partial \theta_{s,j}}$ along with $\boldsymbol{z}_v$ for all $v \in \tilde{\mathcal{V}}_i$. The FEDSTRUCT framework when the server has knowledge of the global graph is described in Alg. 4.

### H.1 PROOF OF PROPOSITION 4

Using Eq. 23 we have

$$\frac{\partial \mathcal{L}_i(\boldsymbol{\theta})}{\partial \theta_j} = \sum_{v \in \tilde{\mathcal{V}}_i} \frac{\partial \boldsymbol{z}_v}{\partial \theta_j} (\hat{\boldsymbol{y}}_v - \boldsymbol{y}_v),$$

where $\boldsymbol{q}_v = (\boldsymbol{\Theta}_{\mathsf{c}}^{(\mathsf{s})})^{\mathsf{T}} \boldsymbol{z}_v + (\boldsymbol{\Theta}_{\mathsf{c}}^{(\mathsf{f})})^{\mathsf{T}} \boldsymbol{h}_v$. Taking the derivative of $\boldsymbol{z}_v$ with respect to different entries of $\boldsymbol{\theta}$ leads to

$$\frac{\partial \boldsymbol{q}_v}{\partial \theta_{\mathsf{s},j}} = \frac{\partial \boldsymbol{z}_v}{\partial \theta_{\mathsf{s},j}} \boldsymbol{\Theta}_{\mathsf{c}}^{(\mathsf{s})} \quad \forall j \in [|\boldsymbol{\theta}_{\mathsf{s}}|] \tag{53}$$

$$\frac{\partial \boldsymbol{q}_v}{\partial \theta_{\mathsf{f},j}} = \frac{\partial \boldsymbol{h}_v}{\partial \theta_{\mathsf{f},j}} \boldsymbol{\Theta}_{\mathsf{c}}^{(\mathsf{f})} \quad \forall j \in [|\boldsymbol{\theta}_{\mathsf{f}}|] \tag{54}$$

$$\frac{\partial \boldsymbol{q}_v}{\partial \theta_{\mathsf{c},j}^{(\mathsf{s})}} = \frac{\partial \big((\boldsymbol{\Theta}_{\mathsf{c}}^{(\mathsf{s})})^{\mathsf{T}} \boldsymbol{z}_v\big)}{\partial \theta_{\mathsf{c},j}^{(\mathsf{s})}} \quad \forall j \in [|\boldsymbol{\theta}_{\mathsf{c}}^{(\mathsf{s})}|] \tag{55}$$

$$\frac{\partial \boldsymbol{q}_v}{\partial \theta_{\mathsf{c},j}^{(\mathsf{f})}} = \frac{\partial \big((\boldsymbol{\Theta}_{\mathsf{c}}^{(\mathsf{f})})^{\mathsf{T}} \boldsymbol{h}_v\big)}{\partial \theta_{\mathsf{c},j}^{(\mathsf{f})}} \quad \forall j \in [|\boldsymbol{\theta}_{\mathsf{c}}^{(\mathsf{f})}|]. \tag{56}$$

Substituting Eq. 53, Eq. 54, Eq. 55, and Eq. 56 into Eq. 23 concludes the proof.

### H.2 RESULTS

As aforementioned, knowledge of the adjacency matrix allows FEDSTRUCT to utilize NSF generators such as GDV and N2V, see Appendix A. In Table 8, we provide some results for these NSFs and benchmark them against GLOBAL DGCN and FEDSTRUCT (H2V) without knowledge of the

**Table 8:** Classification accuracies for FEDSTRUCT with and without knowledge of the adjacency matrix. The results are shown for 10 clients with a 10–10–80 train-val-test split.

| | CORA | | | CITESEER | | | PUBMED | | |
|---|---|---|---|---|---|---|---|---|---|
| CENTRAL DGCN | $83.72 \pm 0.64$ | | | $68.69 \pm 1.04$ | | | $86.26 \pm 0.34$ | | |
| | LOUVAIN | RANDOM | KMEANS | LOUVAIN | RANDOM | KMEANS | LOUVAIN | RANDOM | KMEANS |
| **GLOBAL ADJACENCY MATRIX NOT KNOWN TO SERVER** | | | | | | | | | |
| FEDSTRUCT (DGCN+H2V) | $81.34\pm1.40$ | $79.62\pm0.85$ | $80.34\pm0.62$ | $68.48\pm0.98$ | $66.34\pm1.03$ | $66.62\pm1.59$ | $83.39\pm0.34$ | $85.48\pm0.29$ | $85.93\pm0.30$ |
| **GLOBAL ADJACENCY MATRIX KNOWN TO SERVER** | | | | | | | | | |
| FEDSTRUCT (DGCN+GDV) | $81.27\pm1.56$ | $72.09\pm1.70$ | $74.55\pm1.47$ | $68.82\pm0.91$ | $64.92\pm1.04$ | $66.22\pm1.37$ | $84.98\pm0.21$ | $85.59\pm0.26$ | $85.88\pm0.21$ |
| FEDSTRUCT (DGCN+N2V) | $82.26\pm1.21$ | $80.76\pm0.91$ | $80.97\pm1.01$ | $68.46\pm1.17$ | $66.88\pm1.19$ | $67.23\pm0.92$ | $84.84\pm0.23$ | $86.87\pm0.31$ | $87.08\pm0.24$ |
| FEDSTRUCT (GNN+H2V) | $80.87\pm0.51$ | $66.84\pm0.71$ | $68.06\pm1.22$ | $68.89\pm0.99$ | $63.77\pm1.71$ | $64.66\pm0.96$ | $76.04\pm6.77$ | $85.27\pm0.45$ | $82.92\pm4.92$ |
| FEDSTRUCT (GNN+GDV) | $80.62\pm0.79$ | $69.50\pm1.23$ | $69.86\pm1.94$ | $68.96\pm1.09$ | $64.78\pm1.11$ | $64.90\pm1.17$ | $81.56\pm5.46$ | $85.12\pm0.40$ | $85.36\pm0.30$ |
| FEDSTRUCT (GNN+ N2V) | $81.23\pm1.02$ | $75.24\pm1.00$ | $75.91\pm0.90$ | $69.22\pm1.26$ | $64.80\pm1.71$ | $65.53\pm1.15$ | $76.11\pm6.80$ | $86.24\pm0.87$ | $81.04\pm6.28$ |

| | CHAMELEON | | | AMAZON PHOTO | | | AMAZON RATINGS | | |
|---|---|---|---|---|---|---|---|---|---|
| CENTRAL DGCN | $54.39 \pm 2.24$ | | | $92.27 \pm 0.79$ | | | $40.94 \pm 0.46$ | | |
| | LOUVAIN | RANDOM | KMEANS | LOUVAIN | RANDOM | KMEANS | LOUVAIN | RANDOM | KMEANS |
| **GLOBAL ADJACENCY MATRIX NOT KNOWN TO SERVER** | | | | | | | | | |
| FEDSTRUCT (DGCN+H2V) | $53.15\pm1.26$ | $52.37\pm0.95$ | $52.85\pm2.44$ | $90.56\pm0.58$ | $90.81\pm0.73$ | $91.07\pm0.79$ | $40.93\pm0.39$ | $41.00\pm0.27$ | $40.86\pm0.50$ |
| **GLOBAL ADJACENCY MATRIX KNOWN TO SERVER** | | | | | | | | | |
| FEDSTRUCT (DGCN+GDV) | $48.08\pm1.64$ | $39.91\pm1.07$ | $40.76\pm2.76$ | $91.68\pm0.72$ | $90.51\pm0.55$ | $90.33\pm0.57$ | $41.43\pm0.51$ | $39.54\pm0.41$ | $39.65\pm0.43$ |
| FEDSTRUCT (DGCN+N2V) | $49.23\pm1.58$ | $43.34\pm1.43$ | $44.99\pm2.88$ | $91.86\pm0.78$ | $91.53\pm0.46$ | $91.65\pm0.44$ | $41.74\pm0.30$ | $41.84\pm0.38$ | $41.28\pm0.55$ |
| FEDSTRUCT (GNN+H2V) | $40.42\pm2.01$ | $38.36\pm2.10$ | $39.47\pm1.53$ | $93.36\pm0.49$ | $91.07\pm0.57$ | $90.81\pm0.39$ | $37.58\pm0.57$ | $35.13\pm0.44$ | $35.65\pm0.36$ |
| FEDSTRUCT (GNN+GDV) | $46.40\pm2.32$ | $38.80\pm2.27$ | $38.91\pm2.31$ | $93.17\pm1.10$ | $91.52\pm1.00$ | $91.63\pm0.34$ | $41.17\pm0.53$ | $36.74\pm0.83$ | $37.41\pm0.78$ |
| FEDSTRUCT (GNN+N2V) | $48.46\pm2.21$ | $45.12\pm2.14$ | $46.02\pm2.77$ | $91.78\pm1.47$ | $92.00\pm0.59$ | $91.59\pm0.89$ | $40.96\pm0.33$ | $40.29\pm0.50$ | $40.12\pm0.60$ |

adjacency matrix. While FEDSTRUCT (GDV) is mostly inferior to FEDSTRUCT (H2V), leveraging NODE2VEC as the NSF generator boosts the performance even further in several scenarios. For example, on Cora with Louvain partitiong, the performance is improved from 81.23% to 82.78%. Even more, arbitrary GNN architectures are supported when the global adjacency matrix is available at the server and may potentially improve performance even further.

## H.3   GNN vs Decoupled Graph Convolutional Network

As mentioned earlier, if the server has knowledge of the global graph, FEDSTRUCT can operate with any underlying GNN model. However, in Figure 6 we demonstrate the advantages of incorporating a decoupled GCN into our framework. To this aim, we provide results for FEDSTRUCT with both an underlying decoupled GCN and a standard GNN (GRAPHSAGE) for a semi-supervised learning scenario with varying fraction of labeled training nodes Specifically, we show the accuracy for scenarios involving 50%, 10%, and 1% labeled nodes. For 50% and 10% of labeled nodes, FEDSTRUCT with both underlying decoupled GCN and standard GNN yield similar results. However, in a heavily semi-supervised scenario (as encountered in applications like anti-money laundering), the accuracy of FEDSTRUCT with standard GNN experiences a significant decline. In contrast, FEDSTRUCT with decoupled GCN achieves performance close to that of central GNN even in such a challenging scenario. This highlights the critical role of employing a decoupled GCN to effectively tackle the semi-supervised learning scenarios. Moreover, knowledge of the global graph edges enables the use of NSFs such as NODE2VEC which, as seen in Figure 6, sometimes enhances the performance compared to HOP2VEC.

## H.4   COMMUNICATION COMPLEXITY

In this Section, we discuss the communication complexity of FEDSTRUCT when the server has knowledge of the global graph. Note that, in this case, the NSEs can be computed by the server. Hence, no sharing of information between the clients is needed and clients only communicate with the server.

During the training, in each training round, each client collects $\boldsymbol{\theta}$ and returns $\nabla_{\boldsymbol{\theta}}\mathcal{L}_i(\boldsymbol{\theta}, \boldsymbol{S})$ to the server, totaling a communication of $2E \cdot K \cdot |\boldsymbol{\theta}|$ parameters during the training. This is consistent

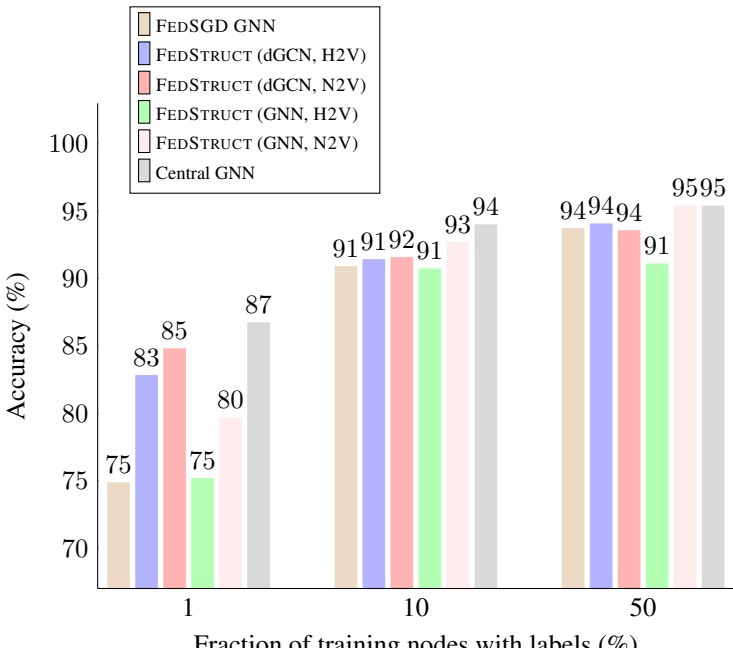

**Figure 6:** Accuracy vs fraction of training nodes with labels for FEDSTRUCT with underlying decoupled GCN and underlying standard GNN (GRAPHSAGE) on the Amazon Photo dataset with K-means partitioning.

**Table 9:** Communication Complexity of FEDSTRUCT when the Server has Knowledge of the Global Graph.

| DATA | BEFORE TRAINING | TRAINING |
|---|---|---|
| FEDSTRUCT | 0 | $\mathcal{O}(E \cdot K \cdot |\boldsymbol{\theta}| + E \cdot n \cdot d \cdot |\boldsymbol{\theta}|)$ |
| FEDSTRUCT + HOP2VEC | 0 | $\mathcal{O}(E \cdot K \cdot |\boldsymbol{\theta}| + E \cdot n \cdot d \cdot |\boldsymbol{\theta}| + E \cdot K \cdot n \cdot d + E \cdot n^2 \cdot d^2)$ |

across all versions of FEDSTRUCT. Additionally, the server sends $\boldsymbol{z}_v$, $\forall v \in \tilde{\mathcal{V}}_i$, and $\{\frac{\partial \boldsymbol{z}_v}{\partial \boldsymbol{\theta}_i}, \forall v \in \tilde{\mathcal{V}}_i\}$ to each client, adding up to $E \cdot n \cdot d(|\boldsymbol{\theta}| + 1)$ parameters. The dominant term of the complexity is $E \cdot n \cdot d \cdot |\boldsymbol{\theta}|$, which scales with $n$. The complexity is therefore of the same order as that of FEDSAGE+.

The use of HOP2VEC entails some additional communication complexity, corresponding to the collection of $\{\frac{\partial \boldsymbol{z}_v}{\partial \boldsymbol{s}_q}, \forall v \in \tilde{\mathcal{V}}_i, q \in \mathcal{V}\}$ by client $i$, constituting $E \cdot n^2 \cdot d^2$ parameters, and sending $\nabla_{\boldsymbol{S}} \mathcal{L}_i(\boldsymbol{\theta}, \boldsymbol{S})$ to the server, constituting $E \cdot K \cdot n \cdot d$ parameters. In practical scenarios with large graphs, $E \cdot n^2 \cdot d^2$ is the dominant term out of the two.

With HOP2VEC, the dominant complexity term is on the order of $n^2$, which is impractical for large networks. It should be noted, however, that real-world graph-structured datasets are sparse. Furthermore, each node only depends on its $L_{\mathsf{s}}$-hop neighbors. Hence, many of the values $\{\frac{\partial \boldsymbol{z}_v}{\partial \boldsymbol{s}_q}, \forall v \in \tilde{\mathcal{V}}_i, q \in \mathcal{V}\}$ are zero and do not need to be communicated. Assuming an average node degree $\bar{d}$ and $L_{\mathsf{s}}$ layers, for which each node has access to $\bar{d}^{L_{\mathsf{s}}}$ nodes, the complexity of FEDSTRUCT with HOP2VEC is on the order of $\min\{n \cdot \bar{d}^{L_{\mathsf{s}}}, n^2\}$. For small average degree $\bar{d}$ and $L_{\mathsf{s}}$, the complexity can therefore be reduced significantly. The communication complexity is reported in Table 9.

