# OpenReview forum: "Decoupled Subgraph Federated Learning"
_ICLR.cc/2025/Conference — ICLR 2025 Poster_

### Official Review · Reviewer_mTWE · 2024-10-30

**Soundness:** 3
**Presentation:** 4
**Contribution:** 3
**Rating:** 6
**Confidence:** 4

**Summary:**

The paper proposes a novel SFL method called FEDSTRUCT, which leverages the augmented explicit structure $\bar{A}$ to promote the SFL model performance. Moreover, they propose HOP2VEC to learn local structure embedding. FEDSTRUCT precalculates the $\bar{A}$ with privacy protection and prunes the $\hat{A}$ matrix to decrease the calculation complexity and communication costs, thus balancing the communication-privacy-accuracy trilemma.

**Strengths:**

1. The proposed method is novel, utilizing augmented explicit structure which can be regarded as global knowledge to promote the performance of the SFL model.
2. Utilizing pruning skills decrease the calculation complexity and communication costs.
3. Well written and well formulated problem.

**Weaknesses:**

1. **Focus on Privacy**. How to  obstain the local L-hop combined adjacency matrix while not share the L-hop global Adjacency Matrix maybe play the core role in FEDSTRUCT. In APP D, the equations [30] [31], [32], what does the $\hat{A}^{[K]}_j$ mean? Should be $\hat{A}^{[k]}_j$? If so , the next question, for client $i$, how does it know all $\hat{A}^{[k]}_j$ for $ k \in [K]$ without sharing the global adjacency matrix in all clients.  So another question when computing, the $\tilde{A}^{[i]}_j \in \mathbb{R}^{|\tilde{V}_i| \times |{V}_j|}$, the same to $\hat{A}^{[i]}_j$? So $\hat{A}^{[i]}_k \times \hat{A}^{[k]}_j$ should be $\mathbb{R}^{|\tilde{V}_k| \times |{V}_k|}  \times \mathbb{R}^{|\tilde{V}_k| \times |{V}_j|}$， but according the definition before, the $|\tilde{V}_k| \neq |{V}_k| $, how does the computation continue? Maybe I miss something? I really hope you can explain it for me to understand the feasibility of FEDSTRUCT. That's my main concern about this paper.

2. **About the hyperparameters.** **1)** The analysis of $\beta$ is not enough, an essential parameter in FEDSTRUCT for various homophilic and heterophilic graphs, which directly dominate the performance and affect the  judgment of FEDSTRUCT's contributions. **2)**  It is so strange that the parameter $L_s$ and $L$ is set to1 for heterophilic graph chameleon in table 5. As the author states in lines 297-299, a heterophilic graph should own multi-hop nodes and a high-frequency filter to augment the local graph representation.

**Questions:**

see weaknesses.

If the first questions can be well explained, the rating should be higher.

---

> ### Author Response · Authors · 2024-11-15
> **Response to Weakness 1**
>
> We appreciate the reviewer’s thoughtful comments and address them in detail below. If our responses satisfactorily address the reviewer’s concerns, we would be grateful if they would consider adjusting their score accordingly.
>
> **Weakness 1:**
> We sincerely thank the reviewer for their thorough analysis and for highlighting the potential confusion surrounding Equations (30), (31), and (32) in the paper. We confirm that the symbol $K$ should indeed be replaced by $k$, which represents the client index in these equations. Additionally, the dimensions of $\boldsymbol{\tilde{A}}^{[i]}_j$ and $\boldsymbol{\hat{A}}^{[i]}_j$ are $\vert\mathcal{V}_i\vert \times \vert\mathcal{V}_j\vert$, as noted. We have incorporated these corrections in the revised paper.
>
> To clarify how our privacy-preserving scheme works in practice
> (i.e., to show how each client obtains its local partition of L-hop combined adjacency matrix without sharing the L-hop global adjacency matrix), we start by outlining the information accessible to each client  $i \in [K]$  at the outset:
>
> Each client  $i$  has access to:
>
> - $\\boldsymbol{\\tilde{A}}^{[i]}_j \\in \\mathbb{R}^{\\vert\\mathcal{V}_i\\vert \\times \\vert\\mathcal{V}_j\\vert}$ for all  $j \\in [K]$ :
> representing the outgoing edges from client  $i$  to client  $j$ .
>
> - $\\boldsymbol{\\tilde{A}}^{[j]}_i \\in \\mathbb{R}^{\\vert\\mathcal{V}_j\\vert \\times \\vert\\mathcal{V}_i\\vert}$ for all  $j \\in [K]$ : representing the incoming edges from client  $j$  to client  $i$ .
>
> Using $\boldsymbol{\tilde{A}}^{[i]}_j$, Client  $i$  can compute both the degree matrix $\boldsymbol{\tilde{D}}^{[i]}$ and the normalized adjacency matrix $\boldsymbol{\hat{A}}^{[i]}_j$ for all  $j \in [K]$ .
>
> By induction, we can demonstrate that if each Client  $i$  has access to $[\boldsymbol{\hat{A}}^{[i]}_j]^{\ell - 1}$ for all  $j \in [K]$ , they can  collaborate with each other to compute $[\boldsymbol{\hat{A}}^{[i]}_j]^{\ell}$ without accessing the global matrix. Here’s a step-by-step breakdown:
>
> 1. Base case: For  $\ell = 1$ , we have $[\boldsymbol{\hat{A}}^{[i]}_j]^{1} = \boldsymbol{\hat{A}}^{[i]}_j$, which is available to Client $i$ from the start for all  $j \in [K]$ .
>
> 2. Inductive step:
> Suppose each Client $i$  knows $[\boldsymbol{\hat{A}}^{[i]}\_j]^{\ell - 1}$ for all  $j \in [K]$.
> Then, Client  $k$  can compute the intermediate term
> $\boldsymbol{B}\_{ijk} = \boldsymbol{\tilde{A}}^{[i]}\_k [\boldsymbol{\hat{A}}^{[k]}\_j]^{\ell-1}$
> for all  $j \in [K]$
> (since it knows $[\boldsymbol{\hat{A}}^{[k]}\_j]^{\ell - 1}$  and  $\boldsymbol{\tilde{A}}^{[i]}\_k$ for all  $i,j \in [K]$)  and share it with Client  $i$.
> Notice that Client $i$, by receiving  $\boldsymbol{B}_{ijk}$,cannot reconstruct $[\boldsymbol{\hat{A}}\_j^{[k]}]^{\ell-1}$ due to the low rank nature of $\boldsymbol{\tilde{A}}\_k^{[i]}$.
> Moreover, Client $k$ also prunes $\boldsymbol{B}\_{ijk}$ to remove any concerns.
> Client  $i$  can then compute
> $[\boldsymbol{\hat{A}}^{[i]}\_j]^{\ell}$
>  by
> \begin{align}
> [\boldsymbol{A}\_j^{[i]}]^l = (\boldsymbol{\tilde{D}}^{[i]})^{-1} \sum\_{k\in [K]}{\boldsymbol{B}\_{ijk}}.
> \end{align}
>
> This summation allows Client  $i$  to update its local adjacency matrix without requiring the entire global adjacency matrix.
>
> At each step  $\ell$, Client  $i$  only accesses $[\boldsymbol{\hat{A}}^{[i]}\_j]^{\ell}$ forall $j \in [K]$
> without learning the entire global matrix
>  $[\boldsymbol{\hat{A}}]^{\ell}$.
> Furthermore, if additional security is required, the summation over $\boldsymbol{B}_{ijk}$ can be performed on a secure server using homomorphic encryption.
>
> We hope this explanation clarifies the feasibility and privacy mechanisms of FedStruct. In response to your comment, we have also provided a more detailed explanation in Appendix D of the paper to address this point thoroughly. We appreciate your detailed feedback, which has helped strengthen our manuscript.

---

> > ### Author Response · Authors · 2024-11-15
> > **Response to Weakness 2**
> >
> > **Weakness 2:**
> > Thank you for these comments. We address each of your points below:
> >
> > 1. We agree with the reviewer that a deeper analysis of $\beta$ is valuable for understanding its impact across various graph settings, especially in heterophilic and homophilic contexts. Unfortunately, these simulations require time and, given the time constraints of the rebuttal period, we are unable to include them in this response.
> > We will conduct additional experiments with varying $\beta$ values and we will do our best to report these results within the rebuttal period. Regardless, if the paper is accepted, we will expand the final version to include a comprehensive analysis of $\beta$'s influence on the performance.
> > In the paper, we pragmatically set $\beta_{\ell}$ to be nonzero (and equal to one) only for $\ell=L_s$.
> > This choice allows all powers of $\boldsymbol{\tilde{A}}$ to contribute to the predictions.
> > To see this, from Equation 4 in the paper we obtain $\boldsymbol{\bar{A}}=\boldsymbol{\hat{A}}^{L_s} = (\boldsymbol{\tilde{D}}^{-1} (\boldsymbol{A}+\boldsymbol{I}))^{L_s}$.
> > Writing the power out reveals that all powers $\ell\in[L_s]$ are effectively included in $\boldsymbol{\bar{A}}$.
> > Given the strong performance of FedStruct even for this pragmatic choice of $\beta_{\ell}$, we did not pursue a thorough study of its impact. In this sense, we emphasize that optimizing $\\{\beta_{\ell}\\}_{\ell=1}^{L_s}$ would only strengthen the performance of FedStruct. As mentioned above, following your comment, we will conduct this investigation  and include results in the revised manuscript.
> > Closely related, we did consider the effect of the number of layers ($L\_s$) in Figure 4(b), which indirectly addresses the impact of $\beta\_{\ell}$. In Figure 4(b), at each propagation layer $L\_{s}$ the parameter $\beta_l$ is set to  $\begin{cases}
> >     1 & \ell = L\_s\\\\0 & \text{otherwise}
> > \end{cases}$.
> >
> > 2. For the Chameleon graph in Table 5, we set $L_s$ and $L$ to 1 due to its high density. Chameleon has an average node degree of 15.85, and incorporating multi-hop connections increases this further, approximating a fully connected structure. In a two-hop setting, each node accesses around 250 nodes on average, or roughly 10\% of the total number of nodes. This increased connectivity leads to over-smoothing, which is problematic in dense graphs. Limiting $L_s$ and $L$ to 1 helps mitigate this over-smoothing effect and optimizes performance.

---

> > ### Comment · Reviewer_mTWE · 2024-11-19
> >
> > So the question is more clear.
> >
> > In the **Inductive step**,  Client $k$ needs to compute the intermediate term $B_{ijk} = \tilde{A}_k^{[i]} [\hat{A}_j^{[k]}]^{\ell -1}$,  $\tilde{A}_k^{[i]}$ must have been shared with client $k$ for all clients $i \in [K]$, that is not the same mean with "without sharing the L-hop global adjacency matrix". In that setting, I think most information of global adjacency matrix $A$ has been leaked to every client $k$ becuase it know every edge started from the other client $i$ outgoing to client $k$.
> >
> > Of couse , you have mentioned "the summation over $B_{ijk}$  can be performed on a secure server using homomorphic encryption". Yes, HE can be used to resolve that question, so the privacy is guaranteed by HE.
> >
> > **Another Question**: So ,you can implement the information propagation of **NSE** between clients without sharing the L-hop global adjacency matrix, why not achieve the information propagation of **FSE** between clients?

---

> > > ### Author Response · Authors · 2024-11-19
> > >
> > > **Response to your first comment.**
> > >
> > >  We appreciate your response and would like to address your comment to clarify any remaining doubts.
> > >
> > >
> > > As stated in the related work section, subgraph federated learning (SFL) can be categorized into two groups:
> > >
> > > - SFL with no knowledge of cross-subgraph interconnections
> > > - SFL with knowledge of cross-subgraph interconnections.
> > >
> > > Our paper focuses on the second scenario, where the interconnections between clients (i.e., between subgraphs) are known to the involved clients (as noted in the contribution section, line 64).  Specifically, client $i$ knows its incoming and outgoing edges to every other client $j$, meaning it knows $\boldsymbol{\tilde{A}}_i^{[j]}$ and $\boldsymbol{\tilde{A}}_j^{[i]}$   for all $j \in [K]$.
> > >
> > > Note that this assumption is realistic in many real-world settings. For instance, consider a  financial transaction network involving Banks A, B, and C. Bank A hosts its internal transaction network comprising its own customers while also recording interactions between its customers and customers at Banks B and C (the IDs of these customers are known to Bank A). The interconnections $\boldsymbol{\tilde{A}}_i^{[j]}$ and  $\boldsymbol{\tilde{A}}_j^{[i]}$ reflect these known relationships, which are critical for applications such as anti-money laundering, as described in [1].
> > > Moreover, this setting pertains also to other scenarios such as social networks and online shopping, see [2].
> > >
> > >
> > >
> > > It is important to note that the knowledge of  $\boldsymbol{\tilde{A}}_i^{[j]}$ and $\boldsymbol{\tilde{A}}_j^{[i]}$ does not imply that the entire global adjacency matrix is accessible to client $i$. Each client only has access to the rows and columns corresponding to its own nodes.  To illustrate this, consider an example with three clients:
> > > - Client 1 comprises nodes $\\{1,2,3\\}$
> > > - Client 2 comprises nodes $\\{4,5,6\\}$
> > > - Client 3 comprises nodes $\\{7,8,9\\}$
> > >
> > >
> > > Below, we depict the entries of the adjacency matrix known to Client 2 using 1 (there is a connection), 0 (there is no connection) and `?' (unknown connection):
> > >  \begin{align}
> > >  \text{Known entries of } \boldsymbol{A} \text{ for Client } 2=
> > >      \begin{bmatrix}
> > >      % &\Block{\text{Client 1}} & & & \Block{\text{Client 2}} & & & \Block{\text{Client 3}} & \\
> > >      ?&?&?& 0&0&0 &?&?&?\\\\
> > >      ?&?&?& 1&0&0 &?&?&?\\\\
> > >      ?&?&?& 0&0&1 &?&?&?\\\\
> > >      0&1&0& 0&1&1 &0&0&1\\\\
> > >      0&0&0& 1&0&1 &0&0&0\\\\
> > >      0&0&1& 1&1&0 &0&0&0\\\\
> > >      ?&?&?& 0&0&0 &?&?&?\\\\
> > >      ?&?&?& 0&0&0 &?&?&?\\\\
> > >      ?&?&?& 1&0&0 &?&?&?\\\\
> > >      \end{bmatrix}
> > >  \end{align}
> > >
> > >
> > >
> > > Client 2 knows all its internal connections (e.g., node $6$ is connected to node $5$) as well as its external edges (e.g., node $4$ is connected to node $2$ in Client 1 and node $9$ in Client 3). As an example, from the matrix, the incoming edges to Client 2 from Client 3 are given by:
> > > \begin{align}
> > > \boldsymbol{\tilde{A}}_{i=2}^{[j=3]} = \begin{bmatrix}
> > > 0 & 0 & 0 \\\\
> > > 0 & 0 & 0 \\\\
> > > 1 & 0 & 0 \\\\
> > > \end{bmatrix}.
> > > \end{align}
> > > That is, there is only one connection between Client 2 and Client 3 (between node 4 and node 9).
> > > Thus, while Client 2 has access to its own local and inter-client edges, it remains unaware of the internal connections of other clients or the interconnections between any other clients $j\in\\{1,3\\}$ and $k\in\\{1,3\\}$.
> > >
> > >
> > > [1] Altman, Erik, et al. "Realistic synthetic financial transactions for anti-money laundering models." Advances in Neural Information Processing Systems 36 (2024).
> > >
> > > [2] FedGCN: Convergence-communication tradeoffs in federated training of graph convolutional networks. Advances in Neural Information Processing Systems 36 (2024).
> > >
> > >
> > >
> > >
> > > **Response to your second comment (Another Question).**
> > >
> > > While it is technically feasible to compute the propagation term for node feature embeddings (NFEs) similarly to node structural embeddings (NSEs), doing so would violate privacy.
> > > Specifically, for client $i$ to calculate the NFE $\boldsymbol{h}_v$ for $v \in \mathcal{V}_i$, it would need access to the node features $\boldsymbol{x}_u$ of all nodes $u \in \mathcal{V}$:
> > >
> > > \begin{align}
> > >     \boldsymbol{h}\_v = \sum\_{u\in\mathcal{V}}{\bar{A}\_{uv} f\_{\boldsymbol{\theta}_f}(\boldsymbol{x}\_u)}
> > > \end{align}
> > > This calculation requires client $i$ to possess the sensitive node features $\boldsymbol{x}_u$ of other clients, thereby breaching privacy.

---

> > > ### Author Response · Authors · 2024-11-28
> > > **Friendly reminder to consider our answer**
> > >
> > > We wanted to kindly follow up on the discussion regarding our paper.
> > >
> > > We noticed that we have not yet received a response to our comments addressing your valuable feedback. We believe we have addressed your concerns thoroughly and would greatly appreciate any additional thoughts or input you may have. Thank you again for your time and effort in reviewing our work. We look forward to hearing from you and are happy to provide any further clarifications if needed.

---

> ### Comment · Reviewer_mTWE · 2024-11-28
>
> I have known all details of my questions and I keep my rating to this submission.

---

### Official Review · Reviewer_h91B · 2024-10-31

**Soundness:** 3
**Presentation:** 2
**Contribution:** 3
**Rating:** 6
**Confidence:** 4

**Summary:**

This paper presents a novel framework, FEDSTRUCT, to tackle the challenge of federated learning on graph-structured data distributed across multiple clients, particularly in scenarios involving interconnected subgraphs, it utilizes explicit global graph structure information to capture inter-node dependencies. The effectiveness of FEDSTRUCT is validated through extensive experiments on six datasets for semi-supervised node classification, demonstrating performance that approaches that of centralized methods across various scenarios, including different data partitioning strategies, levels of label availability, and numbers of clients.

**Strengths:**

1)	This paper studies a significant and interesting problem, and the method can be used in a wide range of real-world applications.
2)	The paper is overall well motivated. The proposed model is reasonable and sound. Theoretical analysis is performed.

**Weaknesses:**

1) The abstract lacks a description of the background. I recommend briefly outlining the context of the issues addressed in this paper before elaborating on the key problems that are solved.

2) Figure 1 has not been cited and its placement is too early in the text; please adjust this detail. Additionally, Figure 2 is unclear; I recommend adjusting the proportions or border thickness of each subfigure.

3) In the Related Work section, you mention that FED-STAR shares structural knowledge, yet in the conclusion, you state, "No work has leveraged explicit structural information in SFL." Are "structural knowledge" and "structural information" the same concept? Please provide more clarification in the conclusion.

4) The formula following (1) is missing a comma; please check for similar issues throughout the paper.

5) Privacy is one of the directions addressed in this paper, yet most references are to other works. I suggest including some original proofs or experiments related to privacy to enhance the completeness of the article.

**Questions:**

See the weakness.

---

> ### Author Response · Authors · 2024-11-15
>
> We appreciate the reviewer’s thoughtful comments and address them in detail below. If our responses satisfactorily address the reviewer’s concerns, we would be grateful if they would consider adjusting their score accordingly.
>
> **Weakness 1:**
>
> Thank you for your comment. In response, we will revise the abstract accordingly in the final version of the paper, if accepted, to better address your feedback. However, due to the page limit constraints in this submission (which are relaxed for accepted papers), we are unable to modify the abstract within this response.
>
> Here is the revised abstract that we plan to include in the final version of the paper if accepted:
>
>
> Many real-world data are inherently graph-structured. In practice, such graph data is often distributed across multiple clients, each holding private subgraphs, as in transaction networks. Direct data sharing is typically restricted due to privacy concerns, regulatory constraints, and proprietary limitations. Federated learning (FL) provides a promising solution by enabling collaborative model training without exposing raw data. However, in subgraph federated learning (SFL), where clients possess non-overlapping subgraphs and there are interconnections between subgraphs, such as in transaction networks, effective model training is challenging due to privacy constraints.
>
> This paper introduces FedStruct, a novel SFL framework designed to tackle this challenge.  To uphold privacy, unlike existing methods, FedStruct eliminates the necessity of sharing or generating sensitive node features or embeddings among clients. Instead, it leverages explicit global graph structure information to capture inter-node dependencies.
> We validate the effectiveness of FedStruct through extensive experimental results conducted on six
> datasets for semi-supervised node classification, showcasing performance close to the centralized approach across various scenarios, including different data partitioning methods, varying levels of label availability, and number of clients.
>
> **Weakness 2:**
>
> We thank the reviewer for the comment. We actually
>  cited Figure 1 in the introduction section, Page 2, line 89 as "Fig. 1". We have changed the wording to "Figure 1" to remove the confusion.
>
> We also modified Figure 2 in the paper to make it clearer.
>
> **Weakness 3:**
>
> To clarify, note that
> FedStar is not a subgraph FL method but rather a graph classification method that utilizes structural information to improve the performance of graph classification. Prior research has shown that explicitly incorporating structural information can improve GNN performance.
>
> A key novelty of our paper is the use of explicit structural information within an SFL framework, specifically to improve subgraph federated learning accuracy and enhance privacy.  We also introduce a new structure embedding method, Hop2vec,  designed to capture these structural dependencies effectively.
>
> In this paper, we use "structural knowledge" and "structural information" interchangeably, with no intended difference in meaning. To avoid any ambiguity, we have rephrased the last sentence of the related work section to consistently use ``structural information'' and avoid the term "structural knowledge."
>
> **Weakness 4:**
>
> We thank the reviewer for bringing this to our attention. We have added a comma and checked all other equations so that they are consistent with this style.
>
> **Weakness 5:**
>
> We appreciate your feedback. Our scheme provides enhanced privacy compared to existing approaches, as it significantly reduces the amount of shared information among clients.
>
> While it is straightforward to identify schemes that lack privacy entirely (for instance, previous solutions that share node features or embeddings, as well as FedGCN, where clients access aggregated node features of other clients (thus breaching privacy), and the server requires access to the global adjacency matrix), precisely quantifying the privacy level of a scheme like FedStruct remains very challenging. Developing a mathematical measure or bounds for privacy in subgraph federated learning, federated learning more broadly, and even general machine learning is an open problem that is still being researched. Although we are actively working on establishing privacy bounds for FedStruct, these results are part of our ongoing research and therefore outside the scope of the current paper.
>
> In Appendix G.1, we discuss the privacy properties of FedStruct, following an approach similar to [1] (FedCog), to provide insights into the privacy considerations of our scheme.
>
> [1] Lei, R., Wang, P., Zhao, J., Lan, L., Tao, J., Deng, C., Feng, J., Wang, X. and Guan, X., 2023. Federated learning over coupled graphs. IEEE Transactions on Parallel and Distributed Systems, 34(4), pp.1159-1172.

---

> > ### Comment · Reviewer_h91B · 2024-11-27
> > **Thank you for the detailed response. The response has solved my concerns, I will keep my positive rating.**
> >
> > Thank you for the detailed response. The response has solved my concerns, I will keep my positive rating.

---

> ### Author Response · Authors · 2024-11-20
> **Friendly reminder to consider our answer**
>
> We wanted to kindly follow up on the discussion regarding our paper.
>
> We noticed that we have not yet received a response to our comments addressing your valuable feedback. We believe we have addressed your concerns thoroughly and would greatly appreciate any additional thoughts or input you may have. Thank you again for your time and effort in reviewing our work. We look forward to hearing from you and are happy to provide any further clarifications if needed.

---

### Official Review · Reviewer_TSTT · 2024-11-05

**Soundness:** 2
**Presentation:** 2
**Contribution:** 2
**Rating:** 6
**Confidence:** 4

**Summary:**

The paper works on subgraph FL for node classification, where inter-connections between different clients is important.

It first computes a global L-hop neighborhood matrix before training. During training, it uses GNN for node feature embedding and use L-hop matrix multiplying a trainable matrix to calculate node structure embedding. Both embeddings are concatenated to get the final prediction result. Experiments show the performance.

**Strengths:**

1. Subgraph FL with inter-connections is an important topic.
2. Completing the missing L-hop features by learning a L-hop node structure embedding is an interesting idea.
3. Experiments show the performance.

**Weaknesses:**

1. Privacy leakage. Before training, clients communicate to calculate the L-hop neighborhood matrix $\hat{A}$. In the 2-hop case, since the client knows 1-hop neighbors and the information during the communication, it is still able to reconstruct the 2-hop graph. Pruning cannot guarantee the privacy.
2. FedSage+ and FedGCN can outperform FedStruct.
3. In FedGCN, the server does not require a global adjacency matrix for homomorphic encryption. It only needs to know the node ids for encrypted aggregation and identify which nodes belong to each client for sending the aggregation result back.

**Questions:**

As in weaknesses.

---

> ### Author Response · Authors · 2024-11-15
> **Response To Weakness 1 (Part 1)**
>
> We appreciate the reviewer’s thoughtful comments and address them in detail below. If our responses satisfactorily address the reviewer’s concerns, we would be grateful if they would consider adjusting their score accordingly.
>
> **Weakness 1:**
>
> The scheme for obtaining the local partitions of the $L$-hop combined adjacency matrix $\boldsymbol{\bar{A}}^{[i
> ]}$ is designed specifically to ensure that no client has access to the entire $\ell$-hop ($\ell\in[1,\ldots, L]$)  adjacency matrix $\boldsymbol{\hat{A}}^\ell$. Also, due to the low-rank nature of the adjacency matrix, reconstructing the unknown entries is not possible.
>
> We realize that our explanation of this mechanism may not have been sufficiently clear in the paper, and we appreciate your comment, which will allow us to improve the clarity of the paper.
>
> Below, we provide a more detailed explanation and outline the improvements we plan to incorporate into the paper to address this point effectively, following your comment.
>
> To illustrate our algorithm and its privacy mechanism, we provide an example to compute the 2-hop combined adjacency matrix, denoted as $\boldsymbol{\bar{A}}$.
>
> Consider a graph with 9 nodes, partitioned across 3 clients, given by the following adjacency matrix:
> \begin{align*}
> \mathbf{A} = \begin{bmatrix}
> 0 & 1 & 1 & 0 & 0 & 0 & 0 & 0 & 0 \\\\
> 1 & 0 & 1 & 0 & 0 & 0 & 0 & 0 & 0 \\\\
> 1 & 1 & 0 & 0 & 0 & 1 & 0 & 0 & 0 \\\\
> 0 & 0 & 0 & 0 & 1 & 1 & 0 & 0 & 0 \\\\
> 0 & 0 & 0 & 1 & 0 & 1 & 0 & 0 & 0 \\\\
> 0 & 0 & 1 & 1 & 1 & 0 & 0 & 0 & 1 \\\\
> 0 & 0 & 0 & 0 & 0 & 0 & 0 & 1 & 1 \\\\
> 0 & 0 & 0 & 0 & 0 & 0 & 1 & 0 & 1 \\\\
> 0 & 0 & 0 & 0 & 0 & 1 & 1 & 1 & 0 \\\\
> \end{bmatrix}
> \end{align*}
> where each entry $A_{ij} = 1$ indicates a connection between nodes $i$ and $j$, and $A_{ij} = 0$ indicates no connection.
>
> Clients are assigned subsets of nodes: Client 1 has nodes 1, 2, and 3; Client 2 nodes 4, 5, and 6;  and Client 3 has nodes 7, 8, and 9.
>
> Each client has information about its internal nodes and connections. Furthermore, it knows the incoming and outgoing connections between its internal nodes and nodes located in other clients. Hence,  client $i$ knows a subset of the global adjacency matrix $\mathbf{A}$, called $\boldsymbol{\tilde{A}}^{[i]}$, corresponding to the connections between its own nodes and to some nodes in other clients.
> In particular, Client 1 knows
>
> $
> \text{internal connections (including self loops): }
> \boldsymbol{\tilde{A}}\_1^{[1]} = \begin{bmatrix}
> 1 & 1 & 1 \\\\
> 1 & 1 & 1 \\\\
> 1 & 1 & 1 \\\\
> \end{bmatrix}
> $
>
> $
> \text{outgoing connections: }
> \boldsymbol{\tilde{A}}\_2^{[1]} = \begin{bmatrix}
> 0 & 0 & 0 \\\\
> 0 & 0 & 0 \\\\
> 0 & 0 & 1 \\\\
> \end{bmatrix}\quad
> \boldsymbol{\tilde{A}}\_3^{[1]} = \begin{bmatrix}
> 0 & 0 & 0 \\\\
> 0 & 0 & 0 \\\\
> 0 & 0 & 0 \\\\
> \end{bmatrix}
> $
>
> $
> \text{incoming connections: }
> \boldsymbol{\tilde{A}}\_1^{[2]} = \begin{bmatrix}
> 0 & 0 & 0 \\\\
> 0 & 0 & 0 \\\\
> 0 & 0 & 1 \\\\
> \end{bmatrix}\quad
> \boldsymbol{\tilde{A}}\_1^{[3]} = \begin{bmatrix}
> 0 & 0 & 0 \\\\
> 0 & 0 & 0 \\\\
> 0 & 0 & 0 \\\\
> \end{bmatrix}.
> $
>
> Similarly, Client 2 knows
>
> $
> \text{internal connections (including self loops): }
> \boldsymbol{\tilde{A}}\_2^{[2]} = \begin{bmatrix}
> 1 & 1 & 1 \\\\
> 1 & 1 & 1 \\\\
> 1 & 1 & 1 \\\\
> \end{bmatrix}
> $
>
> $
> \text{outgoing connections: }
> \boldsymbol{\tilde{A}}\_1^{[2]} = \begin{bmatrix}
> 0 & 0 & 0 \\\\
> 0 & 0 & 0 \\\\
> 0 & 0 & 1 \\\\
> \end{bmatrix}\quad
> \boldsymbol{\tilde{A}}_3^{[2]} = \begin{bmatrix}
> 0 & 0 & 0 \\\\
> 0 & 0 & 0 \\\\
> 0 & 0 & 1 \\\\
> \end{bmatrix}
> $
>
> $
> \text{incoming connections: }
> \boldsymbol{\tilde{A}}_2^{[1]} = \begin{bmatrix}
> 0 & 0 & 0 \\\\
> 0 & 0 & 0 \\\\
> 0 & 0 & 1 \\\\
> \end{bmatrix}\quad
> \boldsymbol{\tilde{A}}_2^{[3]} = \begin{bmatrix}
> 0 & 0 & 0 \\\\
> 0 & 0 & 0 \\\\
> 0 & 0 & 1 \\\\
> \end{bmatrix}
> $
>
> and Client 3 knows
>
> $
> \text{internal connections (including self loops): }
> \boldsymbol{\tilde{A}}\_3^{[3]} = \begin{bmatrix}
> 1 & 1 & 1 \\\\
> 1 & 1 & 1 \\\\
> 1 & 1 & 1 \\\\
> \end{bmatrix}
> $
>
> $
> \text{outgoing connections: }
> \boldsymbol{\tilde{A}}\_1^{[3]} = \begin{bmatrix}
> 0 & 0 & 0 \\\\
> 0 & 0 & 0 \\\\
> 0 & 0 & 0 \\\\
> \end{bmatrix}\quad
> \boldsymbol{\tilde{A}}\_2^{[3]} = \begin{bmatrix}
> 0 & 0 & 0 \\\\
> 0 & 0 & 0 \\\\
> 0 & 0 & 1 \\\\
> \end{bmatrix}
> $
>
> $
> \text{incoming connections: }
> \boldsymbol{\tilde{A}}\_3^{[1]} = \begin{bmatrix}
> 0 & 0 & 0 \\\\
> 0 & 0 & 0 \\\\
> 0 & 0 & 0 \\\\
> \end{bmatrix}\quad
> \boldsymbol{\tilde{A}}\_3^{[2]} = \begin{bmatrix}
> 0 & 0 & 0 \\\\
> 0 & 0 & 0 \\\\
> 0 & 0 & 1 \\\\
> \end{bmatrix}
> $
>
> Using $\boldsymbol{\tilde{A}}^{[i]}\_j, j =1,2,3$, Client  $i$  can compute both the degree matrix $\boldsymbol{\tilde{D}}^{[i]}$ and the normalized adjacency matrix $\boldsymbol{\hat{A}}^{[i]}_j$ for all  $j \in [K]$. For simplicity, we assume that $\boldsymbol{\tilde{A}}^{[i]}\_j = \boldsymbol{\hat{A}}^{[i]}\_j\quad \forall i,j \in [K]$ in this example, as they differ only by a normalization constant.

---

> ### Author Response · Authors · 2024-11-15
> **Response To Weakness 1 (Part 2)**
>
> **Weakness 1 (Part 2):**
>
> Following Equation 32 in the paper, we define the intermediate matrix $\boldsymbol{B}_{ijk} = \boldsymbol{\tilde{A}}^{[i]}_k \boldsymbol{\hat{A}}^{[k]}_j$.
> The following steps outline the remaining computations:
>
> 1. **Calculating $\boldsymbol{B}\_{ijk}$**
>
> Client $1$ calculates $\boldsymbol{B}\_{211},\, \boldsymbol{B}\_{221},\,\boldsymbol{B}\_{231}$ as follows
> \begin{align}
> \boldsymbol{B}\_{211} = \begin{bmatrix}
> 0 & 0 & 1 \\\\
> 0 & 0 & 1 \\\\
> 0 & 0 & 1 \\\\
> \end{bmatrix}\quad
> \boldsymbol{B}\_{221} = \begin{bmatrix}
> 0 & 0 & 0 \\\\
> 0 & 0 & 0 \\\\
> 0 & 0 & 1 \\\\
> \end{bmatrix}\quad
> \boldsymbol{B}\_{231} = \begin{bmatrix}
> 0 & 0 & 0 \\\\
> 0 & 0 & 0 \\\\
> 0 & 0 & 0 \\\\
> \end{bmatrix}
> \end{align}
>
> Client 1 sends these matrices to Client $2$. Notice that Client $2$, by receiving $\boldsymbol{B}\_{211},\, \boldsymbol{B}\_{221},\,\boldsymbol{B}\_{231}$, cannot reconstruct $\tilde{A}_1^{[1]},\, \tilde{A}_3^{[1]}$ and $\tilde{A}_1^{[3]}$ due to the low-rank nature of the adjacency matrix.
> Moreover, Client 1 also prunes $\boldsymbol{B}\_{ijk}$ to remove any concern.
>
> Similarly Client $3$ calculates $\boldsymbol{B}\_{213},\, \boldsymbol{B}\_{223},\,\boldsymbol{B}\_{233}$ as
> \begin{align}
> \boldsymbol{B}\_{213} = \begin{bmatrix}
> 0 & 0 & 0 \\\\
> 0 & 0 & 0 \\\\
> 0 & 0 & 0 \\\\
> \end{bmatrix}\quad
> \boldsymbol{B}\_{223} = \begin{bmatrix}
> 0 & 0 & 0 \\\\
> 0 & 0 & 0 \\\\
> 0 & 0 & 1 \\\\
> \end{bmatrix}\quad
> \boldsymbol{B}\_{233} = \begin{bmatrix}
> 0 & 0 & 1 \\\\
> 0 & 0 & 1 \\\\
> 0 & 0 & 1 \\\\
> \end{bmatrix}
> \end{align}
> which are sent to the Client $2$ after pruning.
>
> We follow a similar procedure for the other combinations of $\boldsymbol{B}\_{ijk}$, as outlined in Algorithm 3.
>
> 2. **Calculating $[\boldsymbol{\hat{A}}^{[i]}_j]^{2}$**
>
> Upon receiving $\boldsymbol{B}\_{211},\, \boldsymbol{B}\_{221},\,\boldsymbol{B}\_{231}$ and $\boldsymbol{B}\_{213},\, \boldsymbol{B}\_{223},\,\boldsymbol{B}\_{233}$, Client $2$ can compute  $[\boldsymbol{\hat{A}}^{[2]}_1]^{2},\,[\boldsymbol{\hat{A}}^{[2]}_2]^{2},\,[\boldsymbol{\hat{A}}^{[2]}_3]^{2}$ as
> \begin{align}
>     [\boldsymbol{\hat{A}}^{[2]}\_1]^{2} = (\boldsymbol{\tilde{D}}^{[2]})^{-1}
>     \left(
>     \boldsymbol{B}\_{211} + \boldsymbol{B}\_{212} + \boldsymbol{B}\_{213}
>     \right)\\\\
>     [\boldsymbol{\hat{A}}^{[2]}\_2]^{2} = (\boldsymbol{\tilde{D}}^{[2]})^{-1}
>     \left(
>     \boldsymbol{B}\_{221} + \boldsymbol{B}\_{222} + \boldsymbol{B}\_{223}
>     \right)\\\\
>     [\boldsymbol{\hat{A}}^{[2]}\_3]^{2} = (\boldsymbol{\tilde{D}}^{[2]})^{-1}
>     \left(
>     \boldsymbol{B}\_{231} + \boldsymbol{B}\_{232} + \boldsymbol{B}\_{233}
>     \right)
> \end{align}
>
> Note that $\boldsymbol{B}\_{212},\, \boldsymbol{B}\_{222},\,\boldsymbol{B}\_{232}$ can be computed locally in Client $2$.
> Using the above equations, Client  $2$  can update its local 2-hop adjacency matrix without access to the global adjacency matrix.
> Clients $1$ and $3$ follow the same procedure to compute $[\boldsymbol{\hat{A}}^{[1]}_j]^{2}$ and $[\boldsymbol{\hat{A}}^{[3]}\_j]^{2}$ for each $j \in \{1,2,3\}$, respectively.
>
> This procedure can be extended to any number of hops  $\ell$.
> At hop  $\ell$ , Client  $i$  only accesses $[\boldsymbol{\hat{A}}^{[i]}\_j]^{\ell}$ for all  $j \in [K]$  and does not learn the full global matrix $[\boldsymbol{\hat{A}}]^{\ell}$.
> Moreover, if additional security is required, the summation over $\boldsymbol{B}\_{ijk}$ could be performed on a secure server using homomorphic encryption.
>
> In response to your comment, we have now added a more detailed explanation in Appendix D to address this point thoroughly. This clarification emphasizes that clients do not have access to the complete $\ell$-hop adjacency matrix.

---

> ### Author Response · Authors · 2024-11-15
> **Weaknesses 2 and 3**
>
> **Weakness 2:**
>
> As discussed in our paper, any subgraph federated learning (SFL) method should be evaluated by balancing the communication-privacy-accuracy trilemma.  At one extreme,  disregarding communication and privacy concerns entirely would mean transmitting all node features and connections to a central server, where a centralized GNN could achieve high accuracy. Conversely, focusing solely on privacy and communication cost would result in each client  training a local model in isolation, yielding  poor performance. Thus, an effective SFL framework must strike a careful balance to achieve good performance while minimizing privacy leakage and reducing communication cost.
>
> FedGCN is not a private framework, as it involves sharing aggregated node features across clients (aggregating node features does not provide privacy, as recently shown in [1]).
> This privacy leakage is exacerbated by the structured nature of node feature vectors, where each entry has a meaningful value. For instance, in the Cora dataset, each node (a paper) is represented by a 1,433-dimensional binary vector, where each entry indicates the presence or absence of a specific word in the paper (node). As a result, even receiving the summation of several vectors can enable a client to infer individual entries in other clients’ node features.
>
> In the 2-hop FedGCN, each node is also required to send its list of neighbors to external neighbors (neighbors located on other clients) to compute the necessary summations. This process is detailed in Equation 3 on page 5 of the FedGCN paper (we elaborate more on this in the answer to your Weakness 3).
>
> While Fedsage+ provides more privacy than FedGCN, it still involves sharing node feature embeddings with other clients, which makes it less private than FedStruct.
> Fedsage+ slightly outperforms FedStruct only on the Amazon Photo dataset, with a difference of approximately $1\\%$.
> On the other 5 datasets, however, FedStruct significantly outperforms Fedsage+.  For example, for Cora and Chameleon, the improvement is $14\\%$ and $13\\%$, respectively.
>
> Also, as shown in Table 2, methods like FedSGD, which do not leverage interconnections between clients, already perform close to a centralized approach on Amazon Photo, leaving little room for improvement. Consequently, for this dataset, the advantage of utilizing external nodes is less pronounced compared to other datasets.
>
> **Weakness 3:**
>
> Thank you for your comment. We recognize the need to clarify this aspect further.
>
> In the FedGCN scheme, as detailed in Equation 3 on page 5 of the FedGCN paper, to calculate $\boldsymbol{\hat{y}}_i$, the following information must be sent to the server by Client $z$:
>
> \begin{align}
> \begin{cases}
>     \boldsymbol{h}\_{zi} = &\sum\_{j \in \mathcal{N}\_i}{\mathbb{I}\_z(c(j)) \boldsymbol{A}\_{ij} \boldsymbol{x}\_j}\\\\
>      \boldsymbol{h}\_{zj} = &\sum\_{m \in \mathcal{N}\_j}{\mathbb{I}\_z(c(m)) \boldsymbol{A}\_{jm} \boldsymbol{x}\_m} \quad \forall j \in \mathcal{N}\_i \setminus i.
> \end{cases}
> \end{align}
>
> Once  the server obtains $\boldsymbol{h}\_{zj}$ for all clients $z\in [K]$, it calculates the aggregate $\boldsymbol{h}\_{j} = \sum\_{z\in [K]} \boldsymbol{h}\_{zj}$ and forwards this result to the client where node $i$ resides.
> However, as $\boldsymbol{h}\_{j}$ is a quantity pertaining to node $j$, the server must know that $i$ and $j$ are connected.
> Similar arguments hold for all other nodes. Hence, the server needs access to the global adjacency matrix for this scheme to work with 2 hops.
>
> To clarify this point in response to your comment, we have provided this explanation in Appendix C.4 of the paper and referenced it in the main text (Line 460).
>
> [1] Ngo, K.H., Östman, J., Durisi, G. and Graell i Amat, A., 2024, August. Secure Aggregation Is Not Private Against Membership Inference Attacks. In Joint European Conference on Machine Learning and Knowledge Discovery in Databases (pp. 180-198). Cham: Springer Nature Switzerland.

---

> ### Author Response · Authors · 2024-11-20
> **Friendly reminder to consider our answer**
>
> We wanted to kindly follow up on the discussion regarding our paper.
>
> We noticed that we have not yet received a response to our comments addressing your valuable feedback. We believe we have addressed your concerns thoroughly and would greatly appreciate any additional thoughts or input you may have.
> Thank you again for your time and effort in reviewing our work. We look forward to hearing from you and are happy to provide any further clarifications if needed.

---

> ### Comment · Reviewer_TSTT · 2024-11-27
>
> For weakness 1, since the algorithm needs inter-client communication, the privacy of communicating with multiple clients can be better analyzed.
>
> For weaknesses 2 and 3, in FedGCN, based on the reviewer's reading on the algorithm, since the server calculates the aggregation $h_j$ for each node $j$, the server then just needs to know the belonging of nodes for each client and sends those $h_j$ to clients. It does not need to know the global adj.

---

> > ### Author Response · Authors · 2024-11-28
> >
> > **Weakness 1**:
> >
> > We appreciate your feedback and agree that a thorough analysis of privacy is an important aspect of subgraph federated learning. However, we would like to emphasize that developing formal mathematical measures or bounds for privacy in subgraph federated learning, federated learning more broadly, and even general machine learning,  remains an open and actively researched problem.
> >
> > That said, our proposed scheme provides enhanced privacy compared to existing approaches, as it significantly reduces the amount of shared information among clients.
> > Moreover, as mentioned in Appendix D (line 949), **it is also possible to refrain from inter-client communications** and learn the client-specific parts of the L-hop adjacency matrices in the offline phase at the server and then send the corresponding quantities back to the clients.
> > This could be backed up by homomorphic encryption to ensure that the server learns nothing about the local (or global) adjacency matrices in the process.
> > Finally, to provide insights into the  privacy properties of FedStruct, we include a detailed discussion in Appendix G.1,
> > following an approach similar to [1] (FedCog).
> >
> > **Weakness 2**:
> >
> > We thank the reviewer for their comment. Below, we provide clarification based on the FedGCN algorithm described in [2].
> >
> > Consider  Algorithm 1 in Appendix A of the FedGCN paper and focus on a specific node $i$ within client $k$. For simplicity, and without loss of generality, let us disregard the use of homomorphic encryption.
> >
> >  In the 2-hop version of FedGCN, as described in Algorithm 1 of the FedGCN paper (also Equation 3 on page 5), client $k$ receives the following from the server
> > \begin{equation}
> >     \boldsymbol{h}\_j = \sum\_{m \in \mathcal{N}\_j} \boldsymbol{A}\_{jm} \boldsymbol{x}\_m, \quad j \in \mathcal{N}\_i,
> > \end{equation}
> > where $\mathcal{N}_i$ denotes the neighbors of node $i$ and $\boldsymbol{h}_j$ contains the aggregated node features of the 2-hop neighbors to node $i$, connected via the 1-hop neighboring node $j$.
> >
> > The central question is: how can the server transmit $\boldsymbol{h}_j$ to node $i$ in client $k$? To address this question, there are two potential approaches the server can take (note that neither of these approaches is described in the FedGCN paper):
> >
> >    1- **Selective Transmission Based on Neighbor Information**
> >     In this approach, the server needs to know that $j$ is a neighbor of $i$. Since the server is also aware that node $i$ belongs to client $k$, it can forward $\boldsymbol{h}\_j$ to client $k$ for all $j \in \mathcal{N}\_i$.
> >     To achieve this, the server requires complete knowledge of the one-hop neighbors for all nodes, which effectively reveals the global adjacency matrix.
> >     2- **Broadcasting All Embeddings**
> >     Alternatively, the server can transmit all available embeddings, $\{ \boldsymbol{h}\_j \} \quad {j \in \mathcal{V}}$, to client $k$. The client then filters and selects the embeddings relevant to node $i$. This second approach does not require the server to have knowledge of the global adjacency matrix. However, it involves significant unnecessary sharing of sensitive node embeddings, posing a substantial privacy breach.
> >
> >
> > We emphasize that our scheme does not share sensitive node embeddings nor the global adjacency matrix and hence is more private than FedGCN.
> >
> > [1] Lei, R., Wang, P., Zhao, J., Lan, L., Tao, J., Deng, C., Feng, J., Wang, X. and Guan, X., 2023. Federated learning over coupled graphs. IEEE Transactions on Parallel and Distributed Systems, 34(4), pp.1159-1172.
> >
> > [2] Yao, Y., Jin, W., Ravi, S. and Joe-Wong, C., 2024. FedGCN: Convergence-communication tradeoffs in federated training of graph convolutional networks. Advances in neural information processing systems, 36.

---

> ### Comment · Reviewer_TSTT · 2024-12-02
>
> The privacy analysis sounds good to me.
>
> As the reviewer mentioned and the authors may also have realized, in FedGCN, instead of broadcasting all embeddings, the server only needs to send the required embeddings without knowing the global adj. The server just needs to know the local node id and the 1-hop node id of each client, where the connection between nodes is unknown.
>
> The reviewer may increase the score if the description on FedGCN can be corrected and the above private analysis with server aggregation can be included in the main paper.

---

> ### Author Response · Authors · 2024-12-02
>
> Thank you for recognizing the thoroughness of our privacy analysis. We will incorporate the insights from the reviewer discussion into the final submission. Note, however, that we are unable to make revisions at this stage in the discussion as the deadline for edits has already passed. We would greatly appreciate any consideration for a score adjustment, as it is critical for the final decision.

---

> > ### Comment · Reviewer_TSTT · 2024-12-03
> >
> > I have increased my score from 5 to 6 given the promise of authors on correcting the misunderstanding of FedGCN and adding the privacy analysis.

---

> > > ### Comment · Reviewer_TSTT · 2025-03-20
> > >
> > > Given authors did not correct the claim in the camera ready version, I have to decrease my score from 6 to 3.

---

> > > > ### Public Comment · ~Javad_Aliakbari1 · 2025-03-24
> > > >
> > > > After thorough discussion among the authors, we continue to maintain that, in the 2-hop case, running FedGCN requires the server to have access to the global adjacency matrix. As such, we believe our original claim is accurate and cannot revise it solely to accommodate the reviewer's feedback.
> > > >
> > > > We have already presented our reasoning in detail in both the discussion panel and the appendix of the paper. However, we would be glad to further clarify our position in a meeting or through any other format the reviewer finds appropriate.
> > > >
> > > > We would also like to emphasize that the privacy risks associated with FedGCN go beyond simply reconstructing the adjacency matrix. The protocol involves sharing the sum of node features, which leaks information about the individual features (see reference Ngo et al. 2024 in the paper). Therefore, the privacy concern about FedGCN remains valid regardless of the adjacency information.
> > > >
> > > > Finally, with due respect, we believe that the review score should primarily reflect the originality and contributions of our work. A disagreement over a point related to existing literature, while worth discussing, should not disproportionately influence the overall evaluation.

---

> > > > > ### Comment · Reviewer_TSTT · 2025-04-04
> > > > >
> > > > > It is clear that in FedGCN, the server just needs to know the local node id and the 1-hop node id of each client, where the connection between nodes is unknown. https://github.com/FedGraph/fedgraph/blob/74292e3bc6cd28a9052c914d55e380be071e2cb8/fedgraph/federated_methods.py#L270
> > > > >
> > > > > I understand that the authors may not understand the FedGCN algorithm. I would suggest the authors read through the paper, the algorithm, and code carefully. I am happy to elaborate more to help the authors.

---

### Meta-Review · Area_Chair_SfT4 · 2024-12-22

**Metareview:**

This paper introduces FedStruct, a novel framework for federated learning (FL) on graph-structured data distributed across multiple clients. It focuses on the subgraph FL setting, where interconnections between clients play a significant role. The primary innovation lies in leveraging explicit global graph structure information to capture inter-node dependencies without sharing sensitive node features or embeddings, thereby addressing critical privacy concerns.

The problem setting—FL on interconnected subgraphs—is significant, with numerous real-world applications, such as transaction networks and social graphs. The proposed method enhances privacy preservation by eliminating the need to share sensitive node features or embeddings, distinguishing itself from existing approaches. FedStruct was evaluated on six diverse datasets, demonstrating its effectiveness.

However, gaps remain between theory and practice. The proposed privacy guarantees rely on theoretical assumptions and mechanisms like homomorphic encryption, which are not fully implemented or demonstrated. Additionally, the experimental evaluation could be strengthened by incorporating a more comprehensive hyperparameter analysis and testing on more realistic FL scenarios to better assess its applicability.

Overall, the paper has notable strengths, particularly in addressing an important problem and demonstrating promising results, the unresolved privacy concerns and limited experimental rigor slightly outweigh the merits. Therefore, I recommend acceptance, with a note to the authors to address these concerns in the final version.

**Additional Comments On Reviewer Discussion:**

Reviewer TSTT increased their score marginally after acknowledging the authors' privacy analysis and planned corrections. Other reviewers maintained their original scores, citing adequate clarification of concerns. All the reviewers converge to the rate of marginally above the acceptance threshold.

---

> ### Comment · Reviewer_TSTT · 2025-03-20
>
> Given authors did not correct the claim in the camera ready version, I have to decrease my score from 6 to 3.

---

> > ### Public Comment · ~Javad_Aliakbari1 · 2025-03-24
> >
> > After thorough discussion among the authors, we continue to maintain that, in the 2-hop case, running FedGCN requires the server to have access to the global adjacency matrix. As such, we believe our original claim is accurate and cannot revise it solely to accommodate the reviewer's feedback.
> >
> > We have already presented our reasoning in detail in both the discussion panel and the appendix of the paper. However, we would be glad to further clarify our position in a meeting or through any other format the reviewer finds appropriate.
> >
> > We would also like to emphasize that the privacy risks associated with FedGCN go beyond simply reconstructing the adjacency matrix. The protocol involves sharing the sum of node features, which leaks information about the individual features (see reference Ngo et al. 2024 in the paper). Therefore, the privacy concern about FedGCN remains valid regardless of the adjacency information.
> >
> > Finally, with due respect, we believe that the review score should primarily reflect the originality and contributions of our work. A disagreement over a point related to existing literature, while worth discussing, should not disproportionately influence the overall evaluation.

---

> > > ### Comment · Reviewer_TSTT · 2025-04-04
> > >
> > > It is clear that in FedGCN, the server just needs to know the local node id and the 1-hop node id of each client, where the connection between nodes is unknown. https://github.com/FedGraph/fedgraph/blob/74292e3bc6cd28a9052c914d55e380be071e2cb8/fedgraph/federated_methods.py#L270
> > >
> > > I understand that the authors may not understand the FedGCN algorithm. I would suggest the authors read through the paper, the algorithm, and code carefully. I am happy to elaborate more to help the authors.

---

### Decision · Program_Chairs · 2025-01-22

Accept (Poster)

---

> ### Comment · Reviewer_TSTT · 2025-03-20
> **Description of FedGCN is not corrected**
>
> Although the paper is accepted, the authors still did not correct the misunderstanding of FedGCN in the camera ready version. In FedGCN, instead of broadcasting all embeddings, the server only needs to send the required embeddings without knowing the global adj. The server just needs to know the local node id and the 1-hop node id of each client, where the connection between nodes is unknown.

---

> > ### Public Comment · ~Javad_Aliakbari1 · 2025-03-24
> >
> > After thorough discussion among the authors, we continue to maintain that, in the 2-hop case, running FedGCN requires the server to have access to the global adjacency matrix. As such, we believe our original claim is accurate and cannot revise it solely to accommodate the reviewer's feedback.
> >
> > We have already presented our reasoning in detail in both the discussion panel and the appendix of the paper. However, we would be glad to further clarify our position in a meeting or through any other format the reviewer finds appropriate.
> >
> > We would also like to emphasize that the privacy risks associated with FedGCN go beyond simply reconstructing the adjacency matrix. The protocol involves sharing the sum of node features, which leaks information about the individual features (see reference Ngo et al. 2024 in the paper). Therefore, the privacy concern about FedGCN remains valid regardless of the adjacency information.
> >
> > Finally, with due respect, we believe that the review score should primarily reflect the originality and contributions of our work. A disagreement over a point related to existing literature, while worth discussing, should not disproportionately influence the overall evaluation.

---

> ### Comment · Reviewer_TSTT · 2025-03-24
>
> It is clear that in FedGCN, the server just needs to know the local node id and the 1-hop node id of each client, where the connection between nodes is unknown. https://github.com/FedGraph/fedgraph/blob/74292e3bc6cd28a9052c914d55e380be071e2cb8/fedgraph/federated_methods.py#L270
>
> The authors have clear misunderstanding and are unwilling to modify the paper as promised. It is a serious ethics issue.

---

> > ### Comment · Reviewer_TSTT · 2025-03-24
> >
> > I understand that the authors may not understand the FedGCN algorithm. I would suggest the authors read through the paper, the algorithm, and code carefully. I am happy to elaborate more to help the authors.

---

> ### Comment · Reviewer_TSTT · 2026-04-08
>
> Just recall the paper after reviewing ICLR 2026. If there is no update, I will send a message to program chair.